# An ERA5 Climatology of Synoptic-Scale Negative Potential Vorticity-Jet Interactions over the Western North Atlantic

Alexander Lojko[1], Andrew C. Winters[2], Annika Oertel[3], Christiane Jablonowski[1], and Ashley Payne[1,4]

[1]Department of Climate and Space Sciences and Engineering, University of Michigan, Ann Arbor, MI 48109, USA
[2]Department of Atmospheric and Oceanic Sciences,University of Colorado: Boulder, Boulder, CO 80309, USA
[3]Institute of Meteorology and Climate Research Troposphere Research (IMKTRO), Karlsruhe Institute of Technology, Karlsruhe, Germany
[4]Tomorrow.io, 9 Channel Center St, 7th Floor, Boston, MA 02210, USA

**Correspondence:** Alexander Lojko (alojko@umich.edu)

**Abstract.** Recent numerical modelling and theoretical work deduce that potential vorticity (PV) can turn negative in the Northern Hemisphere as a result of localized, convective heating embedded in vertical wind shear. It is further postulated that negative potential vorticity (NPV) may be relevant for the large-scale circulation as it has been observed to grow upscale into elongated, mesoscale bands when in close proximity to the jet stream, accelerating jet stream winds and degrading numerical weather

prediction skill. However, these findings are largely confined to case studies. Here, we use a climatological and composite perspective to evaluate the occurrence of elongated bands of NPV over the North-West Atlantic and its implications for jet stream dynamics. This research focuses on synoptic-scale bands (>1650 km) of NPV that are in close proximity (<100 km) to the jet stream (termed NPV-jet interactions) using ERA5 data from January 2000 – December 2021. Climatological characteristics show that NPV-jet interactions occur most frequently over the coastal West Atlantic during Boreal Winter along

40°N. This latitude band has also seen an 11% increase (relative-change) in NPV-jet interactions over the 22-year time-period. Separating NPV-jet interactions into three distinct, large-scale flow patterns using K-means clustering conceptually illustrates the evolution of NPV features from their initial formation along the westward flank of the ridge to the eastern flank of the ridge. The large-scale environment of NPV-jet interactions is characterized by a trough-ridge couplet adjacent to positive integrated vapor transport (IVT) anomalies, conducive to warm-conveyor belts and mesoscale convective systems. Even when NPV is

positioned in a more adiabatic environment (far away from regions of strong IVT anomalies), robust, positive PV gradient and wind speed anomalies exist along the jet stream. Inspecting three detailed case-studies that serve as archetypes to the three clusters, it is shown that the presence of NPV near the jet stream adiabtically enhances wave activity flux due to NPV mutually strengthening momentum transport and the ageostrophic flux of geopotential. The results show that the close proximity of synoptic-scale NPV to the jet stream is conducive to the occurrence of wind speed maxima, and could be dynamically relevant

in enhancing downstream development, despite NPV's theorized origin from sub-mesoscales.

# 1 Introduction

The mid-latitude jet stream is a fast flowing current of westerly air in the upper-levels of the mid-latitude troposphere. Daily variations of the jet stream wind speeds are influenced by transient eddies (Lorenz and Hartmann, 2003; Barnes et al., 2010). These eddies, which can include a strong contribution from cloud diabatic heating processes, act to perturb the jet stream's large-scale circulation away from its base state(Woollings et al., 2016). Notably, the mechanisms involved in the interactions between cloud processes and the large-scale circulation remains a significant source of theoretical uncertainty (Bony et al., 2015; Ceppi et al., 2017; Baumgart and Riemer, 2019). This uncertainty is reflected in practical applications, particularly in numerical weather prediction, where the quality and reliability of weather forecasting models can be rapidly degraded by cloud microphysical processes interacting with the large-scale flow (Rodwell et al., 2013; Gray et al., 2014; Grams and Archambault, 2016; Grams et al., 2018; Spreitzer et al., 2019). Accordingly, it is crucial to further understand how cloud processes impact jet stream dynamics, both for theoretical advancement and practical applications.

Large-scale cloud systems (i.e., Warm Conveyor Belts, Mesoscale Convective Systems) that develop equatorward of the jet stream are associated with vigorous vertical heating gradients that drive divergent outflow at the tropopause (Wernli and Davies, 1997; Baumgart and Riemer, 2019; Steinfeld and Pfahl, 2019). The irrotational wind field established by large-scale cloud systems make an important contribution to the poleward advection of the outflow and subsequent wind speed enhancements along the jet stream (Grams and Archambault, 2016; Steinfeld and Pfahl, 2019), and has been identified as a key mechanism by which forecast errors along the jet stream manifest (Baumgart and Riemer, 2019; Berman and Torn, 2019). Using a potential vorticity (PV) perspective, the warm and moist air that is brought to the tropopause in cloud systems associated with vigorous diabatic heating can be visualized as an enclosed area of low, but positive PV (in the Northern Hemisphere). Several studies have shown that diabatic processes along the equatorward side of the jet stream substantially contribute to strengthening the PV gradient (Bukenberger et al., 2023; Wernli and Davies, 1997). Sharpening of the PV gradient can serve to kinematically strengthen the jet stream, resulting in faster wind speeds, and the enhanced westerly propagation of Rossby waves (Harvey et al., 2016).

The PV perspective has also been applied to smaller scales to study mesoscale and convective scale weather systems (Hertenstein and Schubert, 1991; Braun and Houze Jr, 1996; Conzemius and Montgomery, 2009; Chagnon and Gray, 2009). At scales where the Rossby number is large, the PV distribution within a diabatic weather system is significantly influenced by horizontal heating gradients, which can lead to the generation of locally strong quasi-horizontal PV dipoles that can exceed +/- 10 PV Units (PVU) (Chagnon and Gray, 2009; Weijenborg et al., 2015). Their generation has been analogously compared to the tilting of horizontal vorticity onto the vertical axis, resulting in a cyclonic and anticyclonic relative vorticity pair (Davies-Jones, 1984; Müller et al., 2020). Composite studies show that quasi-horizontal PV dipoles are coherent features that form around convective updrafts in a vertically sheared environment (Weijenborg et al., 2017; Oertel et al., 2020). Notably, the diabatically reduced PV pole can turn negative (in the Northern hemisphere). PV impermeability theory states that vertical heating gra-

dients cannot lead to the generation of NPV (Haynes and McIntyre, 1990). Hence, NPV preferentially arises from localized, horizontal heating gradients embedded in a vertically sheared environment (Harvey et al., 2020). Oertel et al. (2021) illustrate that a strong upper-level jet and wind shear are key ingredients for elongating convectively generated NPV onto larger scales. Vertical shear stretches convective-scale NPV onto the mesoscales, co-occurring with dilution of the initially strong magnitude NPV towards a near-zero but still negative PV value (Oertel et al., 2020; Prince and Evans, 2022). Gray (1999) also note that convective-scale NPV features may preferentially coalesce together, which could also aid in organizing convective-scale NPV for elongation.

Elongated NPV features have been identified in observations (Harvey et al., 2020) and in a number of non-hydrostatic numerical modelling simulations (Braun and Houze Jr, 1996; Rowe and Hitchman, 2016; Oertel et al., 2020). NPV is unique from other large-scale regions of diabatically reduced PV such as large-scale negative PV anomalies (Hoskins, 1997) as NPV has been linked to the occurrence of frontal rainbands (Bennetts and Hoskins, 1979; Schultz et al., 2000), sting jets (Volonté et al., 2018) and enhanced stratosphere-troposphere exchange (Rowe and Hitchman, 2015). Additionally, since NPV has a negative PVU value, this implies that NPV is associated with hydrostatic, inertial or symmetric instability (Schultz et al., 2000). Oertel et al. (2020) theorized that since mesoscale bands of NPV exhibit negative absolute vorticity (relative vorticity magnitude exceeding the Coriolis parameter) and a temporal persistence of several hours, elongated bands of NPV may likely be analogous to an inertial instability.

When NPV features are within large-scale negative PV anomalies like ridges, the anticyclonic relative vorticity from the NPV has been observed to be an order of magnitude greater than the vorticity from the surrounding large-scale negative PV anomaly (Rowe and Hitchman, 2016; Lojko et al., 2022). The vigorous anticyclonic relative vorticity associated with NPV features is noted to be an important contributor to the enhancement of momentum transport along the jet stream (Rowe and Hitchman, 2016) and may be linked to the occurrence of wind speed maximum (Harvey et al., 2020; Oertel et al., 2020). Misrepresenting anticyclonic vorticity associated with mesoscale NPV in global numerical weather models can lead to the rapid introduction of non-divergent wind errors along the jet stream (Lojko et al., 2022). Additionally, the strong horizontal shear associated with NPV can trigger the occurrence of clear-air-turbulence along the equatorward side of the jet stream (Trier and Sharman, 2016; Thompson and Schultz, 2021).

While the interaction of elongated bands of NPV with the jet stream can be dynamically relevant and have implications for aviation and weather prediction, the aforementioned studies examining NPV interactions with the jet stream employ case-study perspectives. These cases provide detailed mechanistic insights into NPV dynamics. However, a climatological and composite analysis of NPV and its interactions with the jet stream is lacking. Such an analysis will provide insight into common mechanisms and synoptic situations associated with elongated bands of NPV. The obtained results will serve to climatologically identify regions that may be relevant to forecast errors and turbulence associated with NPV features. Two key questions are proposed via the climatology and composite analysis:

- What are the climatological characteristics of elongated bands of NPV when they interact with the jet stream?

- What are the typical circulation patterns and dynamical mechanisms involved when NPV interacts with the jet stream?

The proposed analysis focuses on the Western North Atlantic, a region with numerous well-documented case studies evaluating NPV interactions with the jet stream (Rowe and Hitchman, 2016; Harvey et al., 2020; Lojko et al., 2022). These studies provide a basis for comparison against the proposed composite work. The region of Eastern North America and the West Atlantic is frequently associated with convective storms (Li et al., 2020) and warm conveyor belts (Madonna et al., 2014) which develop in close proximity to the jet stream, and are candidate weather events for the generation of elongated NPV features (Clarke et al., 2019; Oertel et al., 2020). Another rationale for focusing on this specific region is that previous composite studies have identified the Western North Atlantic as a mid-latitude climatological hotspot for regions of inertial instability along the tropopause (Thompson et al., 2018).

The paper is structured such that the data and methodology is presented in Section 2. The climatological characteristics of NPV are presented in Section 3.1. The large-scale circulation patterns during NPV-jet interactions are evaluated in Section 3.2. Three detailed case-studies of synoptic-scale NPV that focus on the dynamics involved in NPV-jet interactions are presented in Section 3.3. A discussion of synoptic-scale NPV is had in Section 4, with particular focus on its relevance for the large-scale atmospheric circulation. The work is concluded in Section 5.

## 2 Data & Methods

### 2.1 Data

The ECMWF reanalysis version 5 (ERA5; Hersbach et al., 2020) is downloaded at a grid spacing of 0.25° (~31 km) for the years of 2000 – 2021. The fine grid spacing and temporal resolution of ERA5 compared to other global reanalysis datasets provides an opportunity to study elongated NPV features in a climatological framework. PV, geopotential height (Z), horizontal winds and integrated vapor transport (IVT) with 6-hourly resolution are downloaded directly from the ERA5 archive. Data is predominantly obtained at 250 hPa to focus on how NPV interacts with the mid-latitude tropopause. While it is more common for PV analysis to be performed on isentropic levels, the isentropic level associated with the tropopause can vary notably with respect to season (Röthlisberger et al., 2018). Using only one particular isentropic level can miss NPV features depending on the season examined. Hence, for simplicity, a single isobaric level that is representative of the tropopause during all seasons is selected as isobaric surfaces near the tropopause tend to vary less with height across seasons. It is also noted that elongated bands of NPV are maximized in frequency at specific isobaric levels near the tropopause (Fig. A1). Prior to any further climatological analysis, the latitude and longitude data are filtered to only include the Western North Atlantic (25°N, 100°W)–(65°N, 50°W). An additional 10° buffer is kept on each side of the domain to minimize any boundary effects when

identifying NPV features.

## 2.2 Methods

### 2.2.1 NPV-Jet Interaction Algorithm

An algorithm is designed to search for time-steps where elongated NPV features are in close proximity to the jet stream, specifically < 100 km to the jet stream. Henceforth, these time-steps are termed NPV-jet interactions. The algorithm is split into three parts. First, the identification of elongated NPV features. Second, the identification of jet stream features and, last, identification of instances when elongated NPV is within close proximity to the jet stream. A schematic is provided below (Fig. 1) to give a general overview of the methodology. PV data is bilinearly interpolated to 0.5° to improve computational

efficiency. This interpolation has no impact on the final results relating to the frequency of NPV-jet interactions.

The first step in the algorithm is to identify and label all connected regions of NPV, specifically where PV is =< -0.01 PVU. The major axis length-scale is then calculated for each label by using the two latitude, longitude coordinate pairs that are located furthest away from each other within the NPV label. Only the longest 2% of features are kept. This threshold corresponds

to NPV length-scales longer than 1650 km (Fig. 2a). These features are henceforth referred to as a synoptic-scale NPV features. As a sanity check, the 97th and 99th percentile are also tested; however, these thresholds do not impact conclusions made regarding NPV-jet interaction frequency. The final step involves obtaining and saving the coordinates of the synoptic-scale NPV features.

For jet stream identification, contours where the PV field has a value of 2 PVU are identified (Barnes et al., 2010). Prior to jet stream identification, time-steps in which no synoptic-scale NPV features were identified are filtered out. Next, the PV field is smoothed by a 10 point Gaussian smoother to improve contour identification by smoothing out mesoscale PV filaments. Further filtering of contours is applied by only keeping circumpolar contours. Circumpolar contours are defined as continuous contours of 2 PVU that extend across the longitudinal extent of the NPV-jet interaction domain. Non-continuous contours are

likely associated with cut-off features (Hoskins, 1997) and are chosen to be removed. A small subset of times (4 time-steps) were not able to identify continuous 2 PVU lines. These times were also filtered out from the analysis. The remaining 2 PVU contour coordinates are saved for evaluation with respect to the identified synoptic-scale NPV features.

In the final step of the algorithm, we use the coordinates from the synoptic-scale NPV feature and 2 PVU contour to find the

150 minimum distance between the two features using the Haversine formula. If an NPV feature is within 100 km to the jet stream, the event is retained. If the minimum distance between NPV and 2 PVU exceeds 100 km for a particular event, the event is filtered out. An interaction point is also defined, which is the coordinate of the 2 PVU contour that is in closest proximity to the synoptic-scale NPV feature. The interaction point is used to perform the centered composites described in the latter section of

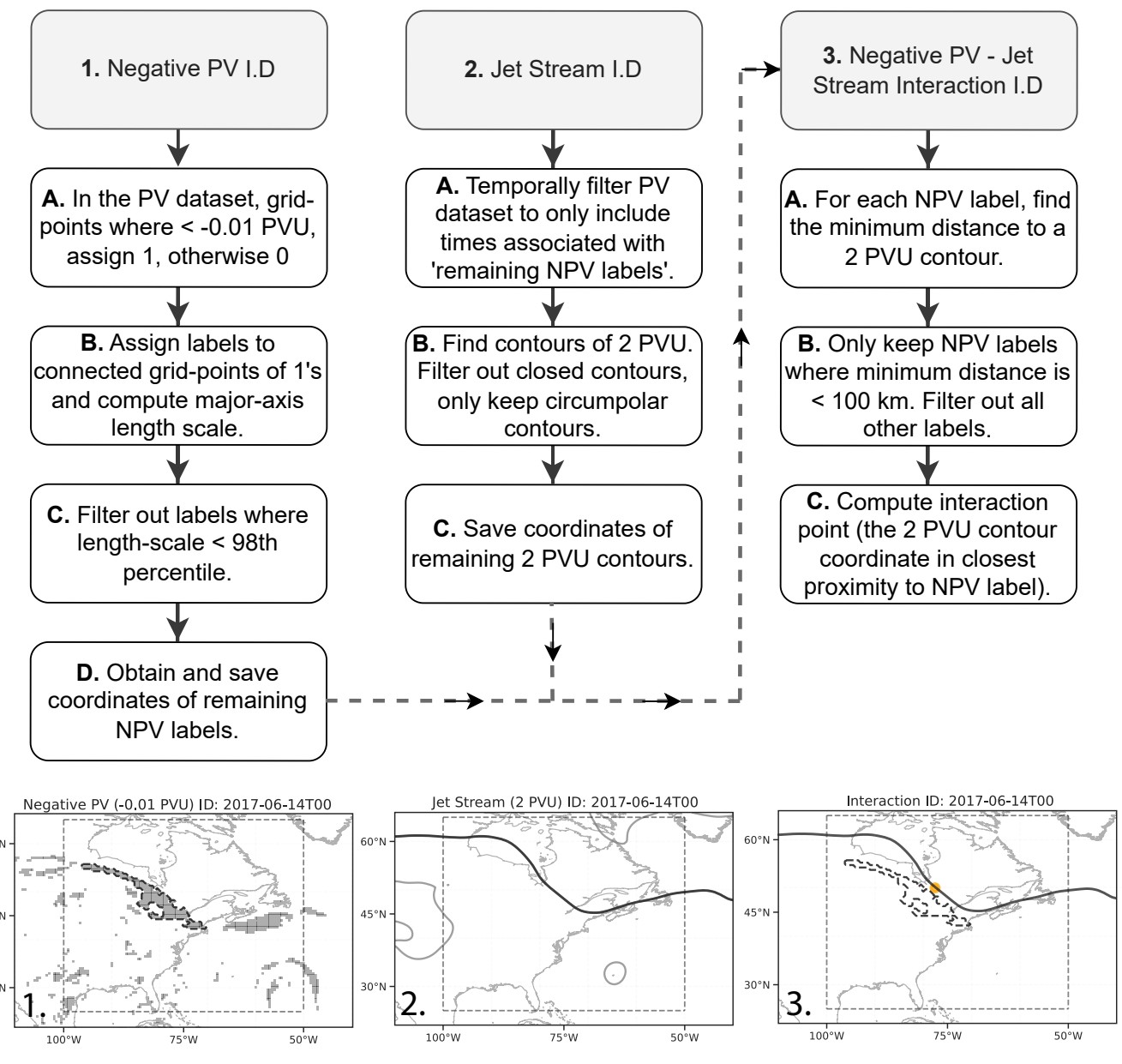

**Figure 1.** Schematic overview of the NPV - Jet Interaction Identification Procedure outlining three key steps in the algorithm design. The images at the bottom of the schematic denote how the algorithm works on a single case. Grey shading shows individual NPV labels. Dark dashed contours denote the perimeter of synoptic-scale NPV features. Dark, continuous contours illustrate the 2 PVU contour (jet stream). The orange circle denotes the 'interaction-point', which is defined as the 2 PVU contour coordinate that is in closest proximity to the synoptic-scale NPV feature. The grey dashed box shows the domain over which NPV-jet interactions are searched for. Identified synoptic-scale NPV features can lie outside of the domain as long as the interaction point itself lies within the domain.

the methods. Sometimes, multiple synoptic-scale NPV features are detected within the 100 km threshold for a particular time-step. In these instances, all NPV features are retained for the climatological analysis. It is also worth noting that no temporal filtering is applied in this study. Hence, instances of consecutive time-steps with NPV-jet interactions are retained as we are not focused on evaluating the life-time of the NPV features in this study.

To provide analogue caes for NPV-jet interactions, the algorithm is also modified to search for synoptic-scale NPV features within 100 - 300 km to the 2 PVU contour. These events are specifically referred to as NPV-jet (100-300 km) interactions. Note that when referring to synoptic-scale NPV features that are within 100 km to the jet stream, these instances are always referred to as NPV-jet interactions. However, when simultaneously comparing both of these interaction distance thresholds, we will use the terminology: NPV-jet ($<$ 100 km) and NPV-jet (100-300km) interactions to prevent confusion.

In total, 21341 synoptic-scale NPV features are detected within the domain during the 22-year time-period. From those, 4983 (23%) synoptic-scale NPV features are detected within 100 km of the 2 PVU contour. Some statistics of the NPV features are detailed in Fig. 2. Figure 2a shows a histogram of the major axis length of all NPV features detected. The vast majority of identified NPV features in ERA5 have rather small length-scales. Over 80% (more than $10^6$) of identified NPV features in ERA5 are characterized by a major axis length shorter than 200 km. We also note that the synoptic-scale NPV features retained in this study are of much larger length-scale than the mesoscale filaments observed in high-resolution numerical modelling studies (Oertel et al., 2020; Blanchard et al., 2021). Deep convection resolving simulations result in a much noisier PV field compared to the smoother and coarser ERA5 dataset.

For additional reference, the area sizes of all NPV features (no matter their size) are plotted in Fig. 2b. The area sizes of NPV features largely follow the distribution of their major axis length scale (Fig. 2a). The majority (over 90%) of NPV features detected in ERA5 are smaller than 1 x $10^5$ km$^2$ in area size. Figure 2c shows that the frequency of synoptic-scale NPV features decreases quasi-exponentially with distance away from the jet stream. Most synoptic-scale NPV features are identified in very close proximity to the jet stream (approximately 25% of synoptic-scale NPV features are detected within 100 km to the jet stream). Of the 32144 time-steps that comprise the period of study, 14580 have a synoptic-scale NPV feature within the domain (45% of the time). 3845 time-steps have a synoptic-scale NPV feature that is within 100 km to the jet stream (12% of the time-steps detect an NPV-jet interaction). For reference, 3835 time-steps have a synoptic-scale NPV feature that is within 100 - 300 km to the jet stream, (approximately 25% of synoptic-scale NPV features are detected between 100-300 km to the jet stream).

### 2.2.2 Composite Approach

The purpose of the centered composite approach is to identify the typical circulation patterns and kinematic processes that occur when synoptic-scale NPV interacts with the jet stream. When computing the centered composite, the mean interaction

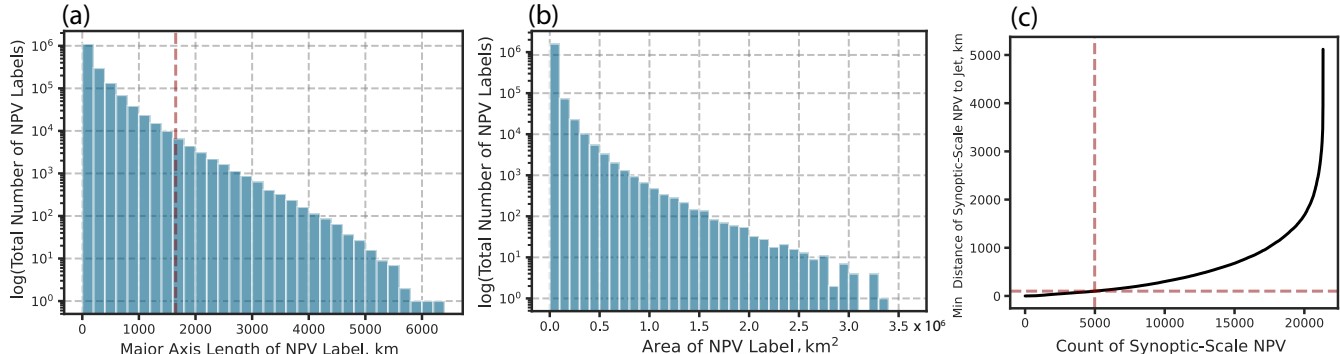

**Figure 2.** Statistics of the identified NPV objects. (a) The total number of NPV objects detected in ERA5 on the y-axis (logged) and the major axis length scale (km) of each NPV object binned into intervals of 200 km. The red dashed lines show where the 98th percentile lies for the major axis length scale. (b) is the same as (a) but with the area size ($\times 10^6$ km$^2$) shown on the x-axis. Binning intervals are set to $0.1 \times 10^6$ km$^2$. (c) shows the distance between each synoptic-scale NPV feature identified and its closest proximity to the jet stream. The red dashed line is used to show where the 100 km threshold lies for NPV-jet interactions (horizontal line) and how many NPV-jet interactions are identified within the 100 km threshold (vertical line).

point is computed from all events. Subsequently, all NPV-jet interaction events are shifted towards the mean using a latitude weighting following Winters (2021). Only unique time-steps are included in the centered composite approach. Hence, in the occasional circumstance that more than one NPV-jet interaction occurs at the same time, only the NPV feature that is in closest proximity to the jet stream is used for centering on the selected time-step.

A problem that arises when computing a mean composite is that important circulation pattern features become smoothed. For example, when computing a principal component analysis on the PV field of all retained events, the dominant modes of variability are characterized by ridging environments (not shown). In contrast, computing a single mean composite of all events illustrates zonal flow. This is due to the synoptic-scale NPV features occurring in different locations along the ridge, hence a single composite centered on the location of NPV-jet interaction leads to too much smoothing of the attendant ridge. To mitigate this effect, K-means clustering is applied to separate events into distinct groups of circulation patterns. The K-means clustering is informed by the latitude weighted PV field. The size of the domain used in the clustering algorithm was a $10°\text{x}10°$ box centered on the interaction coordinate. Enlarging the domain ($15°\text{x}15°$ and $20°\text{x}20°$) had no discernible impact on the K-means clustering results. We also tested the robustness of the clusters by using other fields to organize NPV-jet interaction time-steps such as the use of a binary PV field (1 for stratosphere, 0 for troposphere) in which we obtained similar clustering results.

The number of clusters selected is informed both objectively and subjectively. The Silhouette score is used as an objective metric to determine the optimal number of clusters to use in this study. The Silhouette score metric evaluates how closely

grouped together events are to each cluster centroid (Grazzini et al., 2020). The metric ranges from a score of -1 (poor clustering) to 1 (fully separated clusters). A value of 0 indicates that events tend to equally resemble other cluster centroids (i.e., equal distance to more than one cluster). The Silhouette score is computed for 2-8 clusters. Two clusters provides the highest Silhouette score (0.24). However, it was noted that the interaction point was located in the same region (along the western flank of the ridge) for both of the clusters, so there was not a lot of variability in the location of NPV-jet interactions. Three, four and five clusters provided the next best Silhouette scores (0.18, 0.16 and 0.17 respectively). While the use of more clusters reduced the Silhouette score (the large-scale circulation patterns became more similar between each cluster), there was more variance in the location of the NPV-jet interaction coordinate within the broader large-scale ridging environment. Increasing the number of clusters to three led to an interaction point being located along the eastern flank of the ridge in one of the clusters. Increasing the number of clusters to four and five did not improve the variability of the interaction point location, hence three clusters was subjectively deemed appropriate for the analysis.

As an analogue comparison against NPV-jet ($<$ 100 km) interaction time-steps, NPV-jet (100-300 km) interactions are also evaluated through the centered composite and clustering framework. However, a modification is made to the clustering approach. Each time-step associated with an NPV-jet (100-300 km) interaction is grouped to one of the three previously identified clusters. These time-steps are grouped separately from the NPV-jet ($<$ 100 km) interaction time-steps (i.e., two separate groups of three clusters). NPV-jet (100-300 km) interaction time-steps are assigned to one of the three clusters depending on the similarity of the PV field during NPV-jet (100-300 km) interactions to the mean of the PV field of one of the clusters during NPV-jet ($<$ 100 km) interactions (i.e., how similar is the field of a particular time-step to the cluster centroid?). Similarity is measured using a simple Euclidean distance metric as in Pohorsky et al. (2019). This provides clusters of similar synoptic-scale conditions (i.e., flow analogues) that differ with respect to the distance of elongated NPV to the jet.

### 2.2.3 Kinematic Analysis of NPV

Several methods are used to interpret the impact of NPV on the large-scale flow. Amplification of the jet stream by different components of the wind is examined by calculating PV advection by the irrotational and non-divergent wind (Archambault et al., 2013) at 250 hPa. The partitioning of the winds into these two components is completed via Helmholtz partitioning using the Python Package *Windspharm* (Dawson, 2016), which utilizes spherical harmonics on the global domain. The PV advection fields are computed for each event prior to applying the composite approach.

The wave activity flux (WAF; Takaya and Nakamura, 2001) is computed for each NPV-jet time-step to explain the relevance of how NPV enhances kinetic energy along the jet stream. The horizontal form of WAF can quantify the propagation and energy transport associated with horizontally propagating atmospheric waves along a single level of the atmosphere. WAF assumes quasi-geostrophy and is often used in the evaluation of large-scale flow patterns, particularly in the study of atmospheric Rossby waves and their downstream propagation. The WAF equation is shown below:

$$\overrightarrow{W} = \frac{1}{2|\overrightarrow{U}|} \left( \begin{array}{c} U[\psi_x^{'2} - \psi' \psi_{xx}'] + V[\psi_x' \psi_y' - \psi' \psi_{xy}'] \\ U[\psi_x' \psi_y' - \psi' \psi_{xy}'] + V[\psi_y^{'2} - \psi' \psi_{yy}'] \end{array} \right) \tag{1}$$

U and V denote the base state wind in the zonal and meridional direction, where the base state is determined from the seasonal climatology: Boreal Winter, Spring, Summer and Autumn (DJF, MAM, JJA, SON). $|\overrightarrow{U}|$, which is the combination of the $|(U,V)|$ components, is the magnitude of the base state wind. $\psi'$ is the streamfunction anomaly. Note that the anomaly is computed from the seasonal climatological mean. $\psi'$ is computed from ERA5 wind data using the Windspharm package. The $x$ and $y$ derivatives relate to longitude and latitude. $xx$, $xy$ and $yy$ denote second derivative terms. Derivatives are computed using spherical harmonics.

As mentioned above, two-dimensional WAF is a quasi-geostrophic and dry kinematic metric. While the generation of NPV implies diabatic activity, once NPV grows to mesoscales, it has been observed to persist quasi-adiabatically (Oertel et al., 2020; Lojko et al., 2022). Hence, a dry kinematic approach employed in the WAF equation is assumed to be an appropriate framework to study how synoptic-scale NPV interacts with the jet stream. Individual components of the WAF equation can be assessed to understand the dry, dynamical mechanism by which NPV influences the magnitude of two-dimensional WAF. Note that the phase velocity terms of the WAF equation are not used (Takaya and Nakamura, 2001). This is because only instantaneous time-steps are evaluated, hence Equation 1 assumes the analysis of standing waves.

The first terms inside the square bracket refer to momentum transport, providing information on Rossby wave propagation by geostrophic motion. This term represents the square of the non-divergent wind terms. The second terms in the square brackets refer to the ageostrophic flux of geopotential (Takaya and Nakamura, 2001), and serves as the source or sink for wave activity (Orlanski and Katzfey, 1991). $\psi_{xx}'$ and $\psi_{yy}'$ are of particular interest when focusing on the ageostrophic flux of geopotential as these terms can be easily related to the NPV feature. $\psi_{xx}'$ and $\psi_{yy}'$ represent wind shear anomalies that relate to the $v_x$ and $u_y$ terms of the relative vorticity equation. NPV is uniquely associated with anitcyclonic vorticity maxima (Rowe and Hitchman, 2016; Lojko et al., 2022), hence it is expected that these terms will be magnified in regions of NPV. A partitioning of the WAF equation is performed in Section 3.3 to mechanistically illustrate how synoptic-scale NPV contributes to WAF.

One additional technique that is used involves relative vorticity inversion (Oertel and Schemm, 2021) to illustrate the circulation pattern associated with a spatially confined relative vorticity field. NPV cannot be inverted via PV inversion as it is dynamically unstable (Davis and Emanuel, 1991; Davis et al., 1993; Oertel and Schemm, 2021). In contrast, relative vorticity inversion is independent of the sign of the relative vorticity field. The relative vorticity is equal to the Laplacian of the streamfunction on a horizontal surface. Solving the Poisson equation, and using the streamfunction's relation to vorticity, the non-divergent winds can be obtained along a horizontal two-dimensional surface (i.e., at 250 hPa) (Oertel and Schemm, 2021). The inversion is computed regionally based on using a two-dimensional Green's Function approach. There is no requirement to define a background flow field and we assume zero vorticity outside of the inversion region. The two-dimensional inversion

is useful in examining synoptic-scale NPV, as it appears to largely be a shallow-layer feature that resides within the upper troposphere (Fig. A1).

The relative vorticity inversion is also used to estimate the contribution of anticyclonic vorticity associated with the NPV feature to the non-divergent wind field. Drawing on the theory from Harvey et al. (2020), prior to the development of a weather system generating diabatic heating, a background PV and relative vorticity field exists. The subsequent diabatic heating reduces PV in the upper troposphere and horizontal heating gradients modify the relative vorticity field. To account for this framework, two time steps are compared: one with the synoptic-scale NPV feature present and another prior to its emergence, serving as the base state. This comparison method follows Oertel et al. (2020), involving computing the difference between two fields (prior to and during NPV occurrence). However, the NPV features in this study are larger and more temporally persistent than those evaluated in Oertel et al. (2020). Significant advection of NPV and large-scale flow occurs between the time of the NPV-jet interaction and the base-state. To address this, the base-state relative vorticity field is shifted in coordinate space to ensure the large-scale flow features overlap with each other and that the NPV feature overlaps with the base-state ridge environment. An illustrative example of this adjustment is shown in Fig. B1, and the inversion results are presented in Section 3.3.1.

## 3    Results

### 3.1    Climatology of NPV over the West Atlantic

The climatological frequency of NPV features are presented for the North American-West Atlantic region in Fig. 3. Figure 3a shows the frequency of all NPV identified using ERA5 regardless of size or distance to the jet stream. A meridional gradient of NPV frequency is observed with a maximum in the sub-tropics (>12%) that decreases towards higher latitudes, where the frequency drops below 4% northward of 50°N. The higher frequency of NPV at lower latitudes is also consistent with maxima in inertial instability frequency (Thompson et al., 2018) where anticyclonic relative vorticity is of greater magnitude than the Coriolis parameter.

Other spatial patterns, such as effects of topography, are also captured in the NPV frequency distribution. A low percentage of NPV frequency downstream of the Rocky Mountains (100°W) sharply transitions into a region of higher NPV frequency farther east until reaching a maximum over the coastal Western Atlantic (NPV frequency >9%). The comparatively higher frequency of NPV over the coastal West Atlantic and Eastern North America aligns with regions of strong latent heating attributed to mesoscale convective systems (Liu et al., 2021), and warm Gulf Stream waters that can drive vigorous diabatic weather systems (Minobe et al., 2008). This region also coincides with climatological hotspots of warm conveyor belts (Madonna et al., 2014)

The spatial distribution and percentage frequency of NPV changes when focusing on synoptic-scale NPV features that meet the NPV-jet interaction criteria (Fig. 3b). NPV-jet interaction features are most frequent about the mid-latitudes, with a maxi-

mum (>1.2%) located over the coastal Western Atlantic near 41°N, 65° W. Generally, NPV-jet interaction features are frequent across the coastal Western Atlantic along a latitude band of 40°N. The location of maximum NPV-jet interaction features coincides with warm-conveyor belt ascent climatologies, notably 24 hours after ascent has begun (Madonna et al., 2014; Joos et al., 2023). For comparison, the NPV-jet algorithm is modified to only search for synoptic-scale NPV features within 100 - 300 km of the jet stream (Fig. 3c). The frequency of synoptic-scale NPV features during NPV-jet (100-300 km) interactions still resembles the pattern in Fig. 3b, with NPV-jet interactions maximized over the West Atlantic. However, the general frequency pattern is shifted south and the maxima frequency is located about 5° south of the maxima in Fig. 3b.

To initially motivate the link between NPV-jet interaction events and jet stream dynamics, the frequency of jet streaks is shown in Fig. 3.In this figure, jet streaks are defined as wind speed anomalies in excess of 40 m s$^{-1}$. Figure. 3d shows the frequency of jet streaks when NPV-jet interactions are not occurring. Jet streak frequency is maximized east of 50°W, with maximum frequency values reaching 2% at particular grid-points. Figure 3e illustrates that when NPV-jet interactions are present, jet streaks become much more frequent over the West Atlantic, with maxima values reaching 6%. In other words, jet streaks become 5 times more likely to occur over the Coastal West Atlantic compared to time-steps with no NPV-jet interactions detected. The maximum frequency of jet streaks is located slightly polewards (about 5°) of the maximum frequency of synoptic-scale NPV features (Fig. 3b). Figure 3f shows jet streak frequency by using the 100 - 300 km threshold for NPV-jet interactions. The overall frequency of jet streaks is displaced equatorwards, with the region of maxima being displaced by approximately 2°. The overall frequency of jet streaks also slightly decreases, with maxima values reaching 4.2%. A two sided student t-test is also performed (not shown) at the 2% significance level where it is found that the reduced frequency and equatorward displacement of jet streaks in NPV-jet (100 - 300 km) compared to NPV-jet (<100 km) exhibits statistical significance over much of the West Atlantic. More on the implication of these results is shown in Section 3.2.

NPV-jet interaction frequencies exhibit a pronounced seasonal cycle. DJF is associated with the most frequent NPV-jet interactions (Fig. 4a) with NPV-jet interaction frequencies exceeding 2.5% over the Western Atlantic. MAM NPV-jet interaction frequency (Fig. 4b) maxima reduce to 1.25% and shift westward towards the Eastern US coastline. JJA frequency maxima (Fig. 4c) of 0.5% are predominantly located over continental North America. SON (Fig. 4d) NPV-jet interaction frequency maxima exceeding 1.5% are predominantly located over the Western Atlantic. The spatial location of the JJA frequency pattern suggests that JJA NPV is predominantly induced by continental convection. In contrast, the location of wintertime NPV features over the West Atlantic is consistent with the aforementioned warm conveyor belt climatology. It is also noted that the spatial seasonal variations of NPV-jet interaction maxima agree with the locations of air masses that contribute to downstream blocking (Pfahl et al., 2015; Steinfeld and Pfahl, 2019), which are postulated to arise from rapidly ascending air-streams in regions of strong latent heating. The seasonal co-location of NPV with the Steinfeld and Pfahl (2019) climatology is not necessarily surprising, as NPV is a byproduct of latent heating within convective weather systems (Harvey et al., 2020; Oertel et al., 2020).

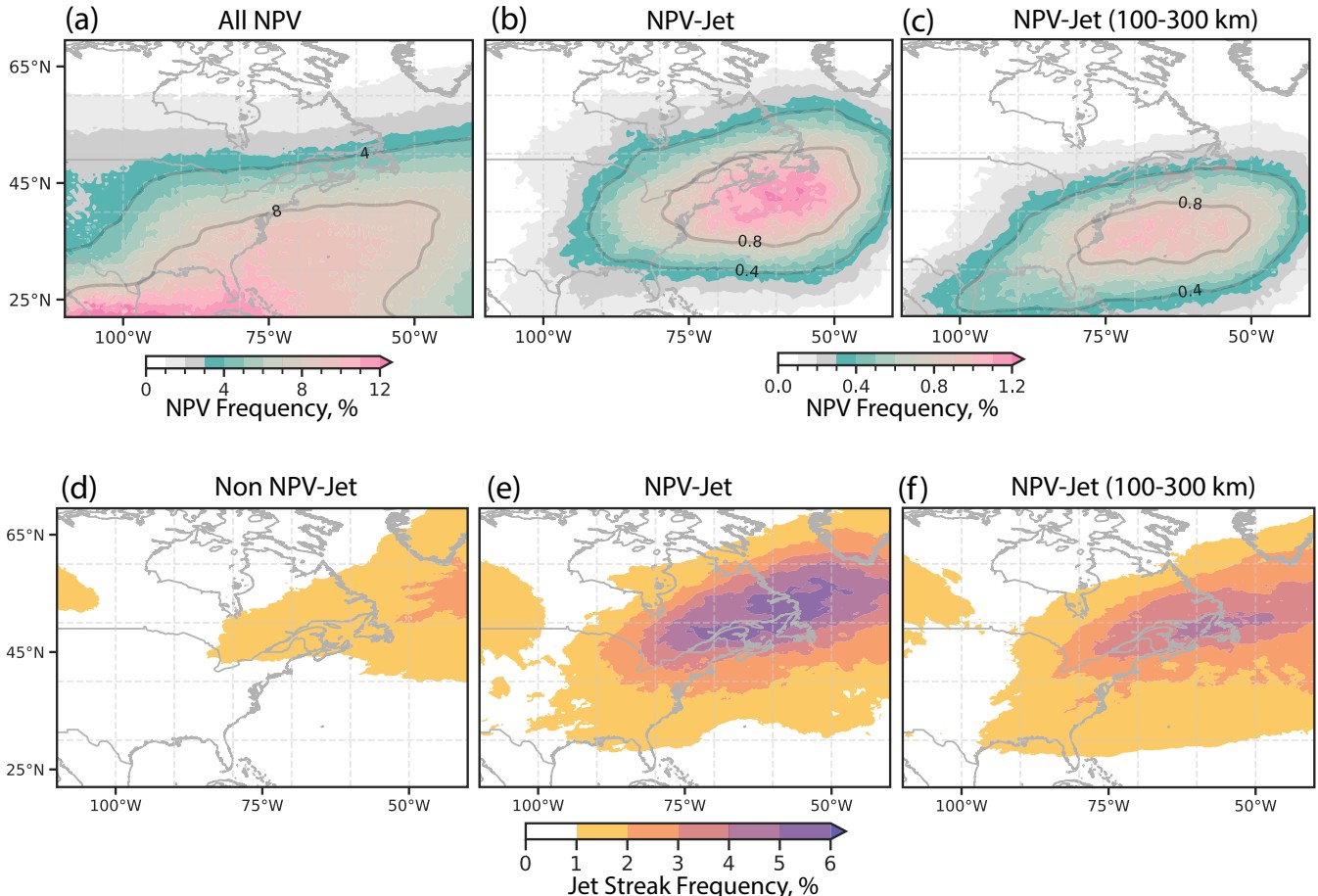

**Figure 3.** Climatological frequency of NPV between 2000-2021 is shown in (a) and (b). (a) shows the percentage of time NPV is observed at a grid-point irrespective of its size. (b) shows the frequency of synoptic-scale NPV features during NPV-jet interactions. (c) shows the frequency of synoptic-scale NPV features during NPV-jet (100-300 km) interactions. The frequency of wind speed anomalies that exceed 40 m s$^{-1}$, termed jet streaks (where anomalies are computed with respect to season) is shown in (d-f). (d) shows the frequency of jet-streaks during time-steps that do not have an NPV-jet interaction over the West Atlantic domain. A random sample of cases are selected such that the number of cases match the number and seasonal distribution of NPV-jet interaction cases. Bootstrapping is then performed 100 times with replacement and a mean is computed. (e) shows the frequency of jet streaks during NPV-jet interactions used to form the climatology in (b). (f) shows the frequency of jet streaks during NPV-jet (100-300 km) interactions used to perform the climatology in (c). Note that time-steps detected in (d), (e) and (f) do not overlap.

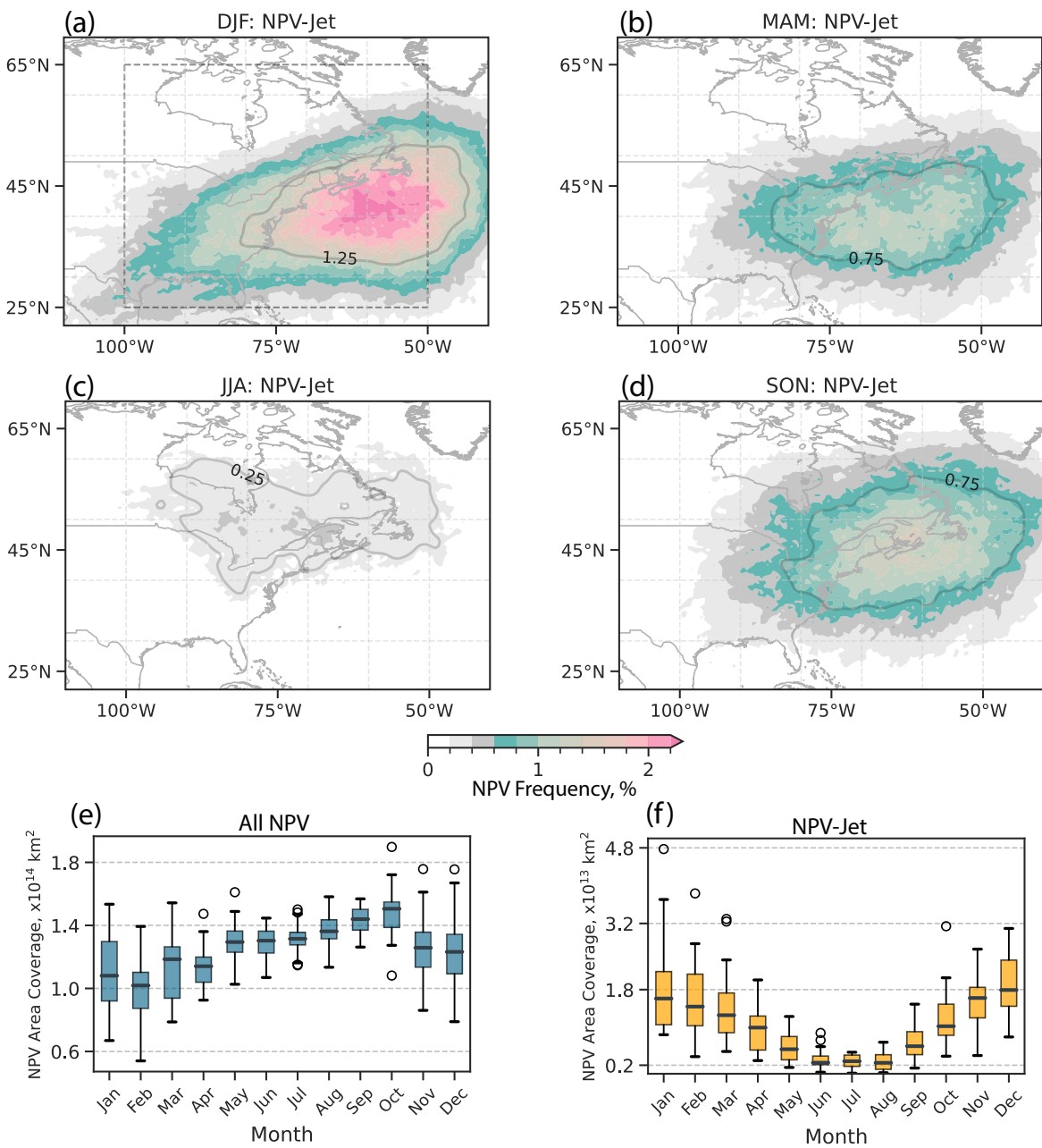

**Figure 4.** As in Fig. 3b but showing the seasonal frequency of NPV-jet interactions between 2000-2021 (a) for DJF, (b) MAM, (c) JJA, and (d) SON. (e) and (f) show boxplots of the monthly area covered by all NPV and NPV-jet interactions, respectively. Area coverage is computed within the dashed box domain shown in (a) (25°N, 100°W)–(65°N, 50°W). The whiskers of the box plot denote the upper and lower extrema. The top and bottom of the box denote the upper and lower quartile. The horizontal lines within the box illustrate the median area coverage of NPV.

Additional detail on the seasonal and spatial climatology of all NPV and NPV-jet interactions is provided in Fig. 4e-f. For all NPV features in the domain (Fig. 4e), the total monthly area coverage of NPV demonstrates a seasonal cycle. NPV area coverage is maximized in October and minimized in February. JJA has higher NPV area coverage in the domain compared to DJF, although, the interannual variability in NPV coverage during DJF is larger and can exceed JJA area coverage for particular years (not shown). It is also noted that the total area coverage for all NPV and NPV-jet interactions is sensitive to the location of the domain. When experimenting with shifting the domain westward, the summer (winter) month frequencies increase (decrease). This relationship reverses when moving the domain eastward.

In Fig. 4f, the seasonal cycle for the area coverage of NPV-jet interactions is shown. In contrast to the area coverage of all NPV occurrences, the area coverage of NPV is now maximized during DJF alongside maxima in interannual variability. JJA has a minimum in area coverage, and an order of magnitude lower area coverage and interannual variability in area coverage relative to DJF. The rapid decrease in NPV-jet interaction area from MAM to JJA is consistent with the climatological, rapid decrease of jet stream wind speeds over the US (Iqbal et al., 2018). The much lower frequency of JJA NPV-jet interaction area suggests synoptic-scale NPV is much less frequent during summer months, with the absence of strong jet stream winds potentially limiting the upscale growth of NPV (Oertel et al., 2021).

Linear trend analysis for NPV frequency from 2000 to 2021 shows an increase in NPV frequency (Fig. 5a), whereby much of the increase is attributed to a narrow latitude band between 35°N-45°N, coincident with the maximum in NPV-jet interaction frequency (Fig. 3b). An increasing, relative trend of 1% per year extends from 100°W-50°W, and farther downstream into the Atlantic. Localized maxima of 3% per year in the linear trend can be observed over the eastern US and coastal Western Atlantic, with an additional increasing trend area that encompasses the Gulf of Mexico and southern US (Over a 22-year time-period, this equates to a relative increase of NPV by 66% in some localized regions).

The relative trend in NPV-jet interactions is weaker but also positive (Fig. 5b), with maximum values exceeding 0.5% per year (a relative increase of 11% at these localized locations over the 22-year time-period). Interestingly, these values coincide with maxima in the trend for all NPV events, predominantly over the eastern US and coastal Western Atlantic. The increasing trend is also generally contained within a narrow latitude band between 35°N-45°N, suggesting that the overall trend is driven, in part, by synoptic-scale NPV features that develop adjacent to, or are advected and interact with, the jet stream. Unlike Fig. 5a, however, the observed increasing trend over the 22-year time-period does not satisfy the false discovery rate.

Addressing the interannual area coverage of all NPV events in the domain (Fig. 5c), much of the increasing NPV frequency occurs after 2010. Prior to 2010, interannual variability of NPV area coverage was considerably less compared to post 2010. In contrast, the interannual variability is much more pronounced for NPV-jet interactions across the full period of study (Fig. 5b). 2010, 2016 and 2019 denote years with maxima in the area coverage of NPV-jet interactions within the domain. Since just 22-years of data are used, the observed trends could be predominantly influenced by decadal variability rather than longer term

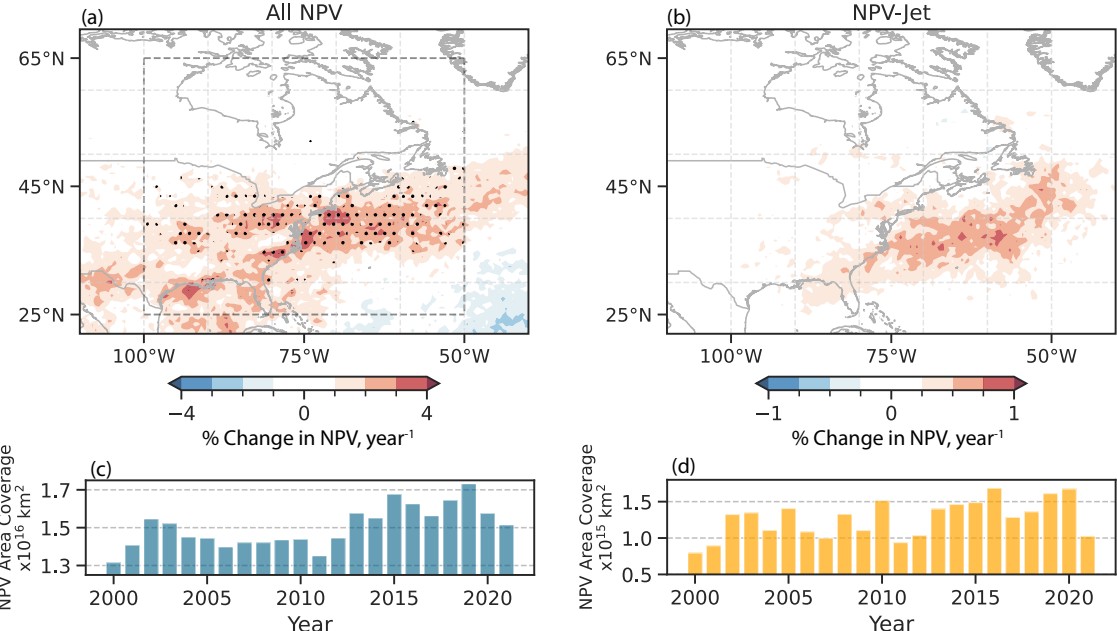

**Figure 5.** Interannual trend of NPV frequency from 2000-2021 expressed as a relative percentage change. (a) shows the total change in NPV frequency per year at each grid-point. (b) shows the same as (a) but for NPV-jet interactions. The false-discovery rate (FDR) is applied to provide a more conservative estimate of statistical significance regarding observed trends (Wilks, 2016). Grid-points that exhibit statistical significance at the 2% level ($\alpha = 0.02$) are shaded. Grid-points where the p value is small enough to satisfy the FDR criterion ($\alpha FDR = 0.1$) are stippled. (c) and (d) shows the total area covered by all NPV (c) and NPV-jet interactions (d) per year computed within the domain (25°N, 100°W)–(65°N, 50°W).

climatic trends. However, recent work published by Lee et al. (2023) indicates that NPV at 250 hPa during the winter months over the Western Atlantic has experienced a statistically significant (using a Student's t-test) increasing trend over a 40-year time-period starting from 1979. Additionally, Prosser et al. (2023) use a variety of metrics related to turbulence (including NPV) over the same 40-year time-period to illustrate a statistically significant increasing trend in turbulence metrics over the continental United States and Western Atlantic. However, our work implies that NPV-jet interactions have approximately increased by a relative amount of 11% during 2000-2021 near the climatological location of the jet stream, and that this pattern is consistently seen even when not filtering for synoptic-scale NPV features near the jet stream.

## 3.2 Circulation Characteristics

### 385  3.2.1  NPV-Jet Centered Composite

To link NPV-jet interaction events over the West Atlantic with distinct circulation pattern, a composite analysis leveraging the use of K-means clustering is performed. NPV-jet interaction events are separated into 3 clusters based on the pattern of the PV field within a $10°\times10°$ box centered on the NPV-jet interaction point. The three clusters reveal ridging environments with different amplitudes and phases (Fig. 6). Cluster 1 shows an amplified ridge with a pronounced trough-ridge couplet.
The interaction point is located on the westward flank of the ridge. Cluster 2 has a broader ridge pattern, with the interaction point also located along the westward flank of the ridge. Lastly, Cluster 3 illustrates another pronounced trough-ridge couplet but with the interaction point located along the eastern flank of the ridge. Note that Fig. 6 illustrates some fields as anomalies while others are not. The caption clarifies which fields plotted are anomalies. All anomaly fields are computed with respect to seasonal climatology.


In Fig. 6a-c, composites of the PV anomalies, PV gradient anomalies, relative vorticity and the frequency of NPV-jet interactions at 250 hPa are shown. The maximum frequency of NPV-jet events lie adjacent to the interaction point (orange dot) along the equatorward side of the jet (per definition of the composite) and is surrounded by a strong anticyclonic relative vorticity field. While not shown, regions of anticyclonic vorticity that satisfy the inertial instability criterion (magnitude of anticyclonic
vorticity > Coriolis parameter) align with the areas enclosed by the 50% contour of synoptic-scale NPV frequency when this contour value is also plotted. The interaction point is straddled by two large-scale PV anomalies of opposing sign. Note that, to a first order, the anomalies result from the large-scale flow pattern due to the presence of the trough-ridge couplet (Teubler and Riemer, 2021). However, NPV features are embedded within the large-scale negative PV anomalies and will thus contribute to the large-scale negative PV anomaly signal. Directly adjacent to the interaction point lies a region of positive PV gradient
anomaly, which reaches maximum values in excess of 2.5 PVU per 100 km. The strengthened gradient lies on the polar side of the 2 PVU contour, where the PV gradient rapidly sharpens towards much higher PVU values (and stratification is particularly large).

In the second set of cluster composites (Fig. 6d-f), wind speed anomalies, upper-level geopotential height anomalies (Z) and
IVT anomalies are shown. In each cluster, the Z anomalies align with the location of the aforementioned PV anomalies. Cluster 1 has the most amplified trough-ridge couplet with both negative and positive Z anomalies reaching magnitudes in excess of 150 m. Cluster 1 also coincides with the strongest IVT anomalies that reach values in excess of 300 kg m$^{-1}$ s$^{-1}$. The close proximity of the IVT anomaly adjacent to the trough indicates a favorable environment for large-scale ascent conducive to squall line and warm conveyor belt development (Dacre et al., 2019).


Cluster 3 features NPV-jet interaction events in a comparatively drier region of the ridge (Fig. 6f). Hence, moist processes are likely to be less important for NPV-jet interactions within this cluster. The IVT anomaly in this case is weakest but still re-

mains positive and in excess of 100 kg m$^{-1}$ s$^{-1}$ near the western flank of the ridge. The weakened IVT anomaly in this cluster may partly result from the region of IVT being further away from the interaction point, and thus more radially smoothed out
by the compositing approach. Nevertheless, this cluster suggests that the IVT anomaly tends to be weaker when synoptic-scale NPV interactions are located within the eastern flank of the ridge. Conceptually, the three clusters resemble the evolution of synoptic-scale NPV features from their initial formation along the westward flank of the ridge, where strong diabatic processes (i.e., latent heating) dominate, through their subsequent advection downstream along the apex and eastern flank of the ridge (Oertel et al., 2020).


The composite of wind speed anomalies shows that all NPV-jet interactions are associated with wind speeds exceeding 40 m s$^{-1}$ (Fig. 6d-f) immediately poleward to the regions of high synotic-scale NPV feature frequency (Fig. 6a-c). This result holds regardless if synoptic-scale NPV is located on the upstream (Fig. 6a-b) or downstream (Fig. 6c) flank of the ridge. The weaker IVT signal in Fig. 6f suggests that a highly amplified PV gradient and spatially coincident positive wind speed anoma-
lies may not require strong, positive IVT anomalies within the near-jet environment, as in Fig. 6d-e. Cluster 3 suggests that synoptic-scale NPV is associated with enhanced jet wind speeds without in-situ influence from moist processes that accompany the positive IVT anomaly. Of course, other larger-scale mechanisms may be co-occurring with the NPV feature, such as supergeostrophic winds around the apex of the ridge due to flow curvature effects (Martin, 2014), which may also contribute to the positive wind speed anomalies.


Figure 6g-i display the ageostrophic wind speed anomalies and PV advection by the irrotational wind (not anomaly). In each cluster, the positive ageostrophic wind anomaly exceeds 15 m s$^{-1}$ and is centered on the interaction point, denoting that NPV-jet interactions are associated with highly ageostrophic environments. The ageostrophic wind speed magnitudes are of equatable magnitude for each cluster. Attributing the ageostrophic wind anomalies to the NPV feature itself is complicated as
the curvature of the flow pattern alongside strong latent heating can also contribute to ageostrophic winds within the near-jet environment (Winters, 2021). The particular amplified flow and much stronger IVT signal in Cluster 1 could provide mechanisms that contribute to the distinct positive ageostrophic wind speed anomaly pattern that cross the westward flank of the ridge (Fig. 6g).

PV advection in Cluster 1 highlights a strong contribution from the irrotational wind field to sharpening the PV gradient along the 2 PVU contour (Fig. 6g). NPV-jet interactions along the westward side of a ridge co-occur with regions of strong divergent wind field anomalies (strong upper-level outflow; Fig. 6g-i). Coupled with positive IVT anomalies, the divergent outflow is likely influenced by strong latent heating in this region (Steinfeld and Pfahl, 2019). Cluster 3 (Fig. 6i) differs with a weaker PV advection signal from the irrotational wind. The irrotational wind vectors do not directly lie over the 2 PVU line,
suggesting a weak influence from the irrotational winds on perturbing the jet stream. Hence, NPV-jet interactions along the downstream flank of the ridge appear to be predominantly associated with PV advection by the non-divergent wind (Fig. 6l). To a first-order, the advection signal from the non-divergent wind is associated with the eastward advection of PV by the mean

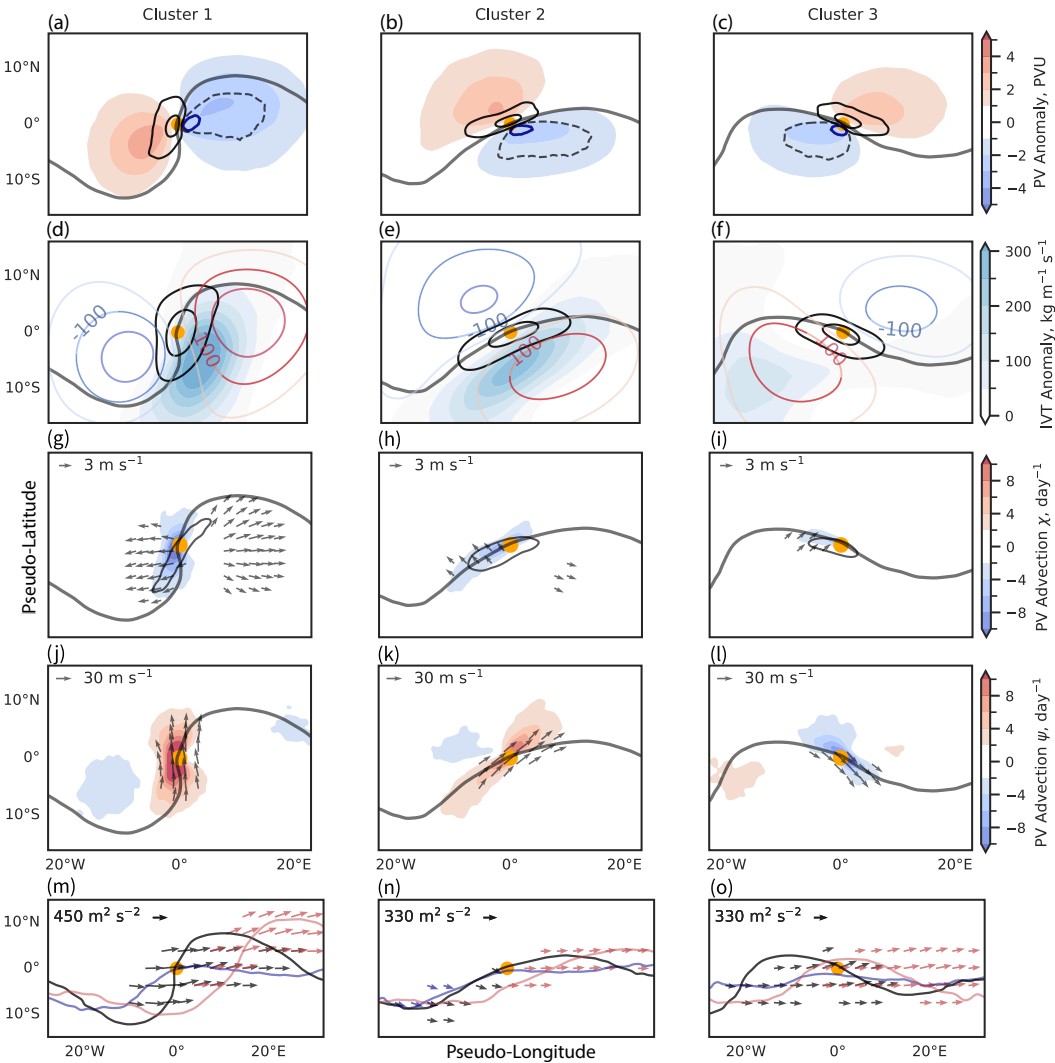

**Figure 6.** The k-means clusters (k = 3) derived from the latitude-weighted PV field at 250 hPa. Cluster 1 = 1136 cases, Cluster 2 = 1458 cases, Cluster 3 = 1251 cases. Fields are plotted at 250 hPa. The x and y axis represent a pseudo latitude and longitude centered on the interaction point (orange dot). (a-c) shows the PV anomaly (shaded, PVU), the magnitude of the PV gradient anomaly (black solid contour: 1.5 and 2.5 PVU per 100 km), and relative vorticity (black dashed contour: $-0.5 \times 10^{-4}$ s$^{-1}$. Grid-points where synoptic-scale NPV frequency > 50% is contoured in blue. (d-f) shows the IVT anomaly (shaded, kg m$^{-1}$s$^{-1}$), Z anomaly contours (red: +50, 100, 150 and blue: -50, 100, 150, m) and positive wind speed anomalies (black contour: 30, 40, m s$^{-1}$). (g-i) illustrates PV advection by the irrotational wind (shaded, PVU day$^{-1}$), vectors show irrotational wind anomalies >3 m s$^{-1}$, positive ageostrophic wind speed anomalies (black contour: 15 m s$^{-1}$). (j-l) shows PV advection by the non-divergent wind (shading, PVU day$^{-1}$). Vectors show non-divergent wind anomalies >30 m s$^{-1}$. (m-o) shows the time-lagged WAF. Black vectors show WAF on the day of interaction. Blue (red) arrows show WAF 24 hours before (after) interaction. The time-lagged 2 PVU contours are shown as solid lines with the same respective colors as the WAF vectors.

flow in all clusters, and consistent with other studies assessing PV advection signals within amplified ridges (Steinfeld and Pfahl, 2019; Winters, 2021). The strongest signal in PV advection by the non-divergent wind arises in Cluster 1, which also coincides with a stronger contribution from the irrotational wind to the total PV advection (Fig. 6g) and is associated with the most amplified flow pattern.

Lastly, the lagged WAF for each of the three clusters is shown in Fig. 6m-o. 24 hours before NPV-jet interaction, the large-scale circulation pattern consists of either a weakly amplified ridge (Fig. 6m-n) or zonal flow pattern (Fig. 6o). WAF is relatively small, and does not exceed the WAF threshold shown in Figs. 6m and 6o. At the time of NPV-jet interaction, the ridge becomes more pronounced and amplified in all clusters. WAF vectors also emerge about the interaction point. The WAF packet emerges predominantly on the equatorward side of the jet stream in Fig. 6n-o. Additionally, the WAF packet is displaced slightly upstream of the interaction point, closer to the base of the trough in Fig. 6n. Examining the WAF 24 hours later, the WAF packet persists following its emission on the day of the NPV-jet interaction event. Furthermore, the WAF packet coherently propagates downstream in all three of the clusters, maintaining similar magnitudes as on the day of the NPV-jet interaction. This maintenance of the amplified WAF packet coincides with the persistence of a more amplified ridge that is of comparable magnitude to the day of NPV-jet interaction. Remember that only the stationary component of WAF (Equation 1) is plotted here, and that there is likely additional contribution from the transient component (Takaya and Nakamura, 2001).

### 3.2.2 NPV-Jet (100 - 300 km) Centered Composite

The NPV-jet (100-300 km) interaction events serve as an analogue to compare against the circulation characteristics during NPV-jet ($<$ 100 km) interactions. In Fig. 7a-c, the large-scale environments largely resemble those illustrated in Fig. 6a-c, with two large-scale PV anomaly dipoles straddling a trough-ridge couplet. The maximum frequency of synoptic-scale NPV features is displaced slightly further equatorward from the jet stream compared to Fig. 6a-c. Some notable differences during NPV-jet (100-300 km) interactions include the occurrence of a much weaker negative PV anomaly and a slightly stronger positive PV anomaly compared to Fig. 6a-c. For reference, the average difference in the negative PV anomaly minima for the three clusters in Fig. 7a-c and Fig. 6a-c is approximately 0.8 PVU. In contrast, the positive PV anomaly maxima difference is about 0.3 PVU.

One reason for observing stronger positive PV anomalies and weaker negative PV anomalies arises from NPV-jet interactions (100 - 300 km) occurring on average at slightly lower latitudes compared to NPV-jet ($<$ 100 km) interactions (Fig. 3b-c). The computation of anomalies is sensitive to latitude. However, the comparatively weaker negative PV anomaly in Fig. 7 implies that processes that lead to deeper, large-scale negative PV anomalies are more pronounced during NPV-jet ($<$ 100 km) interactions compared to NPV-jet (100-300 km) interactions. This coincides with the PV gradient about the interaction point being over 1.25 PVU per 100 km weaker in Fig. 7a-c compared to Fig. 6a-c. A similar response is seen in the anticyclonic relative vorticity field. It is of smaller spatial scale compared to Fig. 6a-c as relative vorticity magnitude is smaller at lower latitudes. It is also noted that the -0.5x10$^{-4}$ s$^{-1}$ relative vorticity contour is located further away from the 2 PVU contour com-

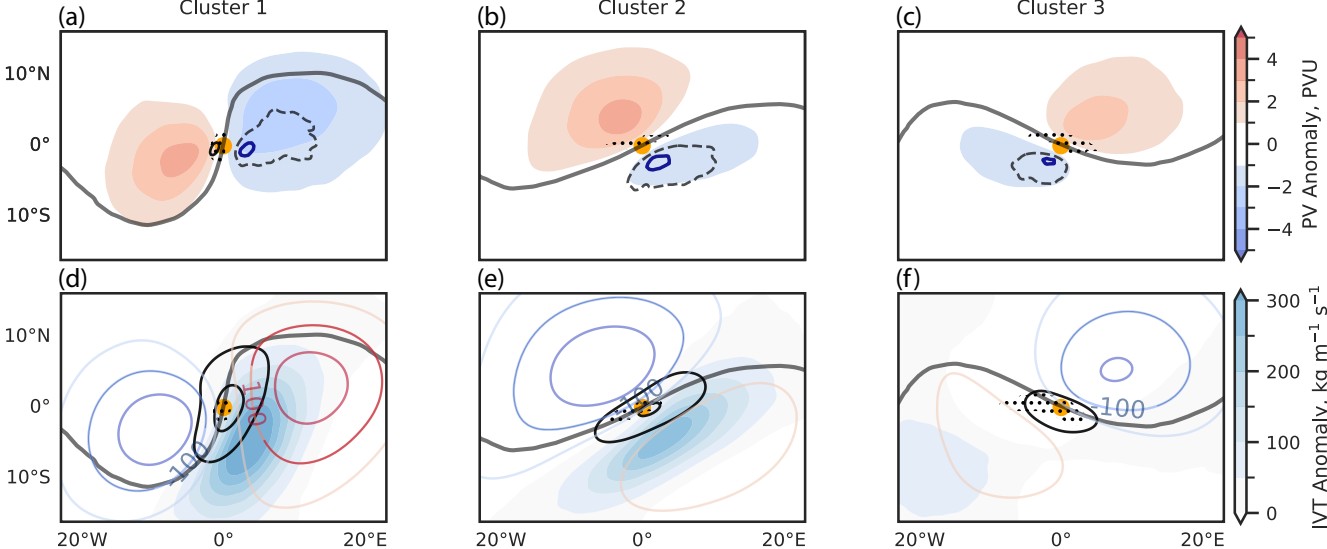

**Figure 7.** As in Figure 6 (first two rows) but for NPV-jet interactions using the 100-300 km threshold. Cluster 1 = 642 cases, Cluster 2 = 1969 cases, Cluster 3 = 1224 cases. Each NPV-jet (100-300 km) event is categorized into one of these clusters based on the event's similarity to the cluster centroids from Figure 6. A modification is made to the NPV frequency (dark blue contour) to show regions where synoptic-scale NPV is more frequent than 45% of the time. Stippling shows statistical significance at the 2% level using a student t-test. (a-c) shows statistically significant differences with respect to Figure 6 and where the PV gradient is greater than 1.25 PVU per 100 km. (d-f) shows statistically significant differences with respect to Figure 6 and where the wind speed differences are greater than 5 m s$^{-1}$.

pared to Fig. 6a-c, consistent with the NPV frequency contour being perturbed further equatorward from the 2 PVU contour. We also note that plotting an inertial instability contour (not shown) using the same contour value for NPV frequency also denotes overlap.

Figure 7d-f reaffirms the results of Fig. 6d-f, indicating that positive Z anomalies are weaker on the equatorward side of the jet stream in Fig. 7d-f relative to Fig. 6d-f, but stronger on the poleward side of the jet stream. Similarly, wind speed anomalies in Fig. 7d-f are comparatively weaker compared to Fig. 6d-f. In particular, wind speed anomaly maxima are weaker by 8 m s$^{-1}$ in Fig. 7d-e and over 10 m s$^{-1}$ in Fig. 7f. It is difficult to ascertain whether the weaker wind speeds result from synoptic-scale

NPV being positioned further away or whether it results from differences in the large-scale fields. The results do confirm that NPV-jet ($< 100$ km) interactions occur in more amplified flow patterns compared to NPV-jet (100-300 km) interaction events, and that the combination of more amplified flow patterns and closer proximity of synoptic-scale NPV to the jet stream coincide with statistically significant enhancements of PV gradient and wind speeds about the interaction point.

The changes in jet stream kinematics associated with diabatic heating are likely to play a lesser role in Cluster 3 (Fig. 7f) due to the distance of the interaction point from the positive IVT anomaly. In this drier region of the ridge, it is suggested that

changes in wind speeds and PV gradient are predominantly driven by adiabatic processes (Bukenberger et al., 2023). Thus, the differences in PV gradient and wind speeds between NPV-jet ($<$ 100 km) compared to NPV-jet (100-300 km) imply that large-scale regions of diabatic heating attributed to the IVT anomaly may not explain the observed kinematic differences, and that their may be significant adiabatic differences between the two distance thresholds. To more explicitly demonstrate the importance of synoptic-scale NPV within the near-jet environment and to better separate its impact from the large-scale flow, the WAF equation is partitioned in the following section.

### 3.3 Archetype Case-Study Analysis

The composite approach is not well suited for evaluating the evolution of temporally fast, mesoscale features. Consequently, a case-study approach is favored to further illustrate the kinematic impacts of NPV on the jet stream for each cluster. Three archetype cases are selected that are most representative of their corresponding cluster. Archetype cases are identified using a Euclidean distance metric (lower Euclidean distance denotes greater similarity to the cluster's mean PV field). From 10 cases for each cluster that best resemble the mean PV field, one case for each cluster is subjectively selected. The subjectively chosen cases are deemed to be the most illustrative for highlighting the influence of NPV features on the jet stream. To summarize, three cases that best represent their respective clusters are chosen to evaluate NPV-jet interactions through a detailed, wave activity flux perspective.

### 3.3.1 Synoptic Overview

In Fig. 8a-c, a synoptic overview is provided for each archetype case. The large-scale circulation for each case illustrates a ridge with strong IVT along its westward flank. Fig. 8a-b depict a synoptic-scale NPV feature predominantly along the western flank of the ridge, while the synoptic-scale NPV feature in Fig. 8c is mainly located along the apex and eastern flank of the ridge. For the second and third cluster, the NPV feature does not overlap with the region of strong IVT, implying the NPV feature is located away from the immediate influence of broad regions of latent heat release. For each case, the point where NPV is in closest proximity to the jet stream coincides with wind speed maxima. Wind speeds in excess of 70 m s$^{-1}$ are observed in Fig. 8b-c, and surpassing 90 m s$^{-1}$ in Fig. 8a).

Figure 8d-f analyzes the relative vorticity field and illustrates its influence on the large-scale flow via vorticity inversion. Regions of anticyclonic vorticity with a magnitude exceeding $1\text{x}10^{-4}$ s$^{-1}$ are predominantly situated within the NPV features, indicating that the features are inertially unstable. Anticyclonic relative vorticity outside of the NPV contour but in the ridge is predominantly of much weaker magnitude. On the polar side of the 2 PVU line, strips of cyclonic relative vorticity are observed adjacent to the NPV feature for each case. An inversion of the relative vorticity within the boxed domain predominantly results in an anticyclonic non-divergent wind field with wind speeds maximized about the 2 PVU contour reaching values of 45 m s$^{-1}$ in Fig. 8d and Fig. 8f. Figure 8e involves an NPV feature adjacent to a strip of cyclonic vorticity exceeding $2 \times 10^{-4}$ s$^{-1}$ and a non-divergent wind maximum of 55 m s$^{-1}$. The vorticity inversion illustrates the influence of the strong cyclonic vorticity in

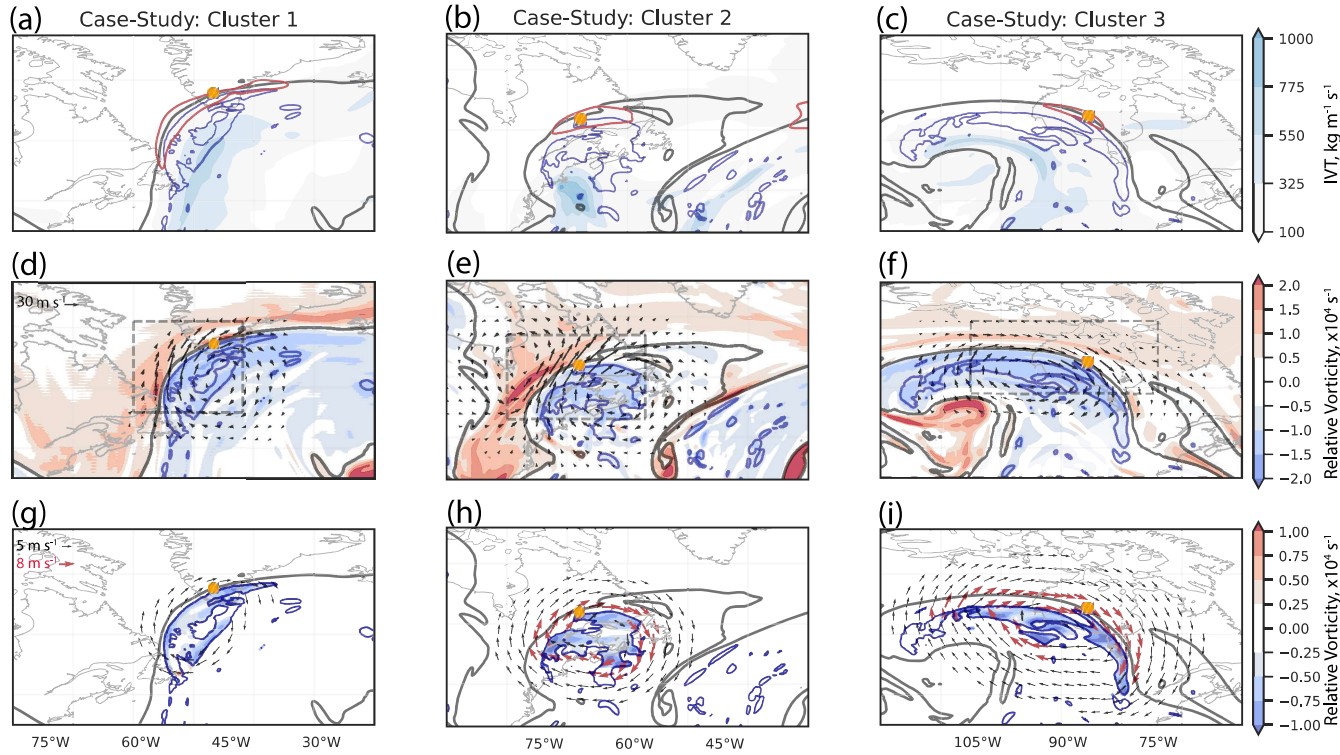

**Figure 8.** Synoptic overview of three cluster archetypes. The Cluster 1 Case (left column) occurs on: 2004-03-24 00 UTC, Cluster 2 Case (middle column): 2014-12-09 18 UTC, Cluster 3 Case (right column): 2021-05-24 18 UTC. In each panel, the 2 PVU contour (black line) and NPV (blue line) are shown. The orange dot illustrates the interaction point for each case (the coordinate at which the NPV feature is in closest proximity to the 2 PVU contour). (a-c) shows the IVT, kg m$^{-1}$ s$^{-1}$ (shaded), and wind speed contours in red at 70 m s$^{-1}$ (the wind speed is shown at 80 m s$^{-1}$ in a). (d-f) shows relative vorticity (shaded), s$^{-1}$, and its subsequent inversion, computed from within the dashed grey box. The vectors are plotted to illustrate the non-divergent winds obtained from the inversion. (g-i) shows the relative vorticity inversion estimating only the contribution to the non-divergent wind field due to the NPV feature with respect to the pre-existing background relative vorticity field. The NPV feature itself is used as the boundary for the inversion. Black (red) vectors denote non-divergent winds exceeding 5 (8) m s$^{-1}$.

this case (Fig. 8e). Namely, two dipoles rotating in opposing directions are illustrated, resulting from the interaction between anticyclonic vorticity (associated with NPV) and cyclonic vorticity on the polar side of the 2 PVU contour (Cunningham and Keyser, 2004; Pyle et al., 2004).

To provide a quantitative (yet conservative) estimate of the kinematic influence of NPV on the jet stream, a relative vorticity inversion is computed for the vorticity associated with the NPV feature (Fig. 8g-i). The relative vorticity attributed to the NPV feature involves estimating the background flow relative vorticity and subtracting it from the total relative vorticity field within the NPV feature. The reader is referred to Section 2.2.3 for further methodological detail. The non-divergent wind speed maximum is approximately 8 m s$^{-1}$ in Fig. 8g, 10 m s$^{-1}$ in Fig. 8h and 11 m s$^{-1}$ in Fig. 8j. Wind speeds then decrease radially outwards with distance from the NPV feature. Although the vorticity inversion provides a conservative estimate, the NPV contributes significantly to the non-divergent wind field, accounting for about 20% of the maximum in Fig.8d-f. Tests with different domains, such as slightly increasing the perimeter about the NPV feature, yields consistent inversion results. However, higher-resolution data and the application of trajectory analysis may be needed for a more robust quantification of the circulation associated with NPV (Oertel et al., 2020).

### 3.3.2 Individual Wave Activity Flux Terms

In this section, the terms that contribute to the total WAF equation are evaluated with respect to the synoptic-scale NPV feature in each archetypal case. As the WAF equation is constructed from multiple different components (Eq. 1), the analysis is narrowed to focus on the first part of the WAF equation, $U(\psi_x^{'2}-\psi'\psi_{xx}')$ (although other terms will also be discussed when appropriate). As will be shown, these terms contribute the most to the WAF equation for each case.

In Fig. 9a-c, the non-divergent wind anomaly magnitude maxima is observed to lie adjacent to the NPV feature along the 2 PVU contour. Drawing insights from the relative vorticity inversion discussed in the preceding sub-section, it is suspected that the maxima in non-divergent wind anomalies stem from the dynamic interplay between the NPV feature and strong cyclonic relative vorticity on the polar side of the jet stream. First, focusing on $\psi_x^{'2}$, which is found in $U(\psi_x^{'2}-\psi'\psi_{xx}')$, and the $\psi_y^{'2}$ momentum term, it follows that the maximum in momentum transport overlaps with regions of strong non-divergent wind anomalies. For each case, $\psi_y^{'2}$ (zonal momentum transport) is maximized along the zonal extent of the jet stream. The $\psi_x^{'2}$ term (meridional momentum transport) is most prominent along the meridional extent of the jet stream (i.e., the western and eastern flanks of the large-scale ridge). For each case, these terms are maximized adjacent to the NPV features. In the three cases presented, $\psi_x^{'2}$ exceeds 2000 m$^2$ s$^{-2}$, reaching its maxima precisely where the NPV feature is closest to the jet stream. The $\psi_x^{'2}$ term reaches 4000 m$^2$ s$^{-2}$ where NPV is in closest proximity to the jet stream in Fig. 9a . The larger magnitude of the $\psi_x^{'2}$ term in this case corresponds to the much stronger non-divergent wind anomalies observed along the entirety of the jet stream. It can be inferred that while the large-scale environment varies in terms of its magnitude of momentum transport across the different cases, the presence of NPV features appears to consistently coincide with a maximum in momentum transport in

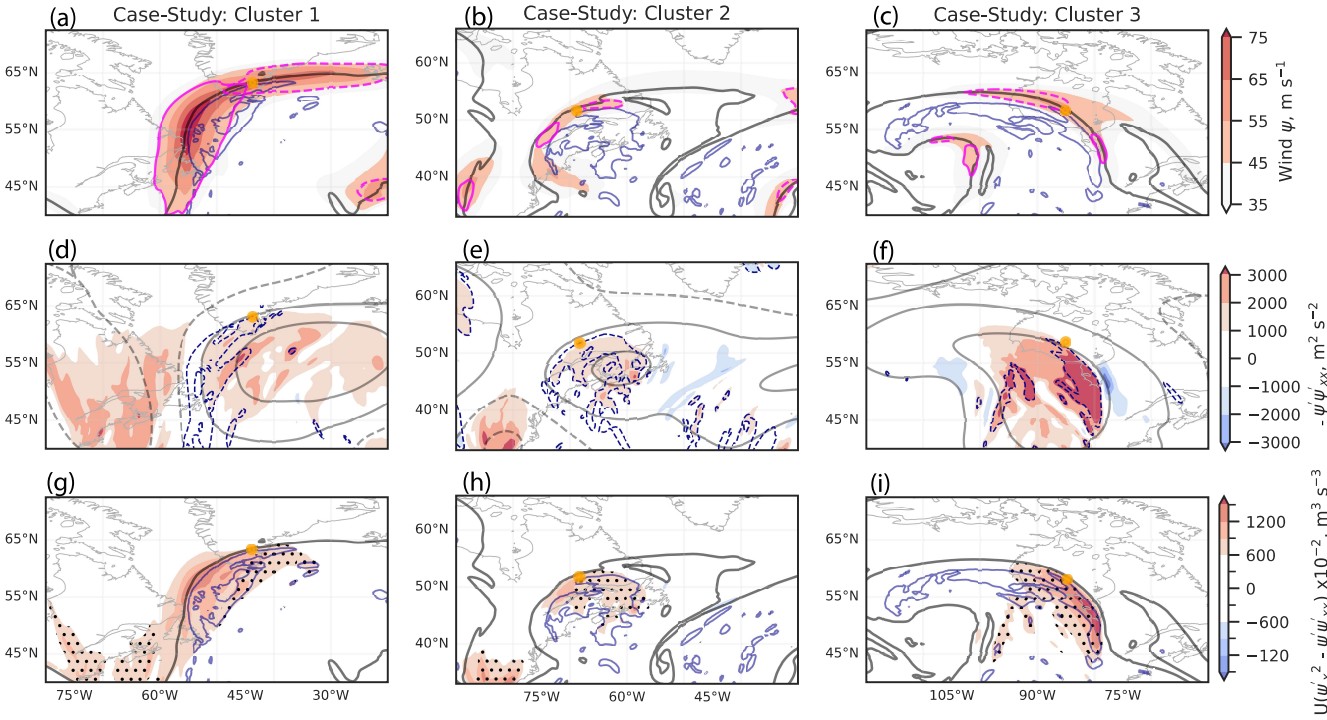

**Figure 9.** Component analysis focusing on the first part of the wave activity flux equation, $U(\psi_x'^2 - \psi'\psi_{xx}')$. (a-c) focuses on momentum: The non-divergent wind anomaly magnitude, m s$^{-1}$ is shaded in red, the magenta contours show the momentum transport anomaly terms at a threshold of 2000 m$^2$ s$^{-2}$. Solid contours denote meridional momentum, $\psi_x'^2$ while dashed contours show zonal momentum, $\psi_y'^2$. The solid dark blue contour shows the synoptic-scale NPV feature. (d-f) examines the ageostrophic flux of geopotential. The solid (dashed) grey contours denote positive (negative) streamfunction anomaly, $\psi'$ at intervals of +/- 1, 3x10$^7$ m$^2$ s$^{-1}$. The dark blue dashed contour denotes a negative shear anomaly $\psi_{xx}'$ at -0.75x10$^{-4}$ s$^{-1}$. The red shading illustrates the $-\psi'\psi_{xx}'$ term, relating to the ageostrophic flux of geopotential. The term is computed by multiplying the streamfunction anomaly and shear anomaly. Note that following sign convention in the WAF equation, a negative sign is placed at the front of $-\psi'\psi_{xx}'$. In (g-i), the entirety of $U(\psi_x'^2 - \psi'\psi_{xx}')$ is shown in shading. Regions where $\psi'\psi_{xx}'$ is of larger magnitude than $\psi_x'^2$ are stippled. The solid dark blue contour is as in (a-c). The orange dot in all panels denotes the interaction point.

all three archetype cases.

In Fig. 9d-f, the ageostrophic geopotential flux is evaluated. Focus is placed on the $\psi'$ and $\psi_{xx}'$ terms of $U(\psi_x'^2 - \psi'\psi_{xx}')$. The ridge is characterized by positive $\psi'$ (positive streamfunction anomaly). The $\psi_{xx}'$ term is related to relative vorticity as it equates to a $v_x$ anomaly. In each case, negative values of this term largely overlap with the NPV feature observed in Fig. 9a-c. This is because NPV is associated with a maximum in anticyclonic vorticity (Fig. 8d-f). It can also be seen in Fig. 9d-f that

negative $\psi_{xx}'$ overlaps with positive $\psi'$. These two terms must be multiplied together to obtain $\psi'\psi_{xx}'$. Given that $\psi'\psi_{xx}'$ is

negative when anticyclonic shear overlaps with a positive streamfunction anomaly, subtraction of this term following the sign convention in $U(\psi_x^{'2} - \psi'\psi_{xx}')$ means that $\psi'\psi_{xx}'$ becomes a positive contribution to WAF.

In Fig. 9e-f, $-\psi'\psi_{xx}'$ is maximized within the ridge where overlap between $\psi_{xx}'$ and $\psi'$ occurs. In Fig. 9d, $\psi_{xx}'$ does not completely overlap with $\psi'$, thus reducing the contribution of $-\psi'\psi_{xx}'$ to WAF. In other words, NPV must be optimally embedded within a ridge environment such that its anticyclonic shear can enhance the ageostrophic flux of geopotential.

In Fig. 9g-i, $\psi_x^{'2}$ and $-\psi'\psi_{xx}'$ are combined to obtain $U(\psi_x^{'2} - \psi'\psi_{xx}')$. In other words, the ageostrophic flux associated with anticyclonic shear is additive with the momentum flux term, $\psi_x^{'2}$. This means that $U(\psi_x^{'2} - \psi'\psi_{xx}')$ positively contributes to the WAF equation (assuming the base state wind, U, is positive). For each case shown in Fig. 9g-i, the region where NPV interacts with the jet stream is characterized by a maximum in $U(\psi_x^{'2} - \psi'\psi_{xx}')$. It is proposed that this maximum arises due to the NPV feature enhancing both the momentum transport and the ageostrophic geopotential flux. In Fig. 9g, the contribution to the $U(\psi_x^{'2} - \psi'\psi_{xx}')$ component of the WAF equation predominantly arises from momentum transport, with ageostrophic geopotential flux becoming more dominant further away from the 2 PVU contour where momentum transport is minimized. In Fig. 9h and Fig. 9i, the ageostrophic flux term contributes more to $U(\psi_x^{'2} - \psi'\psi_{xx}')$ as a result of the weaker momentum transport in the latter two cases. This indicates case-to-case dependence as to which terms contribute more to enhancing wave activity.

### 3.3.3  Full Wave Activity Flux

Following the evaluation of the individual terms in $U(\psi_x^{'2} - \psi'\psi_{xx}')$, the full WAF equation is now evaluated with respect to NPV-jet interactions. Figure 10a-c illustrates spatial agreement between regions where large magnitudes of the full WAF are observed and large magnitudes of $U(\psi_x^{'2} - \psi'\psi_{xx}')$ are observed (Fig. 9g-i). WAF is maximized about the 2 PVU contour in Fig. 10a where momentum transport dominates the contribution to the WAF equation for the case-study for Cluster 1 (fig, 9g). In contrast, WAF is maximized over the NPV feature in Fig. 10c, where the ageostrophic geopotential flux plays a more dominant role over momentum transport (Fig. 9i).

The $U(\psi_x^{'2} - \psi'\psi_{xx}')$ term appears to dominate the WAF signal for each of the three cases. This observation is made due to the similarity of the spatial patterns shown in Figs. 9a-c with respect to Fig. 9g-i. To confirm this, all four terms of the WAF equation are shown as a bar graph in Fig. 10d-f. When focusing on the first two parts of the equation, which contribute to the zonal component of the WAF equation, $U(\psi_x^{'2} - \psi'\psi_{xx}')$ dominates for each of the three cases. In contrast, the $U(\psi_x'\psi_y' - \psi'\psi_{xy}')$ part tends to be weaker. One reason for this occurrence is that in the second part of the WAF equation, $\psi_{xy}'$ is associated with flow divergence. In the cases selected, relative vorticity associated with the NPV feature tends to dominate the WAF signal over divergence, indicating that the NPV region is predominantly defined by its relative vorticity.

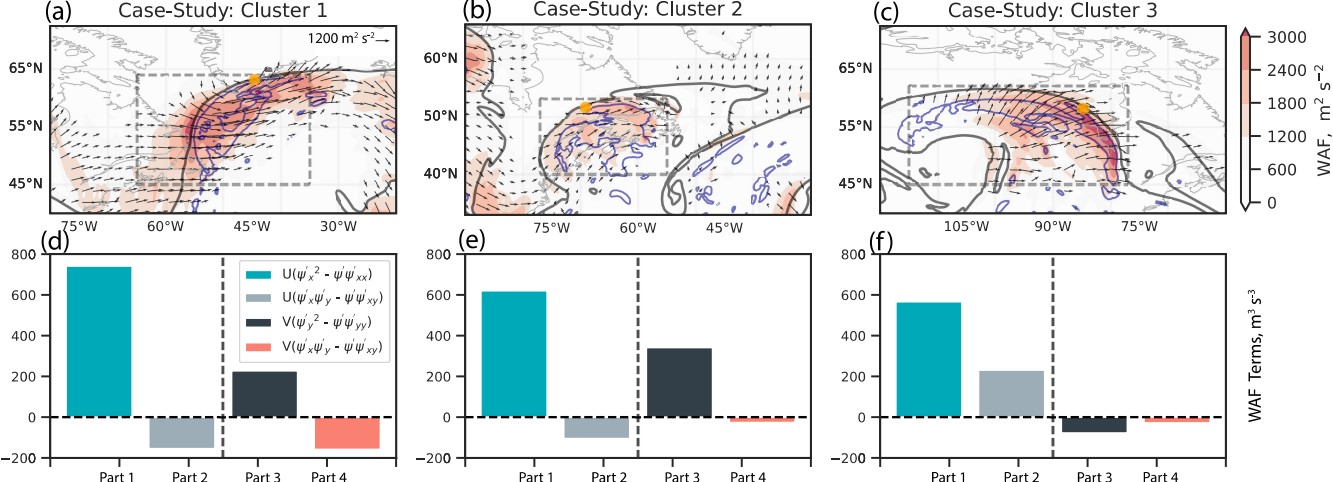

**Figure 10.** Evaluation of all terms in the WAF equation (Section 2.2.3, Equation 1). (a-c) shows the magnitude of WAF, $m^2\ s^{-2}$, shaded in red and the direction of WAF propagation shown by the vectors. All other features plotted are as in Fig. 9 (g-i). (d-f) shows a bar-chart in which the WAF equation is split into its four different parts as shown by the legend with units of $m^3\ s^{-3}$. The units are this way because the $\frac{1}{|\vec{U}|}$ term is excluded. Exclusion of the base state wind term is chosen to more explicitly focus on the momentum transport and ageostrophic geopotential flux terms. Part 1 and Part 2 refer to the zonal component of the WAF equation. Part 3 and Part 4 are associated with meridional component of the WAF equation. The bar charts represent the mean of each WAF part, computed within the dashed box domain from grid-points that exceed $1200\ m^2\ s^{-2}$.

The meridional component of WAF is weaker than the zonal component for each case, which likely results from the meridional base-state wind (V) being weaker than the zonal base-state wind (U) when computing WAF. When evaluating the meridional WAF terms further, the $V(\psi_x'^2-\psi'\psi_{yy}')$ part also dominates $V(\psi_x'\psi_y'-\psi'\psi_{xy}')$ in Fig. 10e-f, implying once more that relative vorticity may be dominating divergence in the boxed regions. However, Fig. 10d highlights the importance of case-to-case variability. Strong shear linked to divergence nearby the NPV feature could also modulate the WAF signal during NPV-jet interactions by potentially enhancing the $\psi'\psi_{xy}'$ parts. Hence, while the WAF equation can be used to clearly illustrate the impact of synoptic-scale NPV features on the jet stream, a partitioning approach is recommended to carefully determine how relevant each term is in the equation for explaining how NPV modifies wave activity.

To summarize, the three archetype cases imply that synoptic-scale NPV within a ridge (on the equatorward side of the jet stream) can be dynamically relevant for enhancing the instantaneous WAF. Figure 11 illustrates two key NPV-jet interaction mechanisms from a WAF perspective. First, stronger anticyclonic vorticity associated with NPV compared to that associated with the broader large-scale ridge within which the NPV is embedded leads to locally enhanced momentum transport, which positively contributes to WAF. Second, strong horizontal shear from NPV embedded within a positive streamfunction anomaly

# Schematic of NPV in ridge at 250 hPa

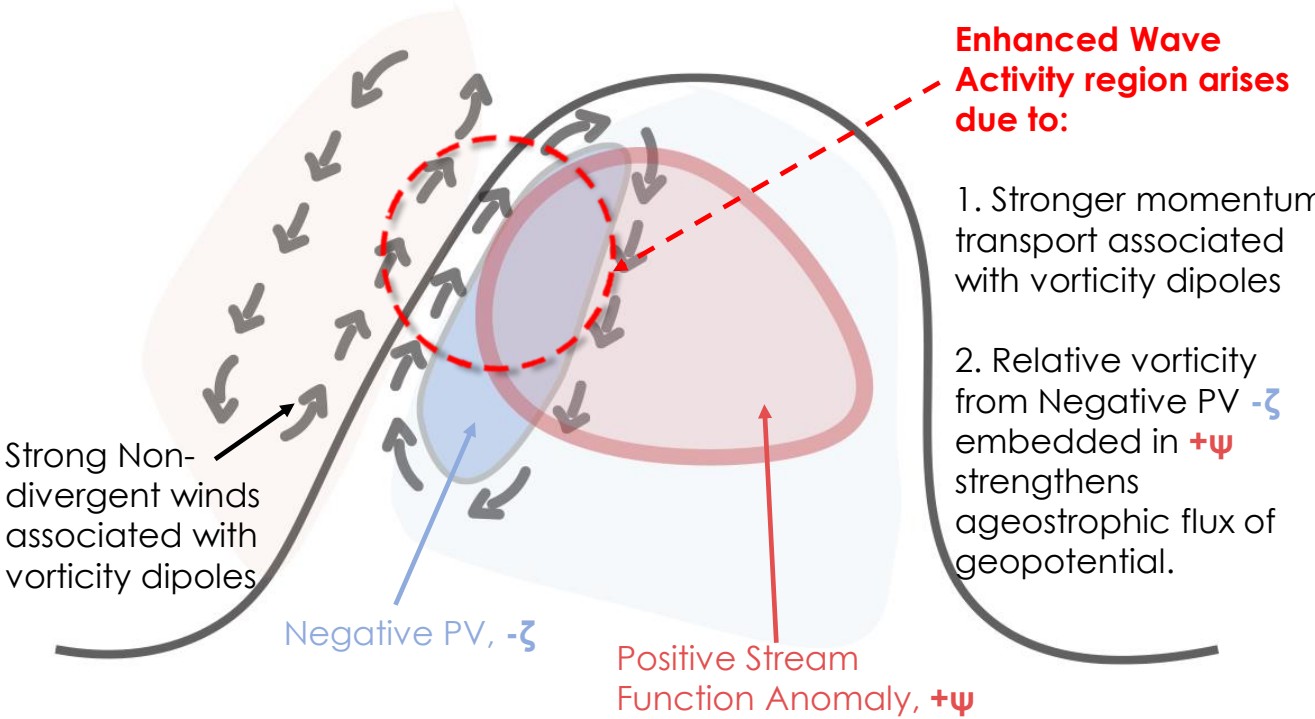

**Figure 11.** Schematic illustrating dry-dynamics mechanisms by which NPV interacts with the jet stream through the wave activity flux perspective. Features of importance are labeled. Other features not labeled include the black contour, representing the 2 PVU line, light blue shading represents a negative PV anomaly associated with the ridge, the light orange shading represents a positive PV anomaly associated with the trough.

(e.g., an upper-level ridge) intensifies the ageostrophic geopotential flux, which also positively contributes to the WAF.


## 4   Discussion

This section contextualizes the results with respect to relevant literature and addresses some limitations of the study. Our climatological analysis demonstrates that NPV occurs more frequently at lower latitudes and decreases towards higher latitudes. These patterns are consistent with other climatological studies of NPV using ERA5 (Lee et al., 2023) and of inertial instability using ERA-interim (Thompson et al., 2018) at 250 hPa. These studies also agree in the observed poleward extension of enhanced NPV (and inertial instability) frequency over the western North Atlantic (Fig. 3a). Lee et al. (2023) demonstrates that the more common occurrence of turbulence indices, such as NPV, over the western North Atlantic coincides with more

frequent instances of negative Brunt–Väisälä frequency, particularly during winter months. We suspect that enhanced negative Brunt–Väisälä frequency over the western North Atlantic may result from the common occurrence of trough-ridge couplets

characterized by positive IVT anomalies (Fig. 6d,e,f), which prime the environment to be more conducive to severe convection (O'Brien et al., 2024). Since NPV is generated in environments characterized by pronounced convective instability (Weijenborg et al., 2017), winter season trough-ridge couplets may provide a suitable genesis region for NPV-jet interactions.

The increasing trend of NPV-jet interactions from 2000 to 2022 (Fig. 5b) could imply that the synoptic setup favoring NPV-

jet interactions has become more common over the last two decades. However, due to the limited time frame of our study, we cannot infer whether the source of this trend is anthropogenic or natural variability. Studies examining longer time-periods suggest that planetary-scale anthropogenic influences could explain positive trends in wind shear (Lee et al., 2019; Prosser et al., 2023) and jet streaks (Shaw and Miyawaki, 2024) due to the strengthening of the meridional temperature gradient through lower stratospheric cooling in polar regions and upper tropospheric warming in the tropics. Given that NPV is closely

associated with jet streaks and strong horizontal and vertical shear (Oertel et al., 2021), we recommend that future research on kinematic jet stream trends also evaluate the significance of more localized dynamical features, such as regional trends in NPV. Further climatological evaluation of NPV is also important in the context of warm-conveyor belts over the West Atlantic, which are projected to become more intense in terms of their diabatic heating and precipitation (Joos et al., 2023), which could facilitate the increased occurrence of NPV features.


Some additional potential biases to consider in our climatological analysis relate to the use of reanalysis data. ERA5 is an optimal dataset due to the dense observational network over the North Atlantic (Tenenbaum et al., 2022). However, improvements in the quality and quantity of observations from 2000 to 2022 (Hersbach et al., 2020) introduce an unquantified bias that could influence trend analysis. More details on this are discussed in Tenenbaum et al. (2022). Another caveat is the resolution

of the dynamical model used to construct the dataset. We advise against solely using dynamical models that do not explicitly resolve deep convection for detailed evaluations of NPV features, as coarse models struggle with resolving NPV features (Clarke et al., 2019; Lojko et al., 2022). While ERA5 uses a dynamical model that parameterizes deep convection, we suspect that the data assimilation process is crucial for correcting for the location of NPV. New regional reanalysis datasets like CONUS404 (Rasmussen et al., 2023) use horizontal resolutions that explicitly resolve deep convection, but higher-resolution simulations

will have noisier PV fields (Oertel and Schemm, 2021). Consequently, a climatological analysis of NPV using datasets such as CONUS404 may identify fewer (more) synoptic-scale (convective-scale) NPV features compared to coarser reanalysis products. Thus, the results presented here are unique to ERA5 and likely to reanalysis datasets of comparable resolution.

The composite perspective indicates that NPV-jet interactions occur within ridge environments in which positive IVT anoma-

lies are present on the westward flank of the ridge and are most frequent during boreal winter months over the western North Atlantic. Oertel et al. (2021) postulate that a pre-existing, strong jet stream is a conducive environment that favors the development of elongated NPV features. Winter months tend to be associated with the climatologically fastest jet stream winds (Iqbal

et al., 2018), hence alongside increased convective instability during winter (Lee et al., 2023), the winter-time maximum in NPV-jet interactions over the Coastal West-Altnatic may arise from the more favorable jet stream environment for elongating NPV features. Further evidence for this hypothesis is presented in Fig. 4e-f, in which, despite the summer months having a 1.5 times greater area coverage in NPV of all sizes, synoptic-scale NPV was 10 times less likely to be observed compared to winter.

The interaction of synoptic-scale NPV with the jet stream in the composite approach highlights the interaction between two opposing large-scale potential vorticity vorticity dipoles. The archetype case-studies provide additional detail, noting that maximum values in jet stream wind speeds are observed to be situated between more localized relative vorticity dipoles, with anticyclonic relative vorticity being linked to the NPV features. Cunningham and Keyser (2004) use a barotropic framework to illustrate that the interaction of two-dimensional relative vorticity dipoles can explain the development of jet streaks, which has also been observed in real-cases within trough-ridge couplet environments (Pyle et al., 2004). In our research, the interaction of anticyclonic vorticity (associated with NPV) with cyclonic vorticity on the poleward side of the jet stream appears to coincide with strong, positive wind speed anomalies along the jet stream. Hence, while our analysis is predominantly focused on the NPV feature itself, it is likely important to also acknowledge the importance of cyclonic vorticity on the poleward side of the jet stream reinforcing with the anticyclonic vorticity associated with NPV.

Computing an inversion of the anticyclonic relative vorticity associated with the NPV features indicates that it substantially contributes to the non-divergent wind speeds along the jet stream. This may explain why wind speed maxima is commonly observed adjacent to synoptic-scale NPV features in the case-study archetypes (Fig. 9), as opposed to elsewhere in the ridge where the anticyclonic circulation is benign. Acknowledging that the archetypes are representative of the composites in Fig. 6, it should also be expected that the circulation associated with NPV contributes to their distinct positive wind speed and PV gradient anomalies about the NPV-jet interaction point. During NPV-jet (100-300 km) interactions, wind speeds and PV gradient maxima are comparatively weaker (Fig. 7). While an explanation could be that the reinforcement between anticyclonic relative vorticity associated with NPV and cyclonic relative vorticity on the polar side of the jet stream is weakened, we also note that the large-scale flow pattern tends to be less amplified. Hence, the weaker jet stream kinematics in Fig. 7 may likely result from the concomitant influence of NPV being positioned further away from the jet and the large-scale flow being less amplified. However, determining which factor is more influential for jet stream kinematics is not accomplished here.

As noted in a review paper by Keller et al. (2019), two frameworks by which to understand the development of jet streaks is through PV advection by the irrotational wind or through the Cunningham and Keyser (2004) approach to define jet streaks via the interaction between two opposing vorticity dipoles. In our composite analysis, strong irrotational wind fields were present in cluster one and two when synoptic-scale NPV was located near the westward flank of the ridge. This suggests that NPV can be embedded within synoptic-scale divergent outflow and could contribute to the strengthening the magnitude of the overall PV advection signal, and thus favor the occurrence of jet streaks. A caveat of this circulation pattern is that the synoptic-scale NPV features are embedded within the diabatically reduced, large-scale negative PV anomaly that is also experiencing advection by

the irrotational outflow. This flow set-up makes it difficult to quantify how much the NPV feature additionally contributes to enhancing wind speeds and PV gradient compared to the surrounding, diabatically influenced PV field.


We do note that one of the NPV-jet interaction clusters identifies that synoptic-scale NPV can also frequently interact with the jet stream even when strong irrotational outflow is weak, such as along the eastern flank of a ridge (Fig. 6i). This location is observed to have the same wind speed anomaly values as the other clusters which have comparatively stronger irrotational wind fields and much more pronounced IVT anomalies. The result suggests that broad regions of strong latent heating in com-

bination with a strong irrotational wind field may not always be necessary to obtain jet streaks when NPV is present. Instead, this cluster suggests that synoptic-scale NPV features may induce adiabatic enhancement of jet stream wind speeds, despite its initial diabatic origin (Bukenberger et al., 2023).

As an additional point of clarification, care must also be taken in treating jet streaks associated with NPV-jet interactions

as two-dimensional, geostrophic features. While our analysis is performed on a single level and appropriate comparisons are made to the two-dimensional idealized jet streaks discussed in Cunningham and Keyser (2004), the composites here identify NPV-jet interactions to be frequently associated with highly ageostrophic environments (Fig. 6g-i). Ageostrophy can significantly modify the intensification of the PV gradient along the tropopause (Winters, 2021) and influence the three dimensional wind speed profile. Hence, a framework that further evaluates ageostrophic effects of NPV would certainly be warranted.


A WAF approach was recently used to show that synoptic-scale NPV can degrade jet stream forecast skill within global numerical weather prediction models (Lojko et al., 2022). Specifically, it was found that the magnitude of the synoptic-scale NPV feature's anticyclonic relative vorticity was under-represented, and coincident with the manifestation of WAF errors. This finding is particularly interesting given that NPV-jet interactions appear to be geographically focused over the western

North Atlantic. Grazzini and Vitart (2015) show that Rossby wave packets initiated over the western North Atlantic tend to be associated with poorer medium-range predictability over Europe. Our study shows that WAF packets (analogous to Rossby waves) tend to be emitted during NPV-jet interactions (Fig. 6m-o) and that synoptic-scale NPV can be dynamically relevant in enhancing the magnitude of WAF (Fig. 10a-f). Particular components of the WAF equation that NPV exacerbates are also summarized (Fig. 11). It would thus be interesting to further explore the relevance of synoptic-scale NPV for predictability

leveraging the dry-dynamics mechanisms identified.

## 5  Conclusions

The study presents a composite overview on the climatology and dynamical impact of synoptic-scale bands of negative potential vorticity (NPV) on the Western North Atlantic jet stream. The study is conducted using ERA5 data. NPV features are

identified using 6-hourly ERA5 data at 250 hPa over the period 2000 - 2021. Using the PV field, PV values =<-0.01 PVU and

>1650 km (>98th percentile) are used to identify synoptic-scale NPV. 'Interactions' of NPV with the jet stream are identified when NPV features are located within 100 km of a circumpolar 2 PVU contour. The 2 PVU contour is used to represent the jet stream and these interactions are referred to as NPV-jet interactions.

The results are split into three parts: A climatological quantification of the frequency of NPV-jet interactions, a composite analysis of the dynamics during NPV-jet interactions, and three case studies involving a mechanistic evaluation of the impact of synoptic-scale NPV on the jet stream through the wave activity flux perspective. The climatological analysis shows that NPV-jet interactions have a probability of occurring up to 1.2% of the time at particular grid-points over the Western Atlantic, maximized at a latitude of 40°N. Interactions are most frequent during winter ($> 2.5\%$) and least frequent during the summer
months ($< 0.5\%$). The seasonal frequencies reaffirm previous case-study work (Harvey et al., 2020; Oertel et al., 2020) hypothesizing that a pre-existing, strong jet stream (which is climatologically more likely to occur in winter) is an ideal environment in which synoptic-scale NPV features can occur.

An investigation of NPV trends in the study region illustrates an increasing trend over a narrow latitude band centered at
40°N encompassing Eastern North America and the Western Atlantic. For (all NPV) NPV-jet interactions, there has been a relative frequency increase of over (45%) 11% in some localized regions of the Western Atlantic over the 22-year time-period based on a linear trend analysis. This result bares resemblance to Lee et al. (2023) who report increasing NPV frequency during Boreal Winter over the past four decades across localized regions of the North-Western Atlantic. Albeit, the spatial extent and magnitude of the trend in Lee et al. (2023) tends to be weaker than the percentages obtained in our work which uses a shorter
(more recent) time-period.

Generally, NPV-jet interactions are characterized by strong, positive PV gradient anomalies (2.5 PVU per 100 km) which coincide with enhanced (ageostrophic) wind speed anomalies exceeding 40 m s$^{-1}$ (15 m s$^{-1}$). Even when NPV features are located away from regions of large-scale diabatic heating (diagnosed simply as positive IVT anomalies), positive kinematic
anomalies persist along the jet stream. The result suggests that synoptic-scale NPV features may resemble a dry dynamics interaction with the jet stream This result is complimented by the strong anticyclonic vorticity circulation associated with NPV compared to the anticyclonic vorticity observed in the background of the ridge (Cunningham and Keyser, 2004). This result is also complemented by previous case-studies that have observed elongated bands of NPV to advect quasi-adiabatically along the jet stream in association with the simultaneous occurrence of jet streaks (Oertel et al., 2020; Lojko et al., 2022).


To gain additional insight on how NPV amplifies the jet stream, the two-dimensional WAF equation at 250 hPa is applied. Composite analysis of the WAF equation denotes that a packet of WAF manifests at the time of NPV-jet interaction and subsequently propagates downstream. This co-occurs alongside the amplification of a trough ridge couplet, indicating that synoptic-scale NPV features tend to develop as the jet stream becomes increasingly amplified. Partitioning the WAF equation
into its individual components for three archetype case-studies illustrates that the anticyclonic relative vorticity associated with

NPV can locally amplify WAF when optimally positioned within a ridge environment. The amplification arises because: First, NPV enhances momentum transport along the jet stream due to anticyclonic vorticity associated with NPV interacting with cyclonic vorticity on the poleward side of the jet. Second, strengthening of the ageostrophic geopotential flux, driven by the horizontal shear induced by the NPV's alignment within a ridge's positive streamfunction anomaly.

Overall, our results reaffirm previous case-study examinations of elongated NPV features. Elongated NPV features tend to exist along the equatorward side of an amplified jet stream pattern (i.e., trough-ridge couplets). In these flow patterns, the close proximity of synoptic-scale NPV features to the jet stream coincides with PV gradient and wind speed maxima. The vigorous anticyclonic relative vorticity within NPV features appears to play an important role in the kinematic strengthening of the jet stream, both in-situ and potentially downstream by serving to additionally enhance wave activity within pre-existing amplified flow patterns.

Designing a more elegant framework for quantifying the role of NPV on jet stream amplification would be a suitable next-step in constraining kinematic uncertainties associated with NPV-jet interactions. While PV inversion might not suit dynamically unstable NPV regions, the application of a two-dimensional relative vorticity inversion (Oertel and Schemm, 2021) and the two-dimensional WAF equation (Takaya and Nakamura, 2001) shows a pathway to quantitatively assess the effect of elongated NPV on the jet stream. A two-dimensional approach may also be appropriate given that synoptic-scale NPV features appear to predominantly occur along isobaric levels related to the tropopause.

More generally, there still remains plenty of mystery regarding the concept and properties of NPV. For example, it remains unknown why synoptic-scale NPV can persist for long periods of time and by which processes NPV dissipates. Further, it would be interesting to synthesize the full life-cycle of NPV from its suspected generation within convective-scale PV dipoles to its upscale growth into an elongated filament of synoptic-scale NPV. Such research will progress our understanding of the dynamics of NPV, and its implications for weather prediction and aviation applications.

*Code and data availability.* The Python based algorithm for identification of negative PV features and their interaction with the jet stream can be downloaded from: https://github.com/AlexLojko/NPV_Algorithm. The Python script for computing a relative vorticity inversion can be downloaded from: https://github.com/evans36/miscellany. The ERA5 dataset is available from https://cds.climate.copernicus.eu/ which is downloaded using era5cli: https://github.com/eWaterCycle/era5cli

## Appendix A: NPV-jet Interactions: Sensitivity Tests

This section illustrates the motivation for selecting the 250 hPa level and the 100 km distance threshold when defining NPV-jet interactions. Figure A1 illustrates the identification of synoptic-scale NPV features for a selected year. NPV features are most frequent at 250 and 300 hPa for all length-scales of NPV. While the amount of NPV features is of the same order at each isobaric level when examining features smaller than 1000 km, the number of NPV features identified at 250 and 300 hPa are an order of magnitude more frequent when identifying NPV features greater than 1000 km. These results indicate that synoptic-scale NPV is predominantly an upper-tropospheric feature, maximized at isobaric levels that are in close proximity to the tropopause.

## Appendix B: Estimating Relative Vorticity Associated with NPV Features in Archetype Case-Studies

An illustration of the process used to estimate the relative vorticity associated with the NPV feature is shown in Fig. B1. The process begins by identifying the time-step of NPV-jet interaction (Fig. B1a,b,c). Next, we manually inspect a previous time-step prior to the development of synoptic-scale NPV. This time-step is treated as the base state time-step. The base state time-step serves as an estimation for the background vorticity prior to the heating that generates the synoptic-scale NPV feature. This technique similarly follows from the approach described in Oertel et al. (2020) for estimating a background flow not associated with the NPV feature. The additional step in this technique is that the base state relative vorticity field is shifted in coordinate space such that the anticyclonic vorticity in the ridge of the base-state overlaps with the NPV feature during NPV-jet interaction (Fig. B1d,e,f). The shifting is latitude weighted. Afterwards, the difference between the two relative vorticity fields is computed (Fig. B1g,h,i), and thus, a rough estimate for the relative vorticity associated with NPV is obtained.

*Author contributions.* AL designed the project, downloaded the data, performed the data analysis and wrote the manuscript. AW and AO contributed to editing the manuscript. AW, AO and CJ worked on discussing the methodology with AL. All authors contributed to discussing the results.

*Competing interests.* The authors have no competing interests.

*Acknowledgements.* AL conducted the research with grant support from the Momental Foundation: Mistletoe Research Fellowship and the National Center for Atmospheric Research Graduate Visitor Program Fellowship. AW was supported by the National Science Foundation under AGS Grant No. 2114011. AO was supported by the Transregional Collaborative Research Center SFB / TRR 165 "Waves to Weather" (www.wavestoweather.de) funded by the German Research Foundation (DFG). The contribution of AO was also partly carried out within

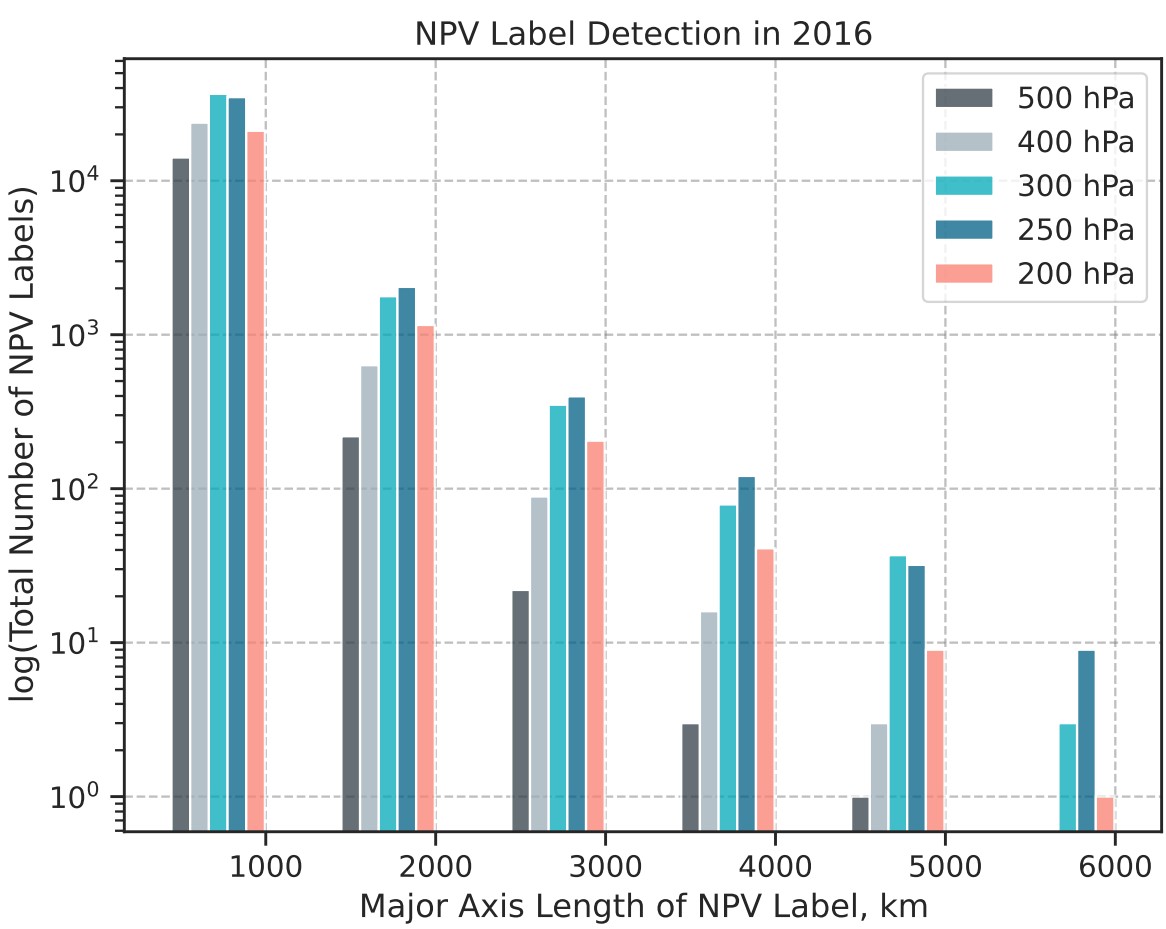

**Figure A1.** Frequency of NPV labels at different isobaric levels in 2016. The year 2016 is selected as it is associated with the most NPV-jet interactions at 250 hPa Fig. 5d. The x-axis illustrates the major axis length scale of NPV labels, binned at intervals of 1000 km. The y-axis shows the logged, total count of NPV labels.

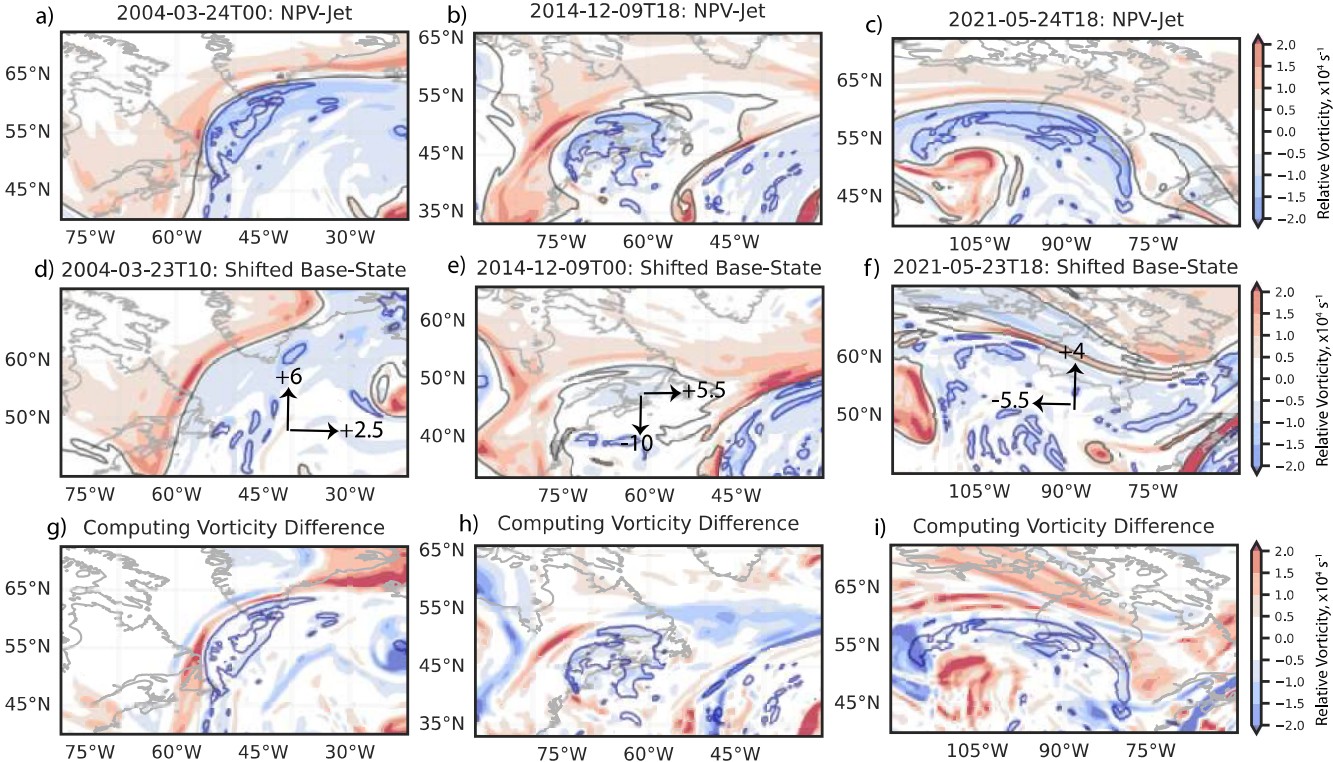

**Figure B1.** Illustration of the relative vorticity field shifting process to estimate the relative vorticity associated with the NPV feature. In the first row (a,b,c), the NPV-jet interaction case is shown. The second row (d,e,f) shows the shifting of the base state relative vorticity field such that the large-scale flow aligns with the NPV-jet interaction relative vorticity field. Arrows denote the direction in which the base state relative vorticity is shifted. The numbers next to the arrows denote the latitude and longitude shifting amount. The final row (g,h,i) illustrates the computed difference.

the Italia – Deutschland science-4-services network in weather and climate (IDEA-S4S; INVACODA, 4823IDEAP6). This Italian-German research network of universities, research institutes and Deutscher Wetterdienst is funded by the BMDV (Federal Ministry of Digital and Transport). Research visits by AL to scientists at the Karlsruhe Institute of Technology and the National Center for Atmospheric Research provided fruitful discussions on the project and the concept of negative potential vorticity. We acknowledge help from Dr. Kevin Prince and Dr. Clark Evans regarding discussions on the relative vorticity inversion script. We also thank the editor and anonymous reviewers for their comments which have served to improve the quality of the manuscript.

830

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
