# Peer review of "An ERA5 Climatology of Synoptic-Scale Negative Potential Vorticity-Jet Interactions over the Western North Atlantic"

_EGUsphere, 2024_

## Referee Comment (RC1)

**Review of «An ERA5 Climatology of Synoptic-Scale Negative Potential Vorticity-Jet Interactions over the Western North Atlantic",**

by Alexander Lojko, Andrew C. Winters, Annika Oertel, Christiane Jablonowski, and Ashley Payne.

**Summary:** In this study, the authors provide a systematic analysis of interactions of 250 hPa negative potential vorticity (NPV) features with the extratropical jet in the North Atlantic sector. The authors first introduce an identification scheme for NPV–jet interactions and then go on to discuss climatological characteristics of NPV–jet interactions in the North Atlantic sector, such as frequency of occurrence, seasonal variability and ~20 year linear trends. Detailed composite analyses of three types of NPV–jet interactions then reveal common dynamical features and differences between NPV–jet interaction events of distinct classes, which are further illustrated based on three detailed case studies. These analyses lead the authors to the conclusion that the presence of NPV features near the jet systematically intensify the latter and are furthermore associated, systematically, with an increase in wave activity, with potential effects on the downstream development of synoptic-scale Rossby wave packets.

Overall, the writing as well as the figures are very clear, and the analyses have been carried out with greatest care. Very evidently, this is work that has been carried out with great attention to detail, which I highly appreciate. However, in its current form the study is somewhat vague regarding the scientific (and any potential societal) relevance of the research that is presented and at times it is not entirely clear why the authors choose certain concepts and analyses to construct their arguments. While I fully see the intellectual appeal in working towards better understanding NPV, I think the authors should invest some time and thought into the framing of their study, and they should state in a very explicit manner why we should all care about NPV. This would avoid conveying the impression that their work is a purely academic exercise (see major comment 1 below). Furthermore, I have a major concern regarding their composite approach, which, in my opinion, might compromise the degree to which conclusions regarding NPV-jet interactions can be drawn from their results (major comment 2). Finally, I wonder why the authors restrict their analyses to a single pressure level, rather than a seasonally varying isentropic level, and I ask the authors to provide more compelling arguments detailing why a single pressure level is the ideal choice (major comment 3).

Despite these concerns I believe the study will be a valuable addition to the growing NPV literature and, if framed more explicitly, it will likely be of relevance also to a broader readership, e.g., colleagues working on clear air turbulence or numerical weather prediction. I thus encourage the authors to revise their manuscript in line with my comments below before I can recommend acceptance of this study.

**Major comments:**

1) The motivation for this work and its scientific (and potentially societal) value needs to be stated more explicitly. In the abstract the authors motivate their work with (lines 2–5) "it is postulated that NPV may be relevant for the large-scale circulation as it has been observed

to …, accelerating jet stream winds and degrading numerical weather prediction skill". Similar statements can be found in the introduction. However, these statements presumably hold for any intense and diabatically influenced negative PV anomaly on the equatorward side of the jet, irrespective of whether the sign of its absolute PV values is negative or positive. In its current form, the manuscript thus does not yet fully convince me that NPV features are a subset of tropospheric negative PV anomalies that warrant special attention.

The authors could alleviate this concern by providing evidence (e.g., from previous studies) that NPV-jet interactions lead to more degraded forecast skills than interactions of benign negative PV anomalies (i.e., with low but positive absolute values) with the jet. Similarly, is there evidence suggesting that CAT is more vigorous/frequent for NPV-jet interactions than for benign "low PV"-jet interactions? If so, please state this more prominently. I acknowledge that, by theoretical arguments, NPV should not be stable and thus it is interesting intellectually to study NPV, which might help to further our theoretical understanding of PV dynamics. In my opinion this puzzling aspect of NPV is a perfectly valid reason for studying climatological characteristics of NPV, but it is not mentioned very prominently as a motivation for their work currently. Rather, the authors currently motivate their study with a need "to further understand how cloud processes impact jet stream dynamics" (line 28). If that is indeed the author's goal, then I believe they look at a subset of events that is unnecessarily narrow. I thus ask the authors to think again about why exactly climatological characteristics NPV–jet interactions in the North Atlantic warrant a dedicated study, and to be more explicit about the benefits and scientific value of this research.

2) It is unclear how much of the signals shows in Fig. 6 (one of the central figures in this manuscript) are due to NPV-jet interactions and how much of these signals simply result from the difference in the large-scale flow during NPV-jet interactions and the climatology. Can the authors exclude the possibility that the composite anomalies in Fig. 6 would look very similar if one considered (instead of the NPV-jet interactions) time steps with just a large-scale flow situation similar to that during the actual NPV-jet interactions? If not, then the key conclusions about the effect of NPV-jet interactions on the upper-level dynamics, e.g., on lines 15–16 "The results show that NPV-jet interactions can in-situ strengthen the mid-latitude jet stream and could be dynamically relevant in enhancing downstream development, …" do not seem justified.

The key question for me is how much of the signals depicted in Fig. 6 would be retained if the authors compared the variables depicted in Fig. 6 during NPV–jet interactions with time steps with a similar large-scale flow pattern occurring without NPV-jet interactions? I think this question needs to be addressed in some form, as otherwise it is very much unclear how much we learn about NPV-jet interactions from the authors composite analysis. As an inspiration, the authors could consider the study of Pohorsky et al. (2019), where these authors faced essentially an analogous problem when examining the interaction between recurving tropical cyclones (TCs) and the jet, which preferentially happen in an amplified flow situation, that, by itself already features considerable upper-level PV and IVT anomalies compared to climatology. Pohorsky et al. addressed the problem by comparing composite fields during instances of actual TC-jet interactions with

a climatology that was constructed from days without TC-jet interactions, but with similar large-scale flow configurations than during the actual TC-jet interactions (i.e., flow analogues). Please consider whether such an approach could be useful here too.

3) I agree that (line 98) "using only one particular isentropic level can miss NPV features", but using only one pressure level has similar caveats. Since you anyways refer to the Röthlisberger et al. (2018) study: Why not choosing a seasonally varying isentropic level that follows the isentropic level of the jet? In that case you would have all the benefits of analyzing PV on isentropic levels whilst still analyzing data from only one level.

**Minor comments:**

1) Line 7: This may be a naïve question, but how sure are you that these synoptic-scale NPV bands in ERA5 are real and not an artefact of the IFS model or assimilation procedure employed in ERA5?
2) Line 9: Not yet clear what the 1.2% mean here. Maybe rephrase to "occur at >1.2% of all time steps at particular grid-points"?
3) Line 34: "causing it to perturb polewards" -> "causing it to move polewards"
4) Line 47: Delete "spatially"
5) Line 63: "stream" instead of "steam"
6) Line 74–75: Related to major comment 1. I don't see why this is an interesting research question. By the PV invertibility principle it is clear that if you move an intense negative PV anomaly (e.g., a NPV feature) towards large positive PV values then the flow in-between accelerates. How could it be any different? Please rephrase this "objective" of the study once you have clarified how you would like to motivate your work.
7) Line 115: How exactly is the major axis length-scale calculated?
8) Comment on Section 2.2.1: I find the description of your algorithm very clear and understandable!
9) Line 172: What is the distance metric that you use in the K-means clustering?
10) Lines 167–188: Why exactly do you use the full PV field for the K-means clustering? This gives a lot of weight to stratospheric PV features (e.g., trough intensity, TPVs etc.), as they simply have larger PV values than tropospheric features. Alternatives focusing more on the large-scale flow pattern rather than stratospheric PV features exist: You could do the clustering on the natural logarithm of the PV field, whose gradients are linearly related to wind speeds (Martius et al., 2010) or you could consider a binary (stratospheric = 1, tropospheric = 0) field to emphasize the geometry of the large-scale flow, as in Pohorsky et al. (2019). Note: I'm not suggesting you need to do that, but perhaps discuss why you chose the full (latitude weighted) PV field for the clustering.
11) Line 186: Delete "enhanced".
12) Line 203: What is the |U-vector| exactly? I assume it is |U-vector| = |(U, V)|, i.e., the magnitude of the base state wind. Currently you just write "|U-vector| is the wind speed". Please clarify.
13) Line 205: How are derivatives computed here?
14) Line 207–208: What does "negligible change to PVU" mean exactly? Do you mean that the PV value of these features doesn't change much?

15) Line 234 and Fig. 3. You report >12% occurrence frequencies of NPV features in the sub-tropics. I find this number surprisingly high, given that these features should not be dynamically stable and, theoretically, should decay very rapidly. I acknowledge your comment regarding the MCS observations, but I still think that more discussion of the magnitude of these numbers is warranted here, including some explanation or hypotheses as to why they are so frequent. Is our theoretical understanding so poor or are they perhaps partly artefacts of the IFS/data assimilation underlying ERA5?

16) Line 238: Brackets around the "Thompson et al (2018)" citation are missing.

17) Line 245 and Fig. 3b: Is it possible that we just see the occurrence frequency of jets in this panel? I'd appreciate some additional panels in Fig. 3, for instance (a) one showing the frequency of jet occurrence, (b) the conditional probability to observe a NPV-jet interaction provided a jet is close by, or (c), conversely, the conditional probability of observing a NPV-jet interaction provided an NPV feature is there.

18) Line 251 and Fig. 3c: Along the same lines as above: Do we see these anomalies in Fig. 3c just because, by design, NPV-jet interactions need a jet streak, i.e., strong winds? Instead of the current Fig. 3c, I'd be more interested in seeing whether wind speeds are stronger during NPV-jet interactions than during instances when "only" a jet is nearby.

19) Line 255: Apologies for potentially misunderstanding something here, but I find this hypothesis test not very convincing. The null-hypothesis you are testing is that there are no wind speed differences between climatology and instances of NPV-jet interactions. However, by definition of NPV-jet interactions, there have to be strong winds (due to the jet). Thus, the result of increased wind speeds compared to climatology is expected by design.

20) Figure 4: Please add a panel on jet frequencies, such that we can see to what extent the results in Fig. 4f simply result from jet frequency variability.

21) Line 272: That's an interesting finding. Could you speculate about why there is larger interannual variability in NPV area in winter compared to summer?

22) Fig. 5 and lines 300–301: What do you mean with the statement "does not satisfy the false discovery rate"? Do you maybe mean "is not significant at alpha=xx, which corresponds to a maximum false discovery rate of 0.1 in Fig. 5b"? Bear in mind that the FDR test is nothing else than a tool to determine on which significance level one should reject the null-hypothesis under scrutiny (see also next comment).

23) Caption to Fig. 5: I find your description of how you determine the significance somewhat confusing. I appreciate that you employ the FDR test, but bear in mind that the FDR test (with e.g., maximum allowed FDR = 0.1) is just a tool to find an appropriate significance level so that less than 0.1*100% of the "discoveries" will be erroneous. That is, the FDR gives you a p-value threshold based on which you deem your results significant or not. Therefore, I think your statement " ... are statistically significant to the 98th percentile" makes no sense. Rather, you should state that you determine the significance level (I called it alpha in the comment above) based on the FDR test with a maximum false discovery rate of 0.1. Ideally, you would also state what that resulting significance level is for panels (a) and (b).

24) Figure 5c,d: Are there reasons to believe that the trends you observe are forced by increasing GHG concentrations or do you believe that they are part of internal variability? Some comments on that important question (also in the text) would be highly appreciated.

25) L316: "patterns" instead of "pattern".

26) Discussion of Fig. 6: Related to major comment 2. Please make it very clear whether you consider the anomalies shown in Fig. 6 as related to the NPV-jet interaction events or whether they are merely features of the composite large-scale flow structure during these events.
27) L350: Do you mean "upstream" and "downstream" part of the ridge or really "poleward or equatorward flank of the ridge" as you state currently. If the latter is the case, then I don't understand this sentence.
28) Figure 6g–l: Please choose the length of the reference vectors such that we can see the vectors in these panels better (i.e., make them larger).
29) L 372–375: I think there is a verb missing in this sentence. As it is now, I don't understand this sentence.
30) Fig. 8d-I and discussion thereof: I appreciate that the authors went through the struggle of applying the WAF diagnostic, but please provide more framing here. Why exactly is this the right method to better understand how NPV features affect wave amplification and propagation here? What exactly do we learn from these analyses in terms of physical and mechanistic understanding?
31) Lines 617–618: Related to major comment 2: I'm not convinced your results allow drawing such a "causal" conclusion. The enhancement of wave activity propagation along the jet does not necessarily have to be related to the NPV features. It could well be that NPV features just preferentially occur in large-scale flow situations with amplifying waves, without the NPV features being causally related to the amplifying waves.

**References:**

Pohorsky, R., M. Röthlisberger, C. M. Grams, J. Riboldi and O. Martius (2019), The climatological impact of recurring North Atlantic tropical cyclones on downstream extreme precipitation events, Mon. Wea. Rev., 147, 1513-–1532, doi:10.1175/MWR-D-18-0195.

Martius, O., C. Schwierz, and H. C. Davies (2010). Tropopause-level waveguides, J. Atm. Sci., 67(3), 866–879, doi: https://doi.org/10.1175/2009JAS2995.1.

---

## Author Comment (AC1)

**Response to Reviewers, Manuscript: egusphere-2024-382**

Alexander Lojko, Andrew C. Winters, Annika Oertel, Christiane Jablonowski, Ashley Payne

**1 Author Comments**

We greatly appreciate the very positive and constructive feedback on this manuscript. We have addressed all the comments made by the reviewers and editor. Some of the major changes we want to highlight include: Rewrite of the introduction section, rewrite of the first half of the discussion section and new/revised figures. Figure 3 has been revised and an additional centered composite figure, Figure 7, has been introduced to compare as an analogue against Figure 6. This has also involved removing Appendix figure 2, which has been revamped to become the current Figure 7.

**2 Response to Reviewer 1**

**2.1 Major Comments**

**Comment 1:** The motivation for this work and its scientific (and potentially societal) value needs to be stated more explicitly. In the abstract the authors motivate their work with (lines 2–5) "it is postulated that NPV may be relevant for the large-scale circulation as it has been observed to ..., accelerating jet stream winds and degrading numerical weather prediction skill". Similar statements can be found in the introduction. However, these statements presumably hold for any intense and diabatically influenced negative PV anomaly on the equatorward side of the jet, irrespective of whether the sign of its absolute PV values is negative or positive. In its current form, the manuscript thus does not yet fully convince me that NPV features are a subset of tropospheric negative PV anomalies that warrant special attention.

The authors could alleviate this concern by providing evidence (e.g., from previous studies) that NPV-jet interactions lead to more degraded forecast skills than interactions of benign negative PV anomalies (i.e., with low but positive absolute values) with the jet. Similarly, is there evidence suggesting that CAT is more vigorous/frequent for NPV-jet interactions than for benign "low PV"-jet interactions? If so, please state this more prominently. I acknowledge that, by theoretical arguments, NPV should not be stable and thus it is interesting intellectually to study NPV, which might help to further our theoretical understanding of PV dynamics. In my opinion this puzzling aspect of NPV is a perfectly valid reason for studying climatological characteristics of NPV, but it is not mentioned very prominently as a motivation for their work currently. Rather, the authors currently motivate their study with a need "to further understand how cloud processes impact jet stream dynamics" (line 28). If that is indeed the author's goal, then I believe they look at a subset of events that is unnecessarily narrow. I thus ask the authors to think again about why exactly climatological characteristics NPV–jet interactions in the North Atlantic warrant a dedicated study, and to be more explicit about the benefits and scientific value of this research.

> **Response:** We appreciate the constructive feedback on improving this work's motivation. We agree that the introduction would benefit from better distinguishing synoptic-scale bands of NPV from other large-scale negative PV anomalies, as there are certainly distinct differences. In response to this comment, the introduction has been elaborated upon by introducing a new paragraph on atmospheric instabilities and their relation to NPV. We discuss that mesoscale bands of NPV appear analagous to inertial instabilities, which are regions of anticyclonic vorticity that exceed the Coriolis parameter. As such, mesoscale NPV is characterized by a very intense anticyclonic vorticity. In contrast, large scale regions of negative PV anomaly (but not NPV) exhibit a much weaker magnitude of relative vorticity. The unique anticyclonic vorticity signature that is associated with NPV is also related to jet stream predictability and clear air turbulence (and it should now be more clear that these impacts are unique from large-scale negative PV anomalies).
>
> We are not aware of published research comparing forecast error growth between benign negative PV anomalies and banded mesoscale NPV features. However, in case-studies by Lojko et al., (2022), Hitchman and Rowe (2016), and Oertel et al. (2020) the banded NPV features are embedded in ridges (i.e., large-scale negative PV anomalies). The jet streaks in these studies that develop along the ridge closely align with location of the NPV feature. So while moving a negative PV anomaly closer to the jet stream should theoretically also lead to enhancements of wind speed along the jet stream, case-studies indicate that NPV is associated with distinct jet stream wind speed maxima. Regarding whether CAT is more frequent for NPV compared to benign negative PV anomalies, theory would suggest that NPV is

conducive to CAT compared to other types of large-scale, benign negative PV anomalies. Notably, the relative vorticity in NPV is an order of magnitude larger compared to the large-scale negative PV anomaly it is embedded in (Lojko et al., 2022, Hitchman and Rowe, 2016). Additionally, NPV closely relates to regions of negative absolute vorticity (relative vorticity greater than planetary vorticity), which primes the atmosphere to be highly sheared and more conducive in triggering CAT (Thompson and Schultz, 2021). We have leveraged these arguments in the introduction section. We have also added more details as to why there is a need for a climatological and composite assessment of NPV-jet interactions over the West Atlantic.

**Comment 2:** It is unclear how much of the signals shows in Fig. 6 (one of the central figures in this manuscript) are due to NPV-jet interactions and how much of these signals simply result from the difference in the large-scale flow during NPV-jet interactions and the climatology. Can the authors exclude the possibility that the composite anomalies in Fig. 6 would look very similar if one considered (instead of the NPV-jet interactions) time steps with just a large-scale flow situation similar to that during the actual NPV-jet interactions? If not, then the key conclusions about the effect of NPV-jet interactions on the upper-level dynamics, e.g., on lines 15–16 "The results show that NPV-jet interactions can in-situ strengthen the mid-latitude jet stream and could be dynamically relevant in enhancing downstream development, ..." do not seem justified.

The key question for me is how much of the signals depicted in Fig. 6 would be retained if the authors compared the variables depicted in Fig. 6 during NPV–jet interactions with time steps with a similar large-scale flow pattern occurring without NPV-jet interactions? I think this question needs to be addressed in some form, as otherwise it is very much unclear how much we learn about NPV-jet interactions from the authors composite analysis. As an inspiration, the authors could consider the study of Pohorsky et al. (2019), where these authors faced essentially an analogous problem when examining the interaction between recurving tropical cyclones (TCs) and the jet, which preferentially happen in an amplified flow situation, that, by itself already features considerable upperlevel PV and IVT anomalies compared to climatology. Pohorsky et al. addressed the problem by comparing composite fields during instances of actual TC-jet interactions with a climatology that was constructed from days without TC-jet interactions, but with similar large-scale flow configurations than during the actual TC-jet interactions (i.e., flow analogues). Please consider whether such an approach could be useful here too.

> **Response:** We agree that this is a key caveat of the results shown in Fig. 6. An issue with looking for similar large-scale flow situations as in Pohorsky et al., (2019) is that our algorithm necessitates the existence of a synoptic-scale NPV feature in order to complete the centering. Since Pohorsky et al., (2019) use a variables not related to the recurring TC object itself, they are able to compare analogue events with and without this weather system. The development of the algorithm in our study was designed to simply look for an NPV object in close proximity to a 2 PVU contour, without accounting for any specific dynamics based threshold (i.e., some minimum PV advection threshold).
>
> However, we believe that we can respond to the reviewer comment to compare against analogue situations. Since our algorithm has a distance based metric of the NPV feature to the 2 PVU contour, we have developed the analysis in Section 3.2 to also include an analogue comparison of NPV-jet interactions when the NPV feature is within 100 - 300 km to the jet stream (compared to the original < 100 km threshold). The 100 - 300 km threshold is chosen as an analogue as it gives virtually the same amount of NPV-jet interaction events as in the < 100 km threshold. Following similar methodological procedures outlined in Pohorsky et al., (2019), we assign each NPV-jet (100 - 300 km) event to one of the clusters from the < 100 km threshold using a Euclidean distance based metric (i.e., we ask the algorithm, how close are these NPV-jet (100 - 300 km) events to the centroid of the clusters from the NPV-jet (<100 km). We have thus expanded the centered composite analysis with a comparison of instances when synoptic-scale NPV features are located slightly further away from the 2 PVU contour (See new section 3.2.2). We have also removed a previous appendix figure that involved testing the NPV-jet interaction distance as the new figure in Section 3.2.2 is essentially a revamped and more detailed version of this old figure.
>
> Ultimately, the analogue analysis provides some slight additional evidence of the role of synoptic-scale NPV on influencing jet stream dynamics and the circulation patterns it typically occurs in. Although, as noted in the new analysis, we still need to be careful with describing how much the NPV influences PV gradient sharpening and enhancement of wind speeds as there are notable changes in the large-scale circulation when the synoptic-scale NPV feature is located slightly further away from the jet stream. We include some discussion of this in the results of Section 3.2.2.

We would also like to note that the purpose of the archetype case-study analysis is to provide the mechanistic detail as to how the synoptic-scale NPV features in each cluster interact with the jet stream. The archetype analysis is meant to aid in providing some more quantitative and mechanistic explanation as to the features we observe in the centered composites. Synthesis between the archetypes and the centered composites occurs in the discussion section. We have added an additional paragraph in the discussion section that better ties the archetype case-study results to the centered composites. Additionally, the part of the abstract highlighted by the reviewer has been edited slightly to better fit the scope of the results obtained.

**Comment 3:** I agree that (line 98) "using only one particular isentropic level can miss NPV features", but using only one pressure level has similar caveats. Since you anyways refer to the Röthlisberger et al. (2018) study: Why not choosing a seasonally varying isentropic level that follows the isentropic level of the jet? In that case you would have all the benefits of analyzing PV on isentropic levels whilst still analyzing data from only one level

**Response:** We appreciate the comment on using isentropic coordinates instead of isobaric levels. However, we believe that isobaric levels are an equally appropriate choice for our study. First, regarding the comment on seasonally varying isentropes: We think it is a lot more convenient to use a single level for all seasons that approximates to the tropopause compared to a vertical coordinate that varies substantially with season. Using seasonally varying isentropes will lead to 'jumps' in the height level of the chosen isentrope when moving from one season to the other. We also prefer using a vertical coordinate that is more vertically 'stationary' rather than one that can be more varied with respect to height, and thus, could be more likely to miss the location of the tropopause on a particular day that may exhibit notable height anomalies. We think 250 hPa is suitable for detecting the frequent occurrence of synoptic-scale NPV features (see for example, the appendix figure 1).

Moreover, we think that the choice of seasonally varying isentropic levels or single isobaric level should not impact the NPV-jet results. For a meteorology based argument as to why, isentropes are approximately parallel to isobars about the jet core due to thermal wind balance. In other words, if we take an isentropic level that resembles the 250 hPa level about the core of the jet stream, the meteorological field about the jet stream will virtually look the same whether we use isobars or isentropes. We have included this particular argument into the methods section.

An additional point to note is the computational bottleneck of redoing this study on isentropic levels. ERA5 isobaric levels are stored on the Copernicus climate data store whereas isentropic levels are stored on MARS tapes. Given that downloading tens of terabytes of data from MARS would take a substantial amount of time (and will very likely not change the meteorological outcome of the results), we do not think that it is possible to re-do this study in a timely manner using isentropic levels.

**2.2 Minor Comments**

**Comment 1:** Line 7: This may be a naïve question, but how sure are you that these synoptic-scale NPV bands in ERA5 are real and not an artefact of the IFS model or assimilation procedure employed in ERA5?

**Response:** This is a very important question to address. We do not doubt the existence of synoptic-scale NPV in the reanalysis. Theory and dropsonde observations (Harvey et al., 2020) alongside several different high-resolution numerical model simulations (Hitchman and Rowe, 2016; Oertel et al., 2020; Clarke et al., 2019) have found the development of synoptic-scale NPV. From personal evaluations, ERA5 has shown agreement with the location of synoptic-scale NPV features resolved by high-resolution simulations. Thus, ERA5 is currently the best available dataset to investigate the climatological frequency of NPV given the lack of adequate observations. Yet there is no published work at this time comparing the performance of reanalysis with respect to another dataset or high-resolution simulation, so we have little with compare against. We can however add that the climatology is likely unique to ERA5 (and reanalysis datasets that produced using a similar horizontal resolution). We include this statement in the discussion section of the manuscript.

Following some additional discussions, it is suspected that the DA system associated with ERA5 enables resolving the synoptic-scale NPV bands. The global models that are used to create the reanalysis appear to struggle with resolving the upscale growth of NPV (Lojko et al., 2022). Hence, the DA system in reanalysis is suspected to try correct for the location of NPV. Since we are also working over

a large domain (West-Atlantic) over a long time-period, reanalysis is essentially one of the only options to perform such a study (running high-resolution simulations is too expensive, and if DA is suspected to correct for NPV, then climate model simulations that parameterize deep convection may potentially not resolve synoptic-scale NPV and thus not be a useful dataset). This might be an interesting topic for further research. We hope that performing this climatological analysis of NPV leveraging reanalysis will provide useful context for future studies that may evaluate synoptic-scale NPV features using different datasets and models.

**Comment 2:** Line 9: Not yet clear what the 1.2% mean here. Maybe rephrase to "occur at >1.2% of all time steps at particular grid-points"?

**Response:** We have made the suggested change.

**Comment 3:** Line 34: "causing it to perturb polewards" $->$ "causing it to move polewards"

**Response:** The suggested change has been made.

**Comment 4:** Line 47: Delete "spatially"

**Response:** We have made the suggested change.

**Comment 5:** Line 63: "stream" instead of "steam"

**Response:** We have made the suggested change.

**Comment 6:** Line 74–75: Related to major comment 1. I don't see why this is an interesting research question. By the PV invertibility principle it is clear that if you move an intense negative PV anomaly (e.g., a NPV feature) towards large positive PV values then the flow inbetween accelerates. How could it be any different? Please rephrase this "objective" of the study once you have clarified how you would like to motivate your work.

**Response:** Following the suggestions made in the major comment section, we have better motivated the need to study synoptic-scale NPV features by relating these features to their unique inertial instability characteristics (See Line: 61-69). In contrast to large-scale negative PV anomalies (that do not have NPV), synoptic-scale bands of NPV denote a very strong anticyclonic relative vorticity signature, which makes up an important basis of our wave activity flux analysis.

**Comment 7:** Line 115: How exactly is the major axis length-scale calculated?

**Response:** Good question, an additional sentence has been added to the text to clarify what is considered the major axis length. "The major axis length is calculated from the two latitude, longitude coordinate pairs that are located furthest away from each other within the NPV label" (See Line: 130-132).

**Comment 8:** Comment on Section 2.2.1: I find the description of your algorithm very clear and understandable!

**Response**: Thank you very much for your comment!

**Comment 9:** Line 172: What is the distance metric that you use in the K-means clustering?

**Response:** The simple distance metric refers to calculating the shortest distance between a synoptic-scale NPV feature coordinate and a 2 PVU contour coordinate. The description of the distance metric has been rewritten slightly to improve clarity and reduce the use of jargon and uncertain language. Notably, instead of stating that we use a simple distance metric, it is now noted that we use the Haversine formula to compute the distance of an NPV feature to the jet stream. We also note that the interaction point (mid-point location between NPV feature and 2 PVU contour) is calculated using Euclidean distance. Euclidean distance is used in the latter calculations as it is less computationally expensive and for NPV features that are within 100 km to the 2 PVU contour (i.e., do not really need to account for Earth curvature).

However, following the reviewer's suggestion for computing analogues, we have edited our definition for the interaction point. The interaction point now represents the 2 PVU contour that is in closest proximity to the synoptic-scale NPV feature. We chose to fix the interaction point on the 2 PVU contour as opposed to the midpoint between the 2 PVU contour and NPV feature to have a more fair comparison of jet stream impacts when shifting the synoptic-scale NPV feature further away from the jet stream (See Line: 146 - 152).

**Comment 10:** Lines 167–188: Why exactly do you use the full PV field for the K-means clustering? This gives a lot of weight to stratospheric PV features (e.g., trough intensity, TPVs etc.), as they simply have larger PV values than tropospheric features. Alternatives focusing more on the large-scale flow pattern rather than stratospheric PV features exist: You could do the clustering on the natural logarithm of the PV field, whose gradients are linearly related to wind speeds (Martius et al., 2010) or you could consider a binary (stratospheric = 1, tropospheric = 0) field to emphasize the geometry of the large-scale flow, as in Pohorsky et al. (2019). Note: I'm not suggesting you need to do that, but perhaps discuss why you chose the full (latitude weighted) PV field for the clustering.

**Response:** Thank you for the great suggestions. Admittedly, we were less familiar with some of these more nuanced methods to 'filter' the PV field. Hence, the initial decision was to either use the full PV field or to use PV anomalies. Since we use a centered composite approach, PV anomalies are less appropriate as they are highly dependent on the latitude location of NPV-jet interaction. The full PV field is not impacted by anomalies due to the latitude location of NPV-jet interaction. Of course, the over emphasis on the stratospheric side of the jet stream is important to note, especially since as you mention, we are more interested in the large-scale atmospheric circulation patterns and to fairly weight the tropospheric side of the 2 PVU contour.

Regarding using the natural logarithm of the PV field. This method may not be appropriate to use in this study since all cases identify negative PV features in the PV field. So we cannot take the natural logarithm of a negative value. On the other hand, switching the PV field into binary (0 for equatorward of 2 PVU, 1 for poleward of 2 PVU) is a great idea to have a more fair weighting between the stratospheric and tropospheric side of the 2 PVU contour. We tested this option and ultimately ended up obtaining the same cluster patterns, hence we have chosen to stick with our current approach of using the latitude weighted PV field for identifying clusters. We have added a sentence to the methods "We tested the robustness of the clusters by using other methods to cluster NPV-jet interactions. We tested the use of a binary PV field (1 for stratosphere, 0 for troposphere) and obtained similar results". (See Line: 198-200)

**Comment 11:** Line 186: Delete "enhanced".

**Response:** We have made the change suggested.

**Comment 12:** Line 203: What is the —U-vector— exactly? I assume it is $|U - vector| = |(U, V)|$, i.e., the magnitude of the base state wind. Currently you just write "$|U - vector|$ is the wind speed". Please clarify.

**Response:** Thank-you for noticing this. We have made the additional change "$|U - vector| = |(U, V)|$" and state this term represents the magnitude of the base state wind.

**Comment 13:** Line 205: How are derivatives computed here?

**Response:** The xx, xy and yy derivatives are calculated in the same manner as the first order derivatives. For example, $\psi_x$ is calculated by taking the 'x' derivative of $\psi$. Subsequently, $\psi_{xx}$ is computed by taking the 'x' derivative of $\psi_x$. In terms of the technique used to compute the derivatives, we use spherical harmonics. A sentence has now been added specifying that spherical harmonics is used.

**Comment 14:** Line 207–208: What does "negligible change to PVU" mean exactly? Do you mean that the PV value of these features doesn't change much?

**Response**: Yes, that is the point we tried to convey. Upon re-reading this sentence, we have removed the "negligible change to PVU" in case this comes across as overly verbose. Additionally, this is already implied when mentioning that the synoptic-scale NPV can "persist quasi-adiabatically".

**Comment 15:** Line 234 and Fig. 3. You report > 12% occurrence frequencies of NPV features in the sub-tropics. I find this number surprisingly high, given that these features should not be dynamically stable and, theoretically, should decay very rapidly. I acknowledge your comment regarding the MCS observations, but I still think that more discussion of the magnitude of these numbers is warranted here, including some explanation or hypotheses as to why they are so frequent. Is our theoretical understanding so poor or are they perhaps partly artefacts of the IFS/data assimilation underlying ERA5?

> **Response:** This is a really good point and lends to the need for a more robust discussion of inertial instability in the introduction section. While most atmospheric instabilities should be quickly released, inertial instability demonstrates a capability to persist for several hours (Thompson et al., 2018; Oertel et al., 2020). The % frequency of NPV in the sub-tropics is similar, and in good agreement to the values observed in the inertial instability climatology conducted by Thompson et al., (2018) using ERA-Interim. We do find marginally higher frequency values in our study, which could be due to using PV rather than inertial instability and finer resolution in ERA5. Lee et al., (2023) recently produced a Northern Hemisphere climatology of CAT and also obtain very similar NPV frequency values over the Atlantic using ERA5. Our study also uses a more recent time-period compared to the two aforementioned studies, which may also influence NPV frequencies. We are not necessarily concerned about the frequency of NPV values computed in this study. It is well known in early works of atmospheric instabilities (namely inertial instability) that inertial instability is very frequent about the tropics as Coriolis is weak, and it is thus a lot easier for the atmosphere to become inertially unstable through rotational based wind perturbations (Tomas and Webster, 1997), which are particularly common in the tropics from disturbances associated with convection. The climatology of Thompson et al., (2018) of inertial instability also indicate its maximum occurrence over the tropics. Our work adds a new perspective with the focus on NPV and relating NPV features to inertial instabilities. We chose to refrain from a detailed discussion of this result in the manuscript to constrain the analysis on the mid-latitudes and specifically on NPV-jet interactions.

**Comment 16:** 16) Line 238: Brackets around the "Thompson et al (2018)" citation are missing.

> **Response:** Thank-you for noticing this, the change has been made.

**Comment 17:** Line 245 and Fig. 3b: Is it possible that we just see the occurrence frequency of jets in this panel? I'd appreciate some additional panels in Fig. 3, for instance (a) one showing the frequency of jet occurrence, (b) the conditional probability to observe a NPV-jet interaction provided a jet is close by, or (c), conversely, the conditional probability of observing a NPV-jet interaction provided an NPV feature is there.

> **Response:** Thank-you for the raising the concern here; however, Figure 3b is explicitly showing the frequency of synoptic-scale NPV features that are within 100 km to a 2 PVU contour (i.e., NPV-jet interaction frequency). We could plot jet frequency by showing the frequency of the 2 PVU contour (without accounting for NPV features), but the method that we use to identify the jet stream may not be appropriate to make such a plot. Please see the revised Figure 3 which plots the occurrence of jet streaks and relates the occurrence of jet streaks to synoptic-scale NPV features. Keep in mind that the conditional probability of observing an NPV-jet interaction interaction provided an NPV feature is there is technically 100%, since in order for an NPV-jet interaction to be identified, a time-step must always exhibit a synoptic-scale NPV feature within 100 km to the jet stream.

**Comment 18:** Line 251 and Fig. 3c: Along the same lines as above: Do we see these anomalies in Fig. 3c just because, by design, NPV-jet interactions need a jet streak, i.e., strong winds? Instead of the current Fig. 3c, I'd be more interested in seeing whether wind speeds are stronger during NPV-jet interactions than during instances when "only" a jet is nearby.

> **Response**: The NPV-jet interaction algorithm does not necessitate a jet streak. The two key components of the algorithm are that a synoptic-scale NPV feature is present in close proximity (100 km), to a circumpolar 2 PVU contour. We developed the algorithm intentionally so it is 'blind' to the kinematic dynamics. We are only searching for PV contours. In order to respond to this comment, we have revised Figure 3 to also plot the occurrence of jet streaks (which are defined in this figure as wind speed anomalies in excess of 40 m/s). Jet streak occurrence is plotted for NPV-jet interactions (<100 km), NPV-jet interactions (100-300 km) and for a more improved definition of climatology (see comment below for revised definition of climatology).

**Comment 19:** Line 255: Apologies for potentially misunderstanding something here, but I find this hypothesis test not very convincing. The null-hypothesis you are testing is that there are no wind speed differences between climatology and instances of NPV-jet interactions. However, by definition of NPV-jet interactions, there have to be strong winds (due to the jet). Thus, the result of increased wind speeds compared to climatology is expected by design.

**Response:** This leads back to the previous points made. The algorithm does not set a requirement for strong winds to be present. It does not use wind speed data at all. The only requirement is that there is an elongated NPV feature in close proximity to the 2 PVU contour. We do agree with the reviewer that increased wind speeds compared to climatology should be expected by design through PV theory (i.e., moving a negative PV anomaly in close proximity to the jet stream will sharpen the PV gradient and strengthen wind speeds). Although, we are dealing with a much messier atmospheric situation in which we likely already have a large-scale region of negative PV anomaly, with filaments of elongated NPV embedded within the large-scale negative PV anomaly. So it is still worth-while to comment on what is observed in such situations.

As part of the revisions to figure 3, we provide two additional comparisons against the occurrence of jet streaks when NPV-jet interactions occurr. First, we revised our definition for a comparison against climatology. We now randomly sample time-steps for each season such that they match the number of time-steps in each season for the NPV-jet interactions. We then run a bootstrapping procedure 100 times and compute the average to have a more fair climatological comparison (We are essentially weighting the climatology with respect to season). Second, we modify the NPV-jet interaction algorithm to compare the occurrence of jet streaks when searching for synoptic-scale NPV features within 100 - 300 km of the jet stream. These events are termed as NPV-jet (100-300 km) interactions. An explanation is provided in Line: 156-161. We hope that the revisions made to this figure now provides more in-depth detail on the occurrence of jet streaks when synoptic-scale NPV features are in close proximity to the jet stream.

**Comment 20:** Figure 4: Please add a panel on jet frequencies, such that we can see to what extent the results in Fig. 4f simply result from jet frequency variability

**Response:** We define jets using the 2 PVU contour. We mention in the manuscript that a 2 PVU contour is observed for all time-steps (except 4 instances where the algorithm did not detect a 2 PVU contour). Hence, plotting the frequency of 2 PVU contours would not explain the variability in Figure 4f. We would also need to plot the strength of jet stream winds (or the frequency of 'strong' jets) to argue why elongated bands of NPV are more common in some seasons compared to others. However, the climatological location and strength of jet stream winds is well documented in the literature already (Woollings et al., 2010; Iqbal et al., 2018) indicating that jet stream wind speeds are climatologically weaker during summer compared to winter. We also reference that strong jets provides a necessary environment to aid elongation of NPV features (Oertel et al., 2020). The results we currently have should provide sufficient detail that synoptic-scale NPV features that are in close proximity to a 2 PVU contour have a very notable seasonal variation.

**Comment 21:** Line 272: That's an interesting finding. Could you speculate about why there is larger inter-annual variability in NPV area in winter compared to summer?

**Response:** It would certainly be valuable to speculate a bit regarding the inter-annual variability of NPV in winter compared to summer. We have mentioned that frequency-wise, NPV-jet interactions are likely more frequent in winter as the jet is stronger and more conducive to the upscale growth of NPV onto larger-scales. Additionally, weather systems such as warm-conveyor belts that may drive convection in winter develop in close proximity to the jet stream. In contrast, summer-time convective weather systems can be locally forced (i.e., strong solar radiation on the surface) without needing large-scale frontal systems coupled to the jet stream, so NPV may develop in regions far away from the jet stream.

To speculate as to why interannual variability is much more pronounced in winter than summer, we may likely assume that this may be to do with the interannual variability of warm-conveyor belts (WCBs). Interannual variability of extratropical weather systems that exhibit embedded convection (as these represent large-scale environments in which NPV is likely to be generated in winter) will very likely influence the yearly distribution of synoptic-scale NPV features in winter. Other climatic drivers could also be important, such as the North Atlantic Oscillation signal which influence the location where trough-ridge couplets preferentially anchor themselves. Further research beyond the scope of the manuscript would be

needed to conclusively explain the variations in NPV inter-annual varability with respect to season.

**Comment 22:** Fig. 5 and lines 300–301: What do you mean with the statement "does not satisfy the false discovery rate"? Do you maybe mean "is not significant at alpha=xx, which corresponds to a maximum false discovery rate of 0.1 in Fig. 5b"? Bear in mind that the FDR test is nothing else than a tool to determine on which significance level one should reject the null hypothesis under scrutiny (see also next comment).

> **Response:** Yes, thank-you for noting this. We are intending to say that the FDR is not significant at our selected alpha.

**Comment 23:** Caption to Fig. 5: I find your description of how you determine the significance somewhat confusing. I appreciate that you employ the FDR test, but bear in mind that the FDR test (with e.g., maximum allowed FDR = 0.1) is just a tool to find an appropriate significance level so that less than 0.1*100% of the "discoveries" will be erroneous. That is, the FDR gives you a p-value threshold based on which you deem your results significant or not. Therefore, I think your statement " ... are statistically significant to the 98th percentile" makes no sense. Rather, you should state that you determine the significance level (I called it alpha in the comment above) based on the FDR test with a maximum false discovery rate of 0.1. Ideally, you would also state what that resulting significance level is for panels (a) and (b).

> **Response:** Thank-you for the correction on this. Admittedly, the caption is not well written. Following your suggestions, we have revised this section to read "The false-discovery rate (FDR) is applied to provide a more conservative estimate of statistical significance regarding observed trends. Grid-points that exhibit statistical significance at the 2% level ($\alpha = 0.02$) are shaded. Grid-points where the p value is small enough to satisfy the FDR criterion ($\alpha$ FDR = 0.1) are stippled. (See New Caption to Figure 5).

**Comment 24:** Figure 5c,d: Are there reasons to believe that the trends you observe are forced by increasing GHG concentrations or do you believe that they are part of internal variability? Some comments on that important question (also in the text) would be highly appreciated.

> **Response:** Great question, this is a point that certainly deserves some discussion in the manuscript. First, we do note that because we are only using a 22-year time-period, we are likely not dealing with a sufficiently long time-period to make detailed statements on the influence of anthropogenic climate change. At time-scales of a few decades, natural decadal variability influences could still be important.
>
> One speculation for a natural form of variability is the North-Atlantic Multi-Decadal Oscillation (AMO), which is highly influential in dictating the magnitude of sea surface temperatures (SST) over the Atlantic (Knudsen et al., 2011). Currently, the positive AMO index (which was increasing from 2000 - 2005), will favor the occurrence of enhanced North Atlantic SSTs, which is important for driving vigorous, large-scale diabatic weather systems along the West Atlantic (Minobe et al., 2008), which we suspect could be important for NPV generation. North Atlantic SST increases are also understood to be influenced anthropogenically, which will also play an important role in generating stronger weather systems by enhancing air-sea fluxes of latent heat.
>
> The 41 year trend analysis in clear-air-turbulence (CAT) by Prosser et al., (2023) and Lee et al., (2023) also speculate the role of climate change in influencing enhancements in CAT over the North Atlantic. The arguments that they cite is that the cooling of lower stratosphere and pronounced warming of the tropical upper troposphere is creating an environment more susceptible for CAT by sharpening the meridional temperature gradient. This of course provides a planetary scale explanation for increases in metrics related to clear-air-turbulence (which include NPV and metrics related to relative vorticity). We suspect that more regional anthropogenic effects could be influencing the CAT trend (and perhaps might be relevant to the NPV trend in our study?). Notably, WCBs are expected to increase in diabatic heating magnitude due to increased moisture availability in a warming world (Joos et al., 2023). This will favor an increase in (intense) precipitation events which hints at the possible occurrence of increased embedded convection in WCBs. These processes have the potential to form elongated NPV more frequently if WCBs allign well with the jet.
>
> Ultimately, trying to pin-point the specific climatic mechanism that is leading to enhancements in NPV-jet interactions and perhaps relating this to recent kinematic trends along the mid-latitude jet stream diagnosed by other studies is outside the scope of the work. Although, we do think that our study provides some useful mechanistic insight that an increase in the occurrence of mesoscale / synoptic-scale

NPV features could be a relevant source to observed enhancements of fast wind speeds in the West Atlantic (i.e., Shaw and Miyawaki, 2024). We have briefly discussed some of these points in the revised discussion section.

**Comment 25:** L316: "patterns" instead of "pattern".

**Response:** We have made the suggested change.

**Comment 26:** Discussion of Fig. 6: Related to major comment 2. Please make it very clear whether you consider the anomalies shown in Fig. 6 as related to the NPV-jet interaction events or whether they are merely features of the composite large-scale flow structure during these events.

**Response:** Thank-you for pointing this out. The large-sclae negative PV anomalies primarily result from the large-scale flow structure in which NPV features during NPV-jet interactions are embedded. The synoptic-scale NPV features are quite filamentary and the features get smoothed into the large-scale background negative PV anomaly, which is why the region of NPV frequency is relatively small in Figure 6. We have revised the text to make it more clear that the PV anomalies arise predominantly due to large-scale flow.

**Comment 27:** L350: Do you mean "upstream" and "downstream" part of the ridge or really "poleward or equatorward flank of the ridge" as you state currently. If the latter is the case, then I don't understand this sentence.

**Response:** Yes, we were referring to upstream and downstream. Thank-you for noticing this, the change has been made.

**Comment 28:** Figure 6g–l: Please choose the length of the reference vectors such that we can see the vectors in these panels better (i.e., make them larger).

**Response:** The vectors have been made thicker to make them more visible. The irrotational wind vectors have also been made slightly longer to also aid in visibility.

**Comment 29:** L 372–375: I think there is a verb missing in this sentence. As it is now, I don't understand this sentence

**Response:** Thank-you for noticing this. This particular section of the results has been slightly re-written to improve clarity (See Lines: 395-413).

**Comment 30:** Fig. 8d-I and discussion thereof: I appreciate that the authors went through the struggle of applying the WAF diagnostic, but please provide more framing here. Why exactly is this the right method to better understand how NPV features affect wave amplification and propagation here? What exactly do we learn from these analyses in terms of physical and mechanistic understanding?

**Response:** We agree with the reviewer that some extra framing on applying WAF for this research question would be helpful. WAF is an appropriate metric for evaluating NPV interactions with the jet stream as the WAF equation makes use of relative vorticity related terms that can be directly connected back to the NPV feature. This enables a clear connection between how relative vorticity in the NPV feature interacts with the large-scale flow. Since NPV can persist quasi-adiabatically (Oertel et al., 2020), it is appropriate to leverage a dry-dynamics equation such as WAF to explain how NPV interacts with the jet stream. It is hypothesized that dry dynamics can explain most of the interaction with the jet stream when NPV grows to larger scales., and we demonstrate the importance of dry dynamics through the WAF perspective, despite the generation of NPV being a result of predominantly diabatic processes (Oertel et al., 2020). These points are expressed in the revised section of the WAF methods section. We also make better mention that WAF is particularly useful since it can be used to explain modification to the large-scale flow of the jet stream. One caveat is that diabatic processes, such as radiation and turbulence likely are still active in these NPV features (and required to "remove" NPV due to material conservation of PV). Hence, some marginal diabatic impacts are not captured through our dry dynamics perspective. This links to our final conclusion suggesting a more elegant analysis that might better incorporate the PV framework. We have also revised the final paragraph of section 3.3.3 to more clearly summarize what

we learn about using the WAF equation to explain how NPV interacts with the jet stream.

**Comment 31:** Lines 617–618: Related to major comment 2: I'm not convinced your results allow drawing such a "causal" conclusion. The enhancement of wave activity propagation along the jet does not necessarily have to be related to the NPV features. It could well be that NPV features just preferentially occur in large-scale flow situations with amplifying waves, without the NPV features being causally related to the amplifying waves.

> **Response:** Thank-you for the comment, we agree that simply stating "...elongated NPV features can amplify jet stream and enhance the propagation of wave activity along the jet stream" may come across as an overly casual conclusion that disregards the importance of the large-scale flow in amplifying wave activity. We by no means dispute that the large-scale flow is the primary contributor to wave activity. However, the component based evaluation of the WAF terms does indicate that wave activity maximizes within the synoptic-scale NPV feature. Our results suggest that synoptic-scale NPV act as additional enhancers of wave activity, leading to NPV regions being associated with wave activity maxima. This result should be clear when examining the full wave activity flux plot in Section 3.3.3. Hence, we have confidence that our archetype results apply to many other cases when synoptic-scale NPV feature is embedded within similar, pre-existing amplified large-scale flow patterns.

> To clarify some key results in our analysis of the wave activity flux, we note that the small-scale terms related to shear (i.e., very strong anticyclonic vorticity) play an important role in leading to 'pockets' of wave activity maxima. As can be seen in the archetype plots, the NPV features are the regions where the strongest anticyclonic vorticity is embedded within. This makes sense since previous studies have drawn conclusions that elongated NPV features exhibit inertial instability (Rowe and Hitchman, 2016, Oertel et al., 2020). Hence, we believe our conclusion is valid (despite its lack of substance). We do want to acknowledge the reviewers comment and we have revised this section of the conclusion as follows to be more precise with the language that we use (See Lines: 746-751).

> Please see the revised conclusion section for additional context. We have now made it more clear that NPV serves as an additional amplifier of wave activity upon already highly amplified flow patterns, rather than suggesting it is a dominant source. We have also removed a paragraph from the conclusion that was a bit speculative about the magnitude of WAF. In doing so, we have focused on making the conclusion more concise.

**3 Response to Reviewer 2**

**3.1 Major Comments**

**Comment 1 - Line 64:** Rowe and Hitchman, 2016 are cited., but I wonder if enough explicit discussion is made of the potential role for inertial instability in linking the various stands of the current study? In particular, it might help clarify the meaning of "NPV-jet interactions", which doesn't seem to be pinned-down in the current text (unless I missed it). When discussing inertial instability, Holton comments that "near neutrality often occurs on the anticyclonic shear side of upper-level jet streaks". So, the argument here (if I understand it) is that NPVs develop where this near neutrality is tipped into instability when convection tilts vertical shear into the vertical. The "interaction" is then the adjustment (via ageostrophic winds) to remove the instability, which (slowly?) dissipates the NPV and shifts the jet northwards and strengthens it(?)

> **Response:** This is a great point, and as mentioned in the response to Reviewer 1, the introduction would benefit from a more in depth exploration of inertial instability and linking it back to synoptic-scale NPV. In response to your comment, we have furthered the breadth of the introduction section regarding atmospheric instabilities and their relation to NPV.

> The processes described by the reviewer for generating NPV (intertial instability) and its interaction with the jet stream follows from current literature understanding of intertial instability and its release (i.e., Thompson and Schultz, 2021). In the literature regarding inertial instability and in the literature for NPV, these two concepts tend to be treated exclusively. Motivated by Rowe and Hitchman (2016) and Oertel et al., (2020) who have synthesized these two concepts together, we provide a composite perspective of NPV, but evaluate it through dry dynamics (and relative vorticity) to illustrate through a composite perspective that NPV consistently exhibits properties that are consistent with an inertial

instability.

We do not comment on the dissipation of the NPV in this manuscript, particularly since the dissipation of the NPV through a PV perspective does not have any published literature on this topic (and is motivated as topic for future research in the conclusion of the manuscript). In Oertel et al., (2020), it is hypothesized that diabatic processes that contribute to positive PV tendencies may lead to the dissipation of the NPV. Whether these positive PV tendency diabatic processes are analogous to ageostrophic adjustment to remove inertial instability is yet to be determined. Although, there is literature describing the relevance of diabatic processes in triggering ageostrophic adjustment (i.e., Lagouvardos et al., 1993).

In the composite evaluation of NPV in this study, we are predominantly focused on: What does the large-scale circulation pattern look like? And whether the presence of synoptic-scale NPV near the jet stream leads to enhancement of certain meteorological variables and whether we can explain these enhancements due to the presence of NPV? In the revised introduction section, while we explore the concept of instabilities further and relate them to NPV features, we choose to limit details regarding dissipation and generation mechanisms simply because the results we present do not warrant such mechanistic detail. However, the additional citations we provide should provide an avenue for the reader to find out more on the life-cycle of NPV features.

We also want to note that by using a relative vorticity inversion and leveraging the WAF perspective to examine momentum terms (i.e., geostrophic components of WAF equation), that the anticyclonic vorticity associated with NPV notably contributes to the geostrophic wind profile along the jet stream. This would have been interesting to explore further (how much does NPV contribute to geostrophic and ageostrophic wind components), but the study is already filled with enough content with the purpose to highlight a general climatology of NPV features and a more broad overview of evidence that synoptic-scale NPV features are dynamically meaningful for the mid-latitude circulation. There are a lot more interesting topics that can be explored regarding inertial instability and NPV, but this will have to be left for a future research question.

**Comment 2 - [equation 1]:** There seems to be a few mistakes (typos?) in the printed equation. I think the second meridional-component should be $\psi_y^2$ - $\psi_{yy}$. Also, I would have expected to see the flux associated with the phase propagation "$C_u * M$" (as in Takaya and Nakamura, 2018). Later on, the various terms within Eq(1) are plotted. In Figure 9, to me it looks like "Part 1" and "Part 4" are the zonal components and "Part 2" and "Part 3" are the meridional components. Please could you clarify all these aspects?

**Response:** Thank-you for catching these errors. We did indeed mean to write: $\psi_y^2$ - $\psi_{yy}$. This change has been made to the equation. Regarding phase propagation, by not including a phase velocity, we must assume a standing wave. To compute $C_u$, we need to figure the phase speed of the feature of interest. We could apply a rough estimate for this term, but this may be inappropriate for the large amount of cases we have. More importantly, the analysis is performed at individual, instantaneous time-steps. At instantaneous time-steps, we only assume a standing wave, hence the phase propagation component can be dropped. If we were looking at a temporal average instead, then we could estimate a general propagation speed since the wave will propagate some distance over a sufficient period of time. However, just to re-state, there is no temporal component that we can work with when dealing with instantaneous time-step. In response to your comment, the methods section regarding WAF has now received more detail as to why we drop the phase propagation term. You are correct regarding the different "Part" terms in Figure 9. We have made it more clear now when referring to the different parts. Additionally, for consistency, we have replaced the use of x and y terms to meridional and zonal components of the WAF equation.

**Comment 3 - [line 312]:** "support the results presented here that NPV-jet interactions have approximately increased by a relative amount of 11% from 2000-2021..." I also wonder what the impact of increasingly better observations over the period is on the diagnosed NPV trends in ERA5. Either directly, or through better constraint of vertical wind sheer, subsequently tilted within the data assimilation model, etc? Even if the 22-year trends are "real", I suspect that more evidence is needed to distinguish these from natural low-frequency variability.

**Response:** This is certainly an important point to bring up. We do not believe a quantitative answer can be provided as to the impact of increased/improved observations from 2000 to 2022 on the trends in NPV and wind. Although, we include some relevant citations in the revised discussion section on this

topic. We also agree that more evidence is needed to distinguish the trend from natural low-frequency variability. While the scope of the paper is not to make a conclusive statement on whether the trend is anthropogenic or natural, it is certainly worth speculating. In response to your comments, we have included additional discussion of the impact of better observations and aspects of ERA5 data assimilation. We choose not to go into extended detail on the trend analaysis because as the reviewer mentions, there needs to be a sufficiently longer time-period for analysis to explore trends in more detail. Additionally, knowing the history of observational changes in ERA5 is not our expertise. However. we do provide relevant citations to provide further detail for the reader. One of the key points also clarified in the revised discussion section is that this analysis is unique to ERA5, and that the use of different types of datasets may reveal different results (hopefully this will motivate future study). See Lines (623-634).

**3.2 Minor Comments**

**Comment - line 95:** I think hourly data are available from the ERA5 archive (even if the authors choose to use 6-hourly).

**Response:** Yes that's correct. We have re-worded this part to say that we have downloaded the data at 6-hourly intervals.

**Comment - Line 102:** . . . are filtered TO ONLY INCLUDE THE WESTERN. . . (?)

**Response:** Thank-you, this change has been made.

**Comment - Figure 1:** I found this figure very helpful.

**Response:** Thank you!

**Comment - Lines 146-147:** "Over 80% . . . shorter than 200km". This seems a slightly confusing remark, since it is the NPV's with length scale larger than 1650km which are the focus of this study.

**Response:** We agree that this may come off as a bit confusing, but we are trying to highlight that the vast majority of NPV features are small, only a very small sub-set of NPV features have long length-scales. We chose to highlight the number 80% because it is the frequency of NPV features associated with the first bin (0-200 km length) of the histogram. We want to keep this statement, but we will introduce this statistic with a more easy to digest sentence: "The vast majority of identified NPV features in ERA5 have rather small length-scales". (See Lines: 164-170)

**Comment - Line 148:** in THIS study.

**Response:** Thank-you, the change has been made.

**Comment - Line 149:** "will smooth the PV field" (not the NPV field)?

**Response:** Correct, we have made the change.

**Comment - Line 155:** "Figure 2c . . . " This sentence seems poorly written.

**Response:** Agreed, thank-you for noticing this. We have got rid of this sentence.

**Comment - Line 156:** "decreases QUASI-exponentially with distance away from the jet stream, AS INDICATED BY DX/DY IN FIG.2C"(?)

**Response:** We have made the change to 'quasi-exponential'. We have decided to not include the latter suggestion as this point should be indicated by the caption of Fig. 2C. However, we have included an additional qualitative point to make this figure to be more easily interpreted. This section now reads: Figure 2c shows that the frequency of synoptic-scale NPV features decreases quasi-exponentially with

distance away from the jet stream. Most synoptic-scale NPV features are identified in very close proximity to the jet stream."

**Comment - Line 171:** "centred on the location of NPV-jet 'INTERACTION' leads to ..."

**Response:** We have made the suggested change.

**Comment - Line 186:** "enhanced led..." Typo?

**Response:** This is a typo and we ended up removing the word "enhanced".

**Comment - Line 209:** "PV", not "PVU"?

**Response:** Correct, this section has been rewritten slightly to improve clarity.

**Comment - Line 215:** The second terms IN THE SQUARE BRACKETS refer..."

**Response:** Thank-you, the change has been made.

**Comment - Line 232:** "frequency of NPV FEATURES are..."?

**Response:** Thank-you, the change has been made.

**Comment - Lines 235-236:** This gradient largely follows the general pattern of mesoscale convection..." Presumably also associated with the gradient in planetary vorticity?

**Response:** Thank-you for commenting on this section. We have modified this sentence to be more specific as to what was actually meant to be conveyed here. While we agree with the author's comment that the gradient resembles the planetary vorticity gradient, we actually intended to make a different statement here. Notably, we have changed the wording as follows: "The comparatively higher frequency of NPV over the coastal West Atlantic and Eastern North America aligns with regions of strong latent heating attributed to mesoscale convective systems". This sentence has been moved to the second paragraph to compliment other citations regarding warm SST's in this region and the high frequency of warm-conveyor belts. (See Lines: 280-285).

**Comment - Lines 245-246:** Are only the "interaction points" used in the plot, or the entire synoptic scale NPV features? I think it is the latter, but it may be useful to note that this is not the frequency of interactions, per se.

**Response:** Great point, it is indeed the latter and this is showing the frequency of synoptic-scale NPV features that are within 100 km to the jet stream (i.e., NPV-jet interaction events). We have made the definition more clear that NPV-jet interaction events refer to time-steps where synoptic-scale NPV features are within 100 km to the jet stream (See Lines: 123-124). Following your suggestion, we now explicitly state when discussing the results of Figure 3 and 4 that we are examining the frequency of synoptic-scale NPV features that meet the NPV-jet interaction criteria.

**Comment - Line 253:** "Wind speed DIFFERENCES are ..."?

**Response:** Thank-you, the change has been made.

**Comment - Line 255:** Change at the "98th percentile" to "at the 2% level"?

**Response:** We have made the suggested change.

**Comment - Lines 272-273:** "for particular years (NOT SHOWN)"?

**Response:** Thank-you, the above change has been made.

**Comment - Lines 324-327:** "The maximum frequency of NPV-jet events lie adjacent to the interaction point (ORANGE DOT) along the equator ward side of the jet...". I don't understand why the maximum frequency is not located at exactly the centre, since fields are all centred on the interaction point. Did I miss something or can this be better described here or in the methods section?

**Response:** The interaction point represents the mid-point between the 2 PVU contour and the NPV feature. Hence, the NPV should be located slightly below the interaction point. We agree that the methods section could be described better. The interaction point is now described in more detail in the methods section of the algorithm. Note that in the revised manuscript, we now compare NPV-jet interaction cases with NPV-jet interaction cases where the synoptic-scale NPV feature is also located 100 - 300 km away from the jet stream. This has led us to change the interaction point to be located on the 2 PVU contour for both distances (as opposed to between the 2 PVU and Synoptic-scale NPV feature).

**Comment - Line 329:** Change "positive PV gradient anomalies" to "large amplitudes in PV gradient"?

**Response:** We have chosen to stick with the language referring to 'positive PV gradient anomalies'. While we agree that the suggestion made by the reviewer is appropriate, we want to keep positive PV gradient anomaly for consistency with respect to the caption in Figure 6.

**Comment - Line 330:** Change "where the PV gradient rapidly sharpens towards much higher PVU values" to "where stratification is particularly large"?

**Response:** Great point, we do want to keep our original sentence because the PV gradient is in fact is much more sharpened poleward of the 2 PVU contour. We also want to keep the use of this language (i.e., referring to PV gradient sharpening) throughout the text. But we acknowledge that your point adds useful dynamical context. So we have added in brackets "where the PV gradient rapidly sharpens towards much higher PVU values (where stratification is particularly large)".

**Comment - Lines 340-347:** I suspect that the last sentence here is the main explanation. Could this paragraph be rewritten to incorporate this sentence in the discussion about lack of local moist processes?

**Response:** Thank-you for the comment, yes the final sentence does provide an overarching explanation. We have not necessarily rewritten this paragraph, although it is slightly modified. However, we have provided some additional discussion in the discussion section regarding the lack of moist processes (See Lines: 677-684).

**Comment - Lines 350-351:** "no matter if the synoptic-scale NPV is located on the poleward (fig. 6d-e) or equator ward (fig. 6f) flank'. I don't see this! Are we looking at the orange dot and thick grey contour?

**Response:** Thank-you for catching this, the citation of captions if a bit incorrect here. The point that we were originally making is that the positive wind speed anomalies (thick grey contours) that lie about the interaction point, are directly adjacent to where the maximum frequency of synoptic-scale NPV features is detected (the dark blue contour from the row above, fig. 6a-c). The composite indicates that these maxima in positive wind anomalies lie adjacent and slightly poleward to the maximum frequency of NPV features. We have reworded this section and better referenced the figure captions to make this point more clear.

**Comment - Fig. 6:** This is a very nice clear figure. Would it be useful to also show the composite-means of NPV somewhere in the figure?

**Response:** Thank-you! We initially tried plotting the composite-mean NPV in the first draft of this figure. Although, because of the number cases used, the composite mean PV field would end up being smoothed too much and it would be too difficult to see the NPV (aside from a tiny spec in close proximity to the interaction point). This is part of the reason why we perform the archetype case-studies in the following section. As an alternative, we currently plot the frequency of occurrence of NPV in each cluster. Notably, we currently plot the region where NPV occurs at least 50 % of the time in the NPV-jet interaction cases. This is plotted as a dark blue contour in Figure 6 a,b,c. Plotting the frequency of NPV is likely as close as we can get in showing the composite-mean NPV. We will therefore leave this figure as is.

**Comment - Fig. 6 (j) - (l):** Does this advection have the wrong sign? I see non-divergent wind in (k) and (l) crossing 2PVU in a sense that would leave the opposite sign for the advection(?)

**Response:** We have cross-checked the calculation for this section and we ended up getting the same result. We have cross-checked this composite with other studies that composite PV advection in ridges such as Steinfeld et al., (2019) and Winters (2021), and our composites are consistent with their results. The direction of the non-divergent wind vectors may be slightly impacted by the centered composite approach, which may make their interpretation less intuitive. Although, we believe that the direction of the non-divergent wind vectors are consistent with the observed sign of PV advection. Keep in mind that the vectors plotted are relevant to the particular grid-point. The PV field in the vicinity of a particular grid-point is being advected by the wind.

As an example for panel k and l. For k, wind vectors move from south west to north east in the ridge (roughly from west to east). In terms of PV about the interaction point, the PV field is being advected from west to east. So very positive PV in the trough is moving into the region of much more weakly positive PV in the ridge. In other words, positive PV is displacing the weakly positive PV ahead of it, leading to positive PV advection, consistent with the vector direction of the non-divergent winds. This argument similarly holds for l. Winds move from west to east. Low PV in the ridge is moving to displace much higher PV downstream, leading to negative PV advection. In case of interest, these concepts are discussed in a lot more detail in Section 3.2 of Winters (2021). However, we do want to ensure that the results presented in our manuscript are more intuitive. As a response to this comment, we cite the Steinfeld and Winters reference in the main text as references to highlight that the PV advection signals are congruent with the typical PV advection signals expected in amplified ridge environments.

**Comment - Fig.6 Caption:** Change "as a result of performing a centred composite" to "as a result of performing the clustering on fields centred on their respective interaction point (indicated by the orange dot)"? Change "and the PV gradient anomaly" to "and the magnitude of the PV gradient anomaly"? For panels (g-i), need to include "The thin black contour shows where the magnitude of ageostrophic wind exceeds 15m/s"(?) Change "in m" to "in panel m" as the m could otherwise refer to metres.

**Response:** Thank-you for the list of suggested changes here, we have implemented all of them in the main text.

**Comment - Lines 352-353:** Is this sentence a re-telling of the previous one? I find it confusing - maybe delete(?)

**Response:** Great point, this section has been re-worded and condensed.

**Comment - Line 359:** Change "that diagonally extend" to "tend to cross"?

**Response:** We have made the suggested change.

**Comment - Line 362:** Change "positive irrotational" to "positively divergent irrotational", or simply "divergent"?

**Response:** We have changed to "divergent".

**Comment - Line 367:** Change "(Fig. 6j-l)" to "(Fig. 6l)"?

**Response:** Thank-you, the change has been made.

**Comment - Line 373:** "Despite THE NPV-JET INTERACTION POINT being far from..."

**Response:** We have made the suggested change.

**Comment - Lines 384-384:** "This maintenance of the amplified WAF packet coincides with the maintenance of the more amplified ridge that is of comparable magnitude to the day of NPV-jet interaction". Is the link between the propagating ridge and WAF mainly associated with the PHASE propagation part of the WAF

(which seemed to be omitted in Eq.1)?

**Response:** Good question. As mentioned in the response to your major comment, the transient part of the WAF equation is not included. Hence, we are showing the total WAF contribution from just the 'standing-wave' component. There is very likely going to be additional contribution to the WAF magnitude if we were to include the transient component for this figure. Although, as argued in the response to the major comment, we do not include temporal evolution of WAF as our analysis is predominantly limited to the analysis of instantaneous time-steps, particularly in Section 3.3, which is the main purpose of leveraging the WAF equation. We wanted to plot WAF for the composite figures to add more context about what is occurring in the large-scale flow. The figure should still clearly illustrate that there is a rapid enhancement of wave activity, and that there may be even more if we included the transient component. This is a very important point to note however, so we have added a sentence stating: "Remember that only the stationary component of WAF (Eq. 1) is plotted here, and that there may be additional contribution from the transient component (Takaya and Nakamura, 2001)" (See Lines: 436-446).

We also argue that what we plot in the composite figure is useful as we still provide the reader with a "snapshot" of the 2D evolution of a stationary Rossby wave composite along the tropopause during NPV-jet interaction events.

**Comment - Line 367:** Change "(Fig. 6j-l)" to "(Fig. 6l)"?

**Response:** The change has been made.

**Comment - Line 403:** "synoptic-scale latent heating". Why is this necessarily synoptic-scale?

**Response:** Perhaps using "synoptic-scale" comes off as weird phrasing. We chose to use the word "synoptic-scale" due to the broad regions of latent heating that will be associated with the broad, positive IVT anomalies. This section has now been rewritten as follows: "away from the influence of broad regions of latent heat release".

**Comment - Line 408:** 'RELATIVE vorticity"?

**Response:** The change has been made.

**Comment - Line 419:** "As a final test (NOT SHOWN)..." ?

**Response:** The change has been made.

**Comment - Fig. 7:** Would it be useful to show the vertical wind shear?

**Response:** It is certainly worth considering vertical wind shear in examining the initial development and upscale growth of NPV features. Since the cases being evaluated already have a synoptic-scale NPV feature (formation and upscale growth has already occurred), we personally do not think that showing the vertical wind shear will contribute more to the analysis of these cases. We can confidently determine that there is pronounced horizontal shear in the environment given the strong anticyclonic vorticity associated with the NPV features in the case-study figures. We think plotting vertical wind shear would only be insightful to the reader if we were evaluating the NPV feature evolution. This idea is offered in the discussion section as an avenue for future work. Since we examine a single time-step here (no temporal evaluation of NPV formation and growth), we choose not to focus on this variable.

**Comment - Fig. 7 Caption:** "The vectors show the non-divergent winds obtained from the vorticity inversion". Does the inversion assume that relative vorticity is zero outside the grey box? If so, does this imply that the vectors outside the grey box may not be non-divergent?

**Response:** Correct, the inversion assumes relative vorticity is zero outside of the bounding box. Vectors outside of the box essentially represent the 'far-field' effect that the relative vorticity inversion inside the box exerts on the surrounding environment. In other words, the vorticity inversion following the method in Oertel and Schemm (2021) reconstructs the streamfunction at a point due to the relative vorticity in the finite domain (the bounding box). Since we are reconstructing the streamfunction, the vectors

outside the grey box should be non-divergent and these vectors are attributed to the relative vorticity within the dashed box. We would need the velocity potential computation to obtain divergent winds. Hence, vectors plotted in our results are only non-divergent.

While we do not do this step in our manuscript, to provide extra context for the reviewer, in order to obtain the vorticity and divergence wind contributions outside of the limited domain that are not due to the inversion (which also includes a harmonic component of the flow), the sum of the inverted wind components (divergent and non-divergent) must be subtracted from the full wind field. We recommend Section 2 of Oertel and Schemm (2021) for a comprehensive explanation of how the relative vorticity inversion works for both non-divergent and divergent wind components.

**Comment - Line 419:** "momentum FLUX term"?

**Response:** The change has been made.

**Comment - Fig. 8 top panel colour bar:** Should this be labelled to indicate that it is the magnitude of the non-divergent wind anomaly? Also the word "magnitude" seems to be needed in several places in the manuscript where a shaded/contoured field represents a vector field.

**Response:** True, the colorbar here refers to the non-divergent wind anomaly. We will keep the label as is, but clarify in the caption that this refers to the magnitude. ADDITIONALLY, WE WILL REVISE SECTION 3.3 TO ENSURE THAT WE CORRECTLY REFER TO THE MAGNITUDE WHEN APPROPRIATE.

**Comment - Fig. 9 bottom panels:** Please clarify which "Parts" relate to zonal and meridional WAF components

**Response:** The caption for this section has been revised to better define how each part relates to each component of the WAF equation. Additionally, instead of referring to the x and y component of the WAF equation, we have changed to the zonal and meridional component respectively. Some slight modifications have also been made to this part of the results section when referencing to the word 'parts', making sure this is only used when referring to one of the 4 components of the WAF equation we outline in Fig. 9 (which is now Fig. 10).

**Comment - Fig. 9 caption:** Change to "Section 2.2.3"? Also, I don't understand the rationale for excluding the normalisation by the base-state windspeed.

**Response:** Thank-you for noticing the section error, this has been fixed. We have revised the caption to improve clarity. The rationale for excluding normalization by the state wind is to focus more on the ageostrophic geopotential flux and momentum terms. The previous sub-sections in 3.3 serve to slowly build-up and evaluate the terms related to the NPV and the WAF equation. The base-state wind is not really relevant to the NPV feature and its inclusion ultimately acts to normalize the different parts of the WAF equation. It will not change the outcome regarding where the WAF parts are strongest. We have added a sentence to Section 2.2.3 noting the choice to exclude normalization by the base state wind.

**Comment - Line 479:** I think there is confusion between the terms with a "U" and a "V" in them, and which relate to x- and y-components.

**Response:** Thank-you for noticing this. We have modified this section of the manuscript to correctly refer to U and V terms. Additionally, we have made it more explicit that the x- and y- components refer to zonal and meridional components.

**Comment - Line 505:** Change "unique" to "based purely on"?

**Response:** We have made the change "only utilizes the ERA5 dataset". Following the major comments made, we have also used this section to explore the discussion of ERA5 data further. We have noted that the quality of observations has also improved over the time-period studied, caveats are discussed (See Lines: 623-634).

**4 Response to Editor**

**4.1 Minor Comments**

**Comment:** There is recent work by Prince and Evans (2022) that seems highly relevant and should be discussed

> **Response:** We have now included the work by Prince and Evans to provide some additional introduction on how wind shear serves to stretch negative PV onto larger scales and to further motivate why our manuscript is interested in evaluating synoptic-scale NPV features. In addition, we have found a few more relevant citations on negative PV and its upscale interactions which have been populated throughout the introduction and discussion section.

**Comment - Fig. 6:** Shows a centered composite but depicts outlines of continents, which gives an incorrect sense of geographical definiteness. It is preferable to remove the continent outlines and show the composite with pseudo-lat and pseudo-lon (0,0) denoting the center of the composite.

> **Response:** We have addressed this comment by implementing the pseudo-lon and lat approach. Additionally, we have removed continent outlines.

**5 References:**

Clarke, S.J., Gray, S.L. and Roberts, N.M., 2019. Downstream influence of mesoscale convective systems. Part 1: influence on forecast evolution. Quarterly Journal of the Royal Meteorological Society, 145(724), pp.2933-2952.

Harvey, B., Methven, J., Sanchez, C. and Schäfler, A., 2020. Diabatic generation of negative potential vorticity and its impact on the North Atlantic jet stream. Quarterly Journal of the Royal Meteorological Society, 146(728), pp.1477-1497.

Iqbal, W., Leung, W.N. and Hannachi, A., 2018. Analysis of the variability of the North Atlantic eddy-driven jet stream in CMIP5. Climate Dynamics, 51, pp.235-247.

Joos, H., Sprenger, M., Binder, H., Beyerle, U. and Wernli, H., 2023. Warm conveyor belts in present-day and future climate simulations–Part 1: Climatology and impacts. Weather and Climate Dynamics, 4(1), pp.133-155.

Knudsen, M.F., Seidenkrantz, M.S., Jacobsen, B.H. and Kuijpers, A., 2011. Tracking the Atlantic Multidecadal Oscillation through the last 8,000 years. Nature communications, 2(1), p.178.

Lagouvardos, K., Lemaitre, Y. and Scialom, G., 1993. Importance of diabatic processes on ageostrophic circulations observed during the FRONTS 87 experiment. Quarterly Journal of the Royal Meteorological Society, 119(514), pp.1321-1345.

Lee, J.H., Kim, J.H., Sharman, R.D., Kim, J. and Son, S.W., 2023. Climatology of clear-air turbulence in upper troposphere and lower stratosphere in the Northern Hemisphere using ERA5 reanalysis data. Journal of Geophysical Research: Atmospheres, 128(1), p.e2022JD037679.

Lojko, A., Payne, A. and Jablonowski, C., 2022. The Remote Role of North-American Mesoscale Convective Systems on the Forecast of a Rossby Wave Packet: A Multi-Model Ensemble Case-Study. Journal of Geophysical Research: Atmospheres, 127(24), p.e2022JD037171.

Minobe, S., Kuwano-Yoshida, A., Komori, N., Xie, S.P. and Small, R.J., 2008. Influence of the Gulf Stream on the troposphere. Nature, 452(7184), pp.206-209.

Oertel, A., Boettcher, M., Joos, H., Sprenger, M. and Wernli, H., 2020. Potential vorticity structure of embedded convection in a warm conveyor belt and its relevance for large-scale dynamics. Weather and Climate Dynamics, 1(1), pp.127-153.

Oertel, A. and Schemm, S., 2021. Quantifying the circulation induced by convective clouds in kilometer-scale simulations. Quarterly Journal of the Royal Meteorological Society, 147(736), pp.1752-1766.

Pohorsky, R., Röthlisberger, M., Grams, C.M., Riboldi, J. and Martius, O., 2019. The climatological impact of recurving North Atlantic tropical cyclones on downstream extreme precipitation events. Monthly Weather Review, 147(5), pp.1513-1532.

Prince, K.C. and Evans, C., 2022. Convectively Generated Negative Potential Vorticity Enhancing the Jet Stream through an Inverse Energy Cascade during the Extratropical Transition of Hurricane Irma. Journal of the Atmospheric Sciences, 79(11), pp.2901-2918.

Prosser, M.C., Williams, P.D., Marlton, G.J. and Harrison, R.G., 2023. Evidence for large increases in clear-air turbulence over the past four decades. Geophysical Research Letters, 50(11), p.e2023GL103814.

Rowe, S.M. and Hitchman, M.H., 2016. On the relationship between inertial instability, poleward momentum surges, and jet intensifications near midlatitude cyclones. Journal of the Atmospheric Sciences, 73(6), pp.2299-2315.

Shaw, T.A. and Miyawaki, O., 2024. Fast upper-level jet stream winds get faster under climate change. Nature Climate Change, 14(1), pp.61-67.

Steinfeld, D. and Pfahl, S., 2019. The role of latent heating in atmospheric blocking dynamics: a global climatology. Climate Dynamics, 53(9), pp.6159-6180.

Takaya, K. and Nakamura, H., 2001. A formulation of a phase-independent wave-activity flux for stationary and migratory quasigeostrophic eddies on a zonally varying basic flow. Journal of the Atmospheric Sciences, 58(6), pp.608-627.

Tomas, R.A. and Webster, P.J., 1997. The role of inertial instability in determining the location and strength of near-equatorial convection. Quarterly Journal of the Royal Meteorological Society, 123(542), pp.1445-1482.

Thompson, C.F. and Schultz, D.M., 2021. The release of inertial instability near an idealized zonal jet. Geophysical Research Letters, 48(14), p.e2021GL092649.

Thompson, C.F., Schultz, D.M. and Vaughan, G., 2018. A global climatology of tropospheric inertial instability. Journal of the Atmospheric Sciences, 75(3), pp.805-825.

Winters, A.C., 2021. Kinematic processes contributing to the intensification of anomalously strong North Atlantic jets. Quarterly Journal of the Royal Meteorological Society, 147(737), pp.2506-2532.

Woollings, T., Hannachi, A. and Hoskins, B., 2010. Variability of the North Atlantic eddy-driven jet stream. Quarterly Journal of the Royal Meteorological Society, 136(649), pp.856-868.

---

## Author Response (AR2)

**Second Response, Manuscript: egusphere-2024-382**

Alexander Lojko, Andrew C. Winters, Annika Oertel, Christiane Jablonowski, Ashley Payne

**1 General Comments**

Thank you for the positive feedback on the revised manuscript. One remaining major concern brought up was the need for further quantification of the kinematic influence of NPV-jet interactions. We have addressed this concern in two key ways. First, we follow the reviewer's and editor's advice to provide a more quantitative approximation of NPV's influence on the jet stream. Second, we hope to have better motivated the importance of some of the qualitative results regarding NPV-jet interactions mechanisms, which we stress are important results in their own right.

Here are the key changes made that we hope provide additional insight on the kinematic impact of NPV on the jet stream.

1. More intuitive relative vorticity inversion to estimate the added impact of NPV on the jet stream. We discuss the approach in more detail in response to the major comment from the reviewer. We hope that this approach is now a satisfying (or at least a reasonable estimate) for how much NPV contributes to the large-scale flow.

2. While there is certainly a desire to quantify these interactions, we stress that there are limitations to how much appropriate quantification we can do with reanalysis. The goal of the study has always leaned towards a more qualitative and exploratory evaluation of NPV-jet interactions through a composite framework. Specifically, highlighting some of the key circulation patterns and mechanisms we observe during NPV-jet interactions. We have added a schematic to the end of the results section to remind the reader of the dry dynamics mechanisms identified during NPV-jet interactions found in this study (and emphasize the fact that NPV-jet interactions appear to be a dry dynamics interaction).

3. Other minor personal changes have been added (slight rewrite of the discussion and conclusion to better emphasize the mechanistic results of NPV-jet interactions). These changes can be best seen in: L721-730 and L766-774. This also includes minor grammatical changes in some parts of the manuscript. We have also added relative vorticity contours to the composites to make it clear that regions of NPV are regions where anticyclonic vorticity is much stronger than background flow. This serves as additional evidence for the uniquely strong vorticity circulation associated with NPV compared to relative vorticity associated with larger-scale, benign negative PV anomalies. This property should also be more clear in the case-studies following a slight adjustment of the colorbar to show the contrast between anticyclonic vorticity within NPV and surrounding ridge.

**2 Response to Reviewer 1**

**2.1 Major Comments**

**Comment 1:** My main concern follows similar lines to RC2 major comment 2, namely that some of the signals attributed to NPVs in this work would be present even if the PV value did not go below zero (i.e. it was small, but not a NPV feature). RC2 focussed on Fig 6 (and their concern has been at least partially addressed by including a discussion of the 100-300km cases in Fig 7), but in my view the same concern holds for the relative vorticity inversion of Fig 8 (new numbering). Specifically, you invert (if i've understood correctly) the part of the relative vorticity field that lies within the NPV contour and then claim (L517) that 'the anticyclonic vorticity associated with NPV appeared to contribute to approximately 50% of the total non-divergent wind field'. I fear this may mislead readers. There is always negative relative vorticity on the southern side of the jet (by definition of vorticity) so claiming it is all 'associated with the NPV' is not a fair assessment. Indeed, one could argue that the NPV feature should only be associated to that part of the (absolute) vorticity field that is negative, and I suspect inverting this (much smaller) anomaly would produce a much smaller estimate of the impact on the wind field. I note that you do include a caveat along these lines towards the end of the discussion section (L675) but feel the conclusions drawn in section 3.3.1 need adjusting to reflect this important caveat, or even better, the alternative inversion calculation presented as well for comparison. I suspect similar is also true for the inferences drawn from the WAF calcuations in sections 3.3.2 and 3.3.3, but am less familiar with that diagnostic to know for sure. Please reread those sections with this concern in mind and adjust the

language used as appropriate.

**Response:** We appreciate the comprehensive comment. As mentioned in the general comment, quantifying the dynamical relevance of NPV on the jet stream is complicated through ERA5 alone without additional high-resolution data and/or more complex methodological tools. The study intends to focus on more qualitative insight into the dynamical importance of NPV features on the large-scale flow, specifically through a climatology and by identifying mechanisms through which NPV interacts with the jet stream (i.e., highlighting the importance of dry dynamics interactions) with some complementary quantitative insight. Following the comments made, we do have an idea to improve our quantitative insight into how NPV influences the jet stream through relative vorticity inversion.

First, we agree with the reviewer, the 50% approximation is simplistic and does not do a good job at teasing out the added impact of relative vorticity associated with NPV on the jet stream. Personally, the section left a lot to be desired. As a response, L529-539 (Now L539-549) has been rewritten following a more quantitative evaluation of the NPV feature and its influence on the jet.

The reviewer suggests inverting the inertial unstable part of the NPV. This has been tested previously. Because synoptic-scale NPV is near-zero PVU (and just slightly inertial unstable), the inertially unstable part of the anticyclonic relative vorticity only contributes about 1 m/s to the total wind field (Fig. 1). However, this is not a fair attribution of the influence that NPV has on the large-scale flow either. This is because NPV has already been identified as having a much stronger anticyclonic vorticity compared to the background vorticity associated with the broader ridge within which it is embedded. Put another way, the inertially unstable part of the NPV feature is only a small component of the anticyclonic vorticity field associated with NPV.

It is better to identify the anticyclonic vorticity associated with NPV by subtracting the background flow from the total anticyclonic vorticity within the NPV feature, since the NPV feature ultimately represents a relative vorticity anomaly with respect to the background flow. This logic also follows from Harvey et al., (2020). Assume we have a pre-existing large-scale flow (i.e., a base-state ridge with no diabatic heating occurring). When heating occurs, PV is reduced in the upper-troposphere and perhaps we may get the generation of an NPV feature. A fair attribution would thus be found by taking the difference between the vorticity associated with the NPV feature and the base-state environment at the earlier time. This method follows a similar attempt to estimate the wind field associated with smaller-scale NPV features in Oertel et al., (2020). Our new method to estimate the circulation associated with NPV relative to the base-state is now outlined in the Methods section (Line 276-286).

We also provide an illustrative example of how this method is applied for synoptic-scale NPV features in the appendix section to help the reader more easily interpret our methodological approach. We stress that this method is only able to provide a rough estimate of the circulation associated with NPV. However, we hopefully provide a far more interesting way to quantify NPV that extends from previous literature compared to our previous version.

As for the WAF calculations, following the introduction of the new schematic at the end of the results section (New Figure 11), we want to emphasize that the WAF analysis is meant to primarily highlight the mechanistic aspects of NPV-jet interactions and not necessarily provide a detailed quantitative estimate of how much NPV contributes to the total WAF equation. We mainly want to show that NPV-jet interactions can be described using dry-dynamics and that they enhance wave activity via: Vorticity dipole interaction and due to strengthening the ageostrophic flux of geopotential by the shear accompanying the anticyclonic vorticity. We feel that the mechanistic insight is sufficient and we do not want it to get lost amid further quantification. We have however revised the conclusion section to better highlight these findings.

Further quantification will be more suitable for high spatial and temporal resolution case-study approaches, which may enable the application of more sophisticated methods (such as trajectory based analysis and outputting physics tendencies) to better quantify how heating influences circulation associated with NPV in future studies.

[Figure]

[Figure]

Figure 1: Inverting just the anticyclonic vorticity that is greater than coriolis parameter in the NPV feature of interest for the 2014-12-09T18 case. The boundary for the relative vorticity inversion is the NPV feature itself at 250 hPa.

**2.2 Minor Comments**

**Comment - L24:** As written, the third sentence of the Introduction overplays the role of diabatic processes relative to dry dynamics. Suggest a minor change to something like 'These eddies, which can include a strong contribution from cloud diabatic heating processes, act to perturb the jet stream's large-scale circulation away from its base state.

**Response:** Thank-you, we have made the suggested change.

**Comment - Line 102:** Does the irrotational wind cause the outflow air to move just polewards, or in all directions?

**Response:** The irrotational wind will advect the outflow radially outwards in all directions. However, we will generally see the preferential acceleration of outflow towards the jet due to the lower inertial stability on the equatorward side of the jet, which favors venting outflow poleward into the jet rather than in other directions.

But for the purpose of perturbing the jet stream (and since the irrotational outflow is established on the equatorward side of the jet stream), we just wanted to imply that the 'important' component of the advection occurs polewards. We have refined the sentence to make this more clear: "The irrotational wind field established by large-scale cloud systems make an important contribution to the poleward advection of the outflow and subsequent wind speed enhancements along the jet stream".

**Comment - L72:** How can the anticyclonic relative vorticity associated with NPV be an order of magnitude larger than the PV anomaly, when they have different units? Please clarify.

**Response:** The wording has been edited to make more sense: "When NPV features are within large-scale negative PV anomalies like ridges, the anticyclonic relative vorticity from the NPV has been observed to be an order of magnitude greater than the vorticity from the surrounding large-scale negative PV

anomaly". The point we tried to make is that the anticyclonic vorticity associated with NPV has larger magnitude than anticyclonic vorticity associated with a benign large-scale negative PV anomaly.

**Comment - L140:** Could you explain what you mean by a 10 point Gaussian smoother? What is the half width of the smoothing kernel?

> **Response:** Smoothing is done with the Python package called MetPy. The Guassian smoother is a moving-average low-pass filter. The moving average is computed over 10 grid-points, hence the '10-point' smoother. We choose not to eidt anything in the manuscript since the exact details of the smoothing approach do not necessarily effect the reproducibility of the study, nor is it an important part of the methods.

**Comment - L241:** What is the streamfunction anomaly computed relative to? The seasonal mean streamfunction?

> **Response:** Correct, it is computed from the seasonal mean. We have added an additional sentence: "Note that the anomaly is computed from the seasonal climatological mean". (Now L243).

**Comment - L265:** Please describe the relative vorticity inversion calculation in more detail. What domain is the calculation performed over? If it's not a global inversion, what boundary conditions are applied?

> **Response:** We have added additional detail regarding the domain of the calculation, boundary conditions and that the inversion is computed regionally to the methods section of the paper. A more detailed summary of the inversion process is also summarized herein:
>
> The inversion of vorticity is based on partitioning the full horizontal flow into a non-divergent flow component attributed to the vorticity in a spatially confined domain, an irrotational component attributed to its divergence, and the effect of the larger-scale environmental flow outside of the domain. Using a Green's function approach facilitates to attribute a horizontal flow field to the vorticity within a limited domain. Since our paper focuses on NPV features associated with a vigorous relative vorticity signature, we exclusively focus on the inversion to obtain non-divergent winds.
>
> For readers that need more detail on the approach, we note that we provide a reference to Oertel and Schemm (2021), who have a paper dedicated to deriving the relative vorticity inversion.

**3 Response to Reviewer 2**

**3.1 Minor Comments**

The authors have made substantial improvements to the manuscript, particularly in the introduction. They have also spent considerable time addressing most of my comments, which I greatly appreciate. I would say that the advection in Fig. 6k still looks to have the wrong sign to me(!), since grad(PV) will be normal to the PV=2 contour. Maybe it is confusion over whether advection is -v.grad(PV) or +v.grad(PV)? The authors could also change "latitude and longitude" to "longitude and latitude" on line 242, to be consistent with the ordering of x and y.

> **Response:** Thank-you for your comment! First, we have changed "latitude and longitude" to "longitude and latitude" in line 242.
>
> Regarding the PV advection plots, we have cross-checked to ensure that the PV advection sign is correct. However, we assume that there might be a slight artefact due to the centered composting approach where the wind vectors appear to be pointing from low to high values of PV for panels j and k, which may give the impression that the sign of PV advection should be the opposite as implied by the reviewer. However, we have cross-checked that the PV advection signal is correct. We suspect that the orientation of the vectors may contributing to some of the confusion, but this is likely that each centered composite may have notable variations in the tilt of the ridge, which may influence the direction of the wind vectors. Overall, we think no changes are needed as we make appropriate references to other studies following the previous round of revisions to validate the PV advection results.

**4    References:**

Harvey, B., Methven, J., Sanchez, C. and Schäfler, A., 2020. Diabatic generation of negative potential vorticity and its impact on the North Atlantic jet stream. Quarterly Journal of the Royal Meteorological Society, 146(728), pp.1477-1497.

Oertel, A., Boettcher, M., Joos, H., Sprenger, M. and Wernli, H., 2020. Potential vorticity structure of embedded convection in a warm conveyor belt and its relevance for large-scale dynamics. Weather and Climate Dynamics, 1(1), pp.127-153.

Oertel, A. and Schemm, S., 2021. Quantifying the circulation induced by convective clouds in kilometer-scale simulations. Quarterly Journal of the Royal Meteorological Society, 147(736), pp.1752-1766.

---

## Author Response (AR3)

**Third Response, Manuscript: egusphere-2024-382**

Alexander Lojko, Andrew C. Winters, Annika Oertel, Christiane Jablonowski, Ashley Payne

**1 General Comments**

Thank you for the feedback. We have addressed all the remaining minor points brought up by the reviewer.

**2 Response to Reviewer 1**

**2.1 Minor Comments**

**Comment - Paragraph starting L539**: If I've understood correctly, the new methodology attributes to the NPV feature the change in relative vorticity between the time of NPV identification and that of a background flow of a region slightly upstream at an earlier time. To my mind this is much better than inverting the full relative vorticity anomaly as was done before, because it represents (at least some approximation to) the influence of diabatic heating over the preceding period. However:

> **Response:** Thank-you, your summary of our new methodology is also correctly interpreted.

**Comment:** I do not think it is fair to attribute this wind to 'the NPV feature' which to me suggests it is fully associated with that part of the absolute vorticity field that is below zero. As you now point out, this is very small. I do agree that this quantity is some approximation to the influence of the diabatic heating which caused the negative PV. Can you rephrase?

> **Response:** This is a fair comment, we have rephrased much of this paragraph to make it clear that we are not inverting just the anticyclonic vorticity associated with NPV, but we are in fact inverting the anticyclonic vorticity field that we suspect is attributed to diabatic heating. We reiterate this point in the results section and make it clear that the inversion not only includes the inertially unstable part, but some additional part with respect to the pre-defined base-state prior to heating occurring. In addition, we make note of the inversion results from just the inertially unstable component for the interest of the reader.

**Comment:** There are clearly several approximations going on here which I think need spelling out more clearly, to aid the reader. First, relative vorticity is not conserved by adiabatic flow. Second, the shape of the ridges evolve over the preceding period. Thirdly, you omit the influence of diabatic heating in any regions of positive PV (which I could imagine may counteract the acceleration associated with the negative PV region).

> **Response:** We agree with the suggestion to clearly state some of the approximations made. We have added the approximations to the methods section outlining the inversion process and reference in the paragraph of L539 to check the methods regarding assumptions made. We also make it further clear that relative vorticity inversion omits diabatic heating influence. Although, this is a useful property to have due to acknowledging that synoptic-scale NPV features are quasi-adiabatic and that we are interested in their kinematic properties because of this.
>
> We have added the following section to the methods section: "To clarify some key assumptions made during two-dimensional relative vorticity inversion and coordinate shifting process: First, any diabatic effects that might have induced positive PV tendencies within the region considered for the inversion are neglected. Second, it is important to clarify that relative vorticity is not conserved by adiabatic flow. Lastly, when performing the temporal shifting in coordinate space, it is important to remember that the shape of ridges evolves over the preceding period, which may influence the wind estimates derived from the inversion."
>
> We do want to respond to the reviewer's further thoughts on the omission of diabatic heating. We suspect that any potential diabatic heating will not impact the circulation obtained from the relative vorticity inversion in the three cases. To expand on this point, all of our cases are associated with anticyclonic relative vorticity within the ridge (there is occasionally very minimal, highly localized cyclonic relative vorticity within the ridge, but its spatial scale is much smaller than the NPV feature's length scale). Unless there is sufficient diabatic heating to cause positive PV tendency leading to subsequent development of a mesoscale region of cyclonic relative vorticity to the right side of the NPV feature, we

do not expect there to be a deceleration to the flow associated with the NPV feature.

To make this point clear, we highly suggest checking Figure 6 and Figure 7 in Oertel and Schemm (2021), who examine smaller, mesoscale relative vorticity dipoles produces by convective-scale heating. The figures very clearly illustrate the deceleration phenomena. In their case-study, an anticyclonic relative vorticity pole lies to the left of a cyclonic relative vorticity pole. The resulting inversion leads to a deceleration of the flow between the dipoles. In our study examining synoptic-scale NPV features, the ridges in the three cases are essentially entirely defined by anticyclonic relative vorticity. There is no cyclonic relative vorticity to decelerate the flow to the right of the NPV feature. However, large swaths of cyclonic relative vorticity lie to the left of the NPV feature (on the polar side of the 2 PVU contour). Consequently, the circulation between synoptic-scale NPV and cyclonic vorticity on the poleward side of the jet stream will lead to acceleration of the flow about the jet stream (as shown by our vector arrows in the schematic, Figure 10).

**Comment - L140:** Why do you claim this estimate is 'conservative'? Please clarify.

**Response:** Thank-you for catching this. We were intending to suggest that this is an approximate estimate. But we can see how the use of the word conservative may imply otherwise. We have removed "conservative" and rephrased to "approximate". This also includes removing the section "Although the vorticity inversion provides a conservative estimate".

**3   References:**

Oertel, A. and Schemm, S., 2021. Quantifying the circulation induced by convective clouds in kilometer-scale simulations. Quarterly Journal of the Royal Meteorological Society, 147(736), pp.1752-1766.